# A Bit of Freedom Goes a Long Way: Classical and Quantum Algorithms for Reinforcement Learning under a Generative Model

## Abstract

We propose novel classical and quantum online algorithms for learning finite-horizon and infinite-horizon average-reward Markov Decision Processes (MDPs). Our algorithms are based on a hybrid exploration-generative reinforcement learning (RL) model wherein the agent can, from time to time, freely interact with the environment in a generative sampling fashion, i.e., by having access to a "simulator". By employing known classical and new quantum algorithms for approximating optimal policies under a generative model within our learning algorithms, we show that it is possible to avoid several paradigms from RL like "optimism in the face of uncertainty" and "posterior sampling" and instead compute and use optimal policies directly, which yields better regret bounds compared to previous works. For finite-horizon MDPs, our quantum algorithms obtain regret bounds which only depend logarithmically on the number of time steps $T$, thus breaking the $O(\sqrt{T})$ classical barrier. This matches the time dependence of the prior quantum works of Ganguly et al. (arXiv'23) and Zhong et al. (ICML'24), but with improved dependence on other parameters like state space size $S$ and action space size $A$. For infinite-horizon MDPs, our classical and quantum bounds still maintain the $O(\sqrt{T})$ dependence but with better $S$ and $A$ factors. Nonetheless, we propose a novel measure of regret for infinite-horizon MDPs with respect to which our quantum algorithms have $\text{poly} \log T$ regret, exponentially better compared to their classical counterpart. Finally, we generalise all of our results to compact state spaces.

## 1 Introduction

Reinforcement learning (RL) (Sutton & Barto, 1998) is a subfield of machine learning that studies how an agent can properly interact with a dynamical environment in order to maximise some type of reward. Markov decision processes (MDPs) (Puterman, 2014) serve as the most commonly used framework for modeling such agent-environment interactions. In its most general form, a discrete-time MDP $M$ can be described by a tuple $\langle \mathcal{X}, \mathcal{A}, p, r \rangle$, where the state space $\mathcal{X}$ is the set of possible states the environment can assume, the action space $\mathcal{A}$ is the set of possible actions the agent can choose, the reward function $r : \mathcal{X} \times \mathcal{A} \to [0, 1]$ yields a reward to the agent when interacting with the environment, and the stochastic kernels $p(\cdot|x, a)$ denote the transition probability to another environment state given the current state-action pair $(x, a)$. At any (discrete) point in time, the environment is in some state $x_t \in \mathcal{X}$ and the agent must choose an action $a_t \in \mathcal{A}$, after which they receive a reward $r(x_t, a_t)$ and the environment randomly transitions to a new state $x_{t+1}$ according to $p(\cdot|x_t, a_t)$. The agent chooses an action through a *policy* $\pi = (\pi_t)_t$, which is a sequence of *decision rules* $\pi_t$, which in turn are probability distribution over actions $a \in \mathcal{A}$ given $x \in \mathcal{X}$. The practical utility of MDPs lies in their ability to model decision-making in complex environments (Åström, 1965; Hu & Yue, 2007; Sato & Zouain, 2010; Bäuerle & Rieder, 2011; Feinberg & Shwartz, 2012; Bennett & Hauser, 2013; Chen et al., 2014; Steimle & Denton, 2017; Natarajan & Kolobov, 2022), with successful applications to many problems (Sutton & Barto, 1998; Szepesvári, 2010; Bertsekas, 2012; 2022).

There are several types of MDPs in the literature (Puterman, 2014), the most common ones being *finite-horizon* MDPs, *infinite-horizon discounted* MDPs, and *infinite-horizon undiscounted* MDPs.

In this paper, we shall focus on finite-horizon and infinite-horizon undiscounted MDPs. For the former (also known as tabular MDP), the agent interacts with the environment for a finite pre-determined number of time steps $H$, which is called the horizon. The standard criteria to evaluate the performance of the agent is the *expected total reward* $V_1^\pi(x)$ over the horizon $H$ when executing policy $\pi$ with initial state $x \in \mathcal{X}$,

$$V_1^\pi(x) := \mathbb{E}_{x_1=x}^\pi \left[ \sum_{t=1}^H r(x_t, a_t) \right].$$

A policy $\pi^\varepsilon$ is *$\varepsilon$-optimal* if $V_1^{\pi^\varepsilon}(x) \geq V_1^\pi(x) - \varepsilon$ for all $\pi$ and $x \in \mathcal{X}$, and the optimal total expected reward is $V_1^*(x) = \sup_\pi V_1^\pi(x)$. In infinite-horizon (undiscounted) MDPs, the agent can interact with the environment for an infinite amount of time steps, meaning that $H = \infty$. One of the standard criteria to evaluate the performance of the agent is the *average reward* (or gain) $g^\pi(x)$ when executing policy $\pi$ with initial state $x \in \mathcal{X}$,

$$g^\pi(x) := \lim_{T \to \infty} \mathbb{E}_{x_1=x}^\pi \left[ \frac{1}{T} \sum_{t=1}^T r(x_t, a_t) \right].$$

A policy $\pi^\varepsilon$ is $\varepsilon$-optimal if $g^{\pi^\varepsilon}(x) \geq g^\pi(x) - \varepsilon$ for all $\pi$ and $x \in \mathcal{X}$, and the optimal average reward is $g^*(x) = \sup_\pi g^\pi(x)$.

Parallel to reinforcement learning developments, quantum machine learning (Biamonte et al., 2017), a subfield of quantum computation (Nielsen & Chuang, 2010), has been consolidated as an area of study that can provide quantum speed-ups to several traditional machine learning problems (Lloyd et al., 2013; Rebentrost et al., 2014; Kerenidis & Prakash, 2017; Kerenidis et al., 2019; Doriguello et al., 2023), including reinforcement learning (Dong et al., 2008). It has been found that quantum algorithms are not only capable of achieving speedup in the time complexity of certain tasks, but also have the potential to be better "learners" than their classical counterparts in the online setting (Ganguly et al., 2023; Zhong et al., 2024).

In this work we study two fundamental problems of MDPs: (i) approximating optimal policies under a generative model and (ii) online learning of MDPs, both in the classical and quantum settings. For that, we propose a hybrid generative online-learning model in which classical and quantum computers are capable of offering an improved learning progress compared to several previous works (Auer et al., 2008; Bartlett & Tewari, 2009; Ortner & Ryabko, 2012; Lakshmanan et al., 2015; Fruit et al., 2018; Ganguly et al., 2023; Zhong et al., 2024).

## 2 COMPUTING OPTIMAL POLICIES UNDER A GENERATIVE MODEL

One of the main problems associated with MDPs is that of computing $\varepsilon$-optimal policies, for which several algorithms have been proposed (Bellman, 1966; Watkins & Dayan, 1992; Meyn, 1997; Kearns & Singh, 1998b; Sutton et al., 1999; Lagoudakis & Parr, 2003; Bertsekas, 2011; Azar et al., 2012; Puterman, 2014; Silver et al., 2014; Schulman et al., 2015; Jang et al., 2019). The main interest is thus on the performance of the computed policy (Sutton & Barto, 1998; Kearns & Singh, 1998a; 2002). Several input access models can be considered for computing optimal policies. In this paper, we consider the so-called *generative model* (Kearns & Singh, 1998b; Kearns et al., 2002; Kakade, 2003) where one has full knowledge of state and action spaces $\mathcal{X}, \mathcal{A}$ and of the reward function $r : \mathcal{X} \times \mathcal{A} \to [0, 1]$, but the transition probabilities $p(\cdot|x, a)$ can only be accessed through an oracle. In this scenario one is usually concerned with the sample complexity of employing such an oracle. In the classical setting, one has access to an oracle $\mathcal{C}_p$ that, on input $(x, a) \in \mathcal{X} \times \mathcal{A}$, returns $x' \in \mathcal{X}$ with probability $p(x'|x, a)$. We say we have *classical sampling access* to $p$ if we have access to $\mathcal{C}_p$. In the quantum setting, one has access to an oracle $\mathcal{O}_p$ called quantum accessible-environment (Wang et al., 2021; Wiedemann et al., 2023; Jerbi et al., 2023; Zhong et al., 2024) which is a unitary operator that acts as[1]

$$\mathcal{O}_p : |x\rangle |a\rangle |\bar{0}\rangle \to \int_{x' \in \mathcal{X}} \sqrt{p(\mathrm{d}x'|x, a)} |x\rangle |a\rangle |x'\rangle \quad \text{for all} \quad (x, a) \in \mathcal{X} \times \mathcal{A} \tag{1}$$

---

[1]For $\mathcal{X}$ finite, $\mathcal{O}_p : |x\rangle |a\rangle |\bar{0}\rangle \to \sum_{x' \in \mathcal{X}} \sqrt{p(x'|x, a)} |x\rangle |a\rangle |x'\rangle$ for all $(x, a) \in \mathcal{X} \times \mathcal{A}$.

(details on quantum computation can be found in Appendix A.1). We say we have *quantum sampling access to p* if we have access to $\mathcal{O}_p$ and its inverse (adjoint) $\mathcal{O}_p^\dagger$.

In this work, we assume $\mathcal{X}$ is compact (without loss of generality in an Euclidean space) and $\mathcal{A}$ is finite with size $A$. When $\mathcal{X}$ is finite we use $S$ for its size. For continuous state space $\mathcal{X}$, we will assume that $p$ and $r$ are Hölder continuous with parameters $L, \alpha \geq 0$, i.e., $|r(x, a) - r(x', a)| \leq L\|x - x'\|_2^\alpha$ and $\|p(\cdot|x, a) - p(\cdot|x', a)\|_{\text{tvd}} \leq L\|x - x'\|_2^\alpha$ for all $(x, x', a) \in \mathcal{X} \times \mathcal{X} \times \mathcal{A}$.

## 2.1 Finite-horizon MDPs

There is a long list of works that studied the classical query complexity of obtaining optimal policies (Kearns & Singh, 1998a; Kearns et al., 2002; Gheshlaghi Azar et al., 2013; Wang, 2017b; Sidford et al., 2018b;a; Li et al., 2020). For finite-horizon and infinite-horizon *discounted* MDPs with *finite* state space, Sidford et al. (2018a) and Li et al. (2020) obtained sample-optimal algorithms that output an $\varepsilon$-optimal policy using $\widetilde{O}\big(\frac{H^3SA}{\varepsilon^2}\big)$ queries to $\mathcal{C}_p$, which matches the sample lower bounds from Gheshlaghi Azar et al. (2013); Sidford et al. (2018a) up to polylogarithmic factors. In the quantum setting, on the other hand, we are only aware of a few works related to the problem of computing optimal policies (Wiedemann et al., 2023; Wang et al., 2021; Jerbi et al., 2023; Cherrat et al., 2023), all for infinite-horizon discounted MDPs. More closely related to our paper is the work of Wang et al. (2021). The authors quantised the standard value iteration (Puterman, 2014) and the modern value iteration algorithms from Sidford et al. (2018a); Li et al. (2020) using quantum techniques like quantum minimum finding (Dürr & Høyer, 1996) and quantum mean estimation (Brassard et al., 2002; Montanaro, 2015). Their final result is a quantum algorithm that outputs an $\varepsilon$-optimal policy with quantum query complexity $\widetilde{O}\big(\min\big\{\frac{\Gamma^{1.5}SA}{\varepsilon}, \frac{\Gamma^3S\sqrt{A}}{\varepsilon}\big\}\big)$, improving the classical query complexity $\widetilde{O}\big(\frac{\Gamma^3SA}{\varepsilon^2}\big)$ from Sidford et al. (2018a); Li et al. (2020) ($\Gamma$ is the "effective horizon" in infinite-horizon discounted MDPs).

The problem of computing optimal policies under a quantum generative model for finite-horizon MDPs is, as far as we are aware, still unexplored.[2] As one of our main results, we propose quantum algorithms for computing optimal policies in this setting for both compact and finite state spaces. In the following, $\mathcal{S}_n \subset \mathcal{X}$ is a $\frac{1}{n}$-net for $\mathcal{X}$ if $\min_{s \in \mathcal{S}_n} \|x - s\|_2 \leq \frac{1}{n} \ \forall x \in \mathcal{X}$.

**Result 1** (Finite-horizon). *Let $M = \langle \mathcal{X}, \mathcal{A}, H, p, r \rangle$ be a finite-horizon MDP and $\mathcal{S}_n$ a $\frac{1}{n}$-net for the compact state space $\mathcal{X}$. Assume that $p$ and $r$ are Hölder continuous with parameters $L, \alpha \geq 0$. There are classical and quantum algorithms that output an $(\varepsilon + cH^2Ln^{-\alpha})$-optimal policy with high probability, where $c > 0$ is some constant. The classical algorithm uses $\widetilde{O}\big(\frac{H^3|\mathcal{S}_n|A}{\varepsilon^2}\big)$ classical queries and the quantum algorithm uses $\widetilde{O}\big(\min\big\{\frac{H^2|\mathcal{S}_n|A}{\varepsilon}, \frac{H^3|\mathcal{S}_n|\sqrt{A}}{\varepsilon}\big\}\big)$ quantum queries, where $\widetilde{O}(\cdot)$ hides $\mathrm{poly}\log$ factors. If $\mathcal{X}$ is finite with size $S$, then $L = 0$ and $|\mathcal{S}_n| = S$.*

Our quantum algorithms are quantised versions of the standard value iteration (Puterman, 2014) and the modern value iteration algorithms of Sidford et al. (2018a); Li et al. (2020) for finite-horizon MDPs, similar in spirit to what Wang et al. (2021) did on infinite-horizon discounted MDPs. In order to achieve the stated query complexity, we employ standard quantum minimum finding (Dürr & Høyer, 1996), the quantum mean estimation subroutine from Kothari & O'Donnell (2023), and its multivariate version due to Cornelissen et al. (2022) and Tang (2025). The proof is in Appendix C. We note that Luo et al. (2025) very recently proved the quantum query lower bound $\Omega\big(\frac{H^{1.5}S\sqrt{A}}{\varepsilon}\big)$.

## 2.2 Infinite-horizon MDPs

Regarding infinite-horizon undiscounted MDPs, the list of classical algorithms for computing optimal policies under a generative model is more recent (Wang, 2017a; Jin & Sidford, 2020; 2021; Wang et al., 2022; Zhang & Xie, 2023; Li et al., 2024), one of the best complexities being $\widetilde{O}\big(\frac{\Lambda^2SA}{\varepsilon^2}\big)$ due to Zhang & Xie (2023), where $\mathrm{sp}(h^*) := \max_{x \in \mathcal{X}} h^*(x) - \min_{x \in \mathcal{X}} h^*(x) \leq \Lambda$ is an upper-bound on the span of the optimal bias $h^*(x)$, which is a mild assumption used in several RL works (Bartlett & Tewari, 2009; Ortner & Ryabko, 2012; Lakshmanan et al., 2015; Fruit et al.,

---

[2]Shortly before the submission, we became aware of Luo et al. (2025), who proved the quantum query upper bound $\widetilde{O}\big(\min\big\{\frac{H^{2.5}SA}{\varepsilon}, \frac{H^3S\sqrt{A}}{\varepsilon}\big\}\big)$ for finite-horizon finite-state-space MDPs.

2018). Here $h^\pi(x)$ is the *bias function* of a policy $\pi$, which informally measures the difference in accumulated rewards when starting at $x \in \mathcal{X}$ compared to starting from the stationary distribution (see Appendix A.3 for a formal definition). In order to present a quantum algorithm in this setting, we opted to adapt a simple classical value iteration (Puterman, 2014) to the generative model with oracle $\mathcal{C}_p$ and assumed that the underlying infinite-horizon MDP has some contractive properties (to be more precise, the associated optimal Bellman operator is a 1-stage $\nu$-span contraction for $\nu \in [0, 1)$, Appendix D). Our results apply to a broad class of infinite-horizon MDPs called *weakly communicating* (see Appendix A.3 or (Puterman, 2014, Section 8.3.1)). The proof is in Appendix D.

**Result 2** (Infinite-horizon). *Let $M = \langle \mathcal{X}, \mathcal{A}, p, r \rangle$ be an infinite-horizon (undiscounted) average-reward weakly communicating MDP with $\mathrm{sp}(h^*) \leq \Lambda$ and $\mathcal{S}_n$ a $\frac{1}{n}$-net for the compact state space $\mathcal{X}$. Assume $p$ and $r$ are Hölder continuous with parameters $L, \alpha \geq 0$ and that $M$ has contractive properties. There are classical and quantum algorithms that output an $(\varepsilon + c(1 + \Lambda)Ln^{-\alpha})$-optimal stationary policy with high probability, where $c > 0$ is some constant. The classical algorithm uses $\widetilde{O}\left(\frac{\Lambda^2 |\mathcal{S}_n| A}{\varepsilon^2}\right)$ classical queries and the quantum algorithm uses $\widetilde{O}\left(\frac{\Lambda |\mathcal{S}_n| \sqrt{A}}{\varepsilon}\right)$ quantum queries, where $\widetilde{O}(\cdot)$ hides* poly log *factors. If $\mathcal{X}$ is finite with size $S$, then $L = 0$ and $|\mathcal{S}_n| = S$.*

## 3 Online learning of MDPs

As another main contribution, we propose improved classical and quantum algorithms for learning unknown finite-horizon and infinite-horizon MDPs in an online fashion. Here one is interested in the performance of the learning algorithm during the learning process, an area of study that finds applications in several topics (Crammer et al., 2003; Ying & Zhou, 2006; Liang et al., 2006; Tekin & Liu, 2010; Li & Hoi, 2014; Ouyang et al., 2017; Aaronson et al., 2018; Lim & Rebentrost, 2024). Interacting with an unknown MDP can be naturally framed as an online learning problem: at each time step $t \in \mathbb{N}$, the environment is at some state $x_t \in \mathcal{X}$ and the agent must choose an action $a_t \sim \pi_t(\cdot|x_t)$ according to some policy $\pi = (\pi_t)_t$ in order to receive as large a reward $r(x_t, a_t)$ as possible, after which the environment transitions to a new state $x_{t+1} \sim p(\cdot|x_t, a_t)$. However, since the agent does not know the underlying transition probabilities $p(\cdot|x, a)$, an exploration vs. exploitation trade-off arises: should the agent explore poorly understood states and actions in order to improve its understanding of the MDP and improve its future performance via better policies, or exploit its current knowledge to optimise short-term rewards (Kearns & Singh, 2002).

The task of obtaining large rewards is usually reframed as minimising some measure of how far the agent is from being optimal (Valiant, 1984; Littlestone, 1988; Li et al., 2008; Auer et al., 2008). One of the most famous measures is that of *regret*, which is the difference between the agent's (expected) rewards compared to that of the optimal policy. For finite-horizon MDPs, the regret over $K$ episodes (or $T = KH$ time steps) is defined as

$$\mathrm{Regret}_H(T) := \sum_{k=1}^{K} \left( V_1^*(x_1^{(k)}) - V_1^{\pi^{(k)}}(x_1^{(k)}) \right),$$

where $\pi^{(k)}$ is the policy used for the $k$-th episode and $x_1^{(k)}$ is the environment's initial state in the $k$-th episode. For infinite-horizon MDPs, the (in-path) regret over $T$ time steps is defined as[3]

$$\mathrm{Regret}_\infty^{\mathrm{path}}(T) := Tg^* - \sum_{t=1}^{T} r(x_t, a_t).$$

In the classical setting, there are several proposed algorithms that achieve regret bounds sublinear in the number of time steps $o(T)$ for finite-horizon finite-state-space MDPs (Auer et al., 2008; Osband & Roy, 2014; Azar et al., 2017; Jin et al., 2018; Zanette & Brunskill, 2019; Efroni et al., 2019; Jin et al., 2020; Yang & Wang, 2020). To our knowledge, the best regret bounds are due to Azar et al. (2017); Zanette & Brunskill (2019), which match the lower bound $\Omega(\sqrt{HSAT})$ from Auer et al. (2008) for a certain range of parameters. For infinite-horizon MDPs, the landscape is richer. The algorithm of Auer et al. (2008) ranked among the main ones to be first proposed, although it applied to a smaller class of MDPs called *communicating* (Puterman, 2014). Bartlett & Tewari

---

[3]Works like Bartlett & Tewari (2009) employ instead the regret $Tg^* - \mathbb{E}[\sum_{t=1}^{T} r(x_t, a_t)]$.

(2009) achieved the regret bound $\widetilde{O}(\Lambda\sqrt{S^2AT})$ for the broader class of weakly communicating MDPs, where $\mathrm{sp}(h^*) \leq \Lambda$. Later, Ortner & Ryabko (2012) and Lakshmanan et al. (2015) adapted the algorithm from Auer et al. (2008) for the case when the state space is $\mathcal{X} = [0,1]^D$ under the assumption that $r$ and $p$ are Hölder continuous. The algorithms of Bartlett & Tewari (2009); Ortner & Ryabko (2012); Lakshmanan et al. (2015) are, however, time inefficient, which was later fixed by Fruit et al. (2018), whose time-efficient algorithm still maintains the regret bound $\widetilde{O}(\Lambda\sqrt{S^2AT})$ for finite state spaces. We summarise several results from the literature in Table 1.

In the quantum setting, the number of works on online learning of MDPs is much smaller. We are actually only aware of the works by Zhong et al. (2024) and Ganguly et al. (2023), both for finite-horizon finite-state-space MDPs (there are further works on multi-armed bandits (Lumbreras et al., 2022; Dai et al., 2023; Wan et al., 2023; Su et al., 2025)). By leveraging a modified environment-agent interaction model (described below), both works obtained $\mathrm{poly}(S, A, H, \log T)$ regret, exponentially better in the number of steps $T$. Their precise regret is summarised in Table 1.

### 3.1 OUR ONLINE LEARNING MODEL

Even though the interaction model between agent and environment is straightforward in the classical setting, the same is not true in the quantum setting. Ideally, we would like to employ the quantum oracle $\mathcal{O}_p$ from Eq. (1) to explore the MDP in superposition, which inevitably leads to some apparent conundrums. For once, a repeated interaction between agent and environment through quantum oracles will lead to a superposition of different possible rewards. It is then not clear what the final regret is, especially if the agent performs non-trivial intermediary quantum gates and measurements. More critically, however, we would like to employ quantum subroutines (Brassard et al., 2002; Montanaro, 2015; Cornelissen et al., 2022; Kothari & O'Donnell, 2023) that make use of the inverse of $\mathcal{O}_p$. There is absolutely no equivalent inverse operation in the standard classical model and, moreover, one would imagine that "inverting" quantum operations could potentially "undo" the accumulated regret.

We solve these issues by virtually separating the exploration phase from the policy learning phase. In the standard classical model, these are perform simultaneously: while interacting with the environment, the agent keeps track of all state-action pairs $(x_t, a_t)$ observed so far to come up with an estimation $\widetilde{p}$ of the true transition probability, which is then used to obtain an approximate optimal policy. In our model, however, such interaction is split into *two* types of phases: *classical exploration* phases and *classical/quantum generative* phases.

1. **Exploration phase:** an exploration phase corresponds exactly to the standard classical agent-environment interaction, during which the agent accumulates regret. More specifically, the agent chooses action $a_t$ at state $x_t$, obtains reward $r(x_t, a_t)$, and observes the new state $x_{t+1}$. The interaction is completely classical and lasts as long as the agent desires.

2. **Generative phase:** during the generative phase, the agent is *free* to interact with the environment using quantum accessible-environments, more specifically $\mathcal{O}_p$, *without* accumulating regret. This means that the agent is free to prepare any quantum state and have the environment apply $\mathcal{O}_p$ (or its inverse) onto such quantum state, plus any quantum gate from a universal gate set. The transition from exploration to generative phase and vice versa can be done at any moment. During the generative phase, the agent can *only use the oracle $\mathcal{O}_p$ and its inverse at most $O(\tau)$ times,* where $\tau$ is length of the previous *exploration* phase.

By separating the accumulation of regret from the policy learning phase as above, all the problems previously highlighted are avoided. The result is an alternation between exploration and generative phases. We name the free interaction phase "generative" because the agent is free to interact with the environment in a generative fashion and, as described below, we employ the algorithms from Results 1 and 2 to compute an $\varepsilon$-optimal policy during this phase. The restriction on the number of applications of $\mathcal{O}_p$ during the generative phase is vital to guarantee a meaningful online-learning model, otherwise the agent could obtain a policy as close to optimal as needed and thus incur a regret as small as desired. In order to freely use the probability oracle $\mathcal{O}_p$, the agent must first pay a price in committing to actions and incurring into regret. Finally, we also consider a classical version of the above model wherein, during generative phases, the agent has access to a sampling oracle $\mathcal{C}_p$ that returns $x' \sim p(\cdot|x,a)$ on input $(x,a)$ (see Appendix B for a comparison between our model and the model of Zhong et al. (2024)).

## 3.2 OUR REGRET BOUNDS

Using our exploration-generative model described above, we propose new classical and quantum RL algorithms that achieve better regret bounds compared to prior works. Our algorithms for finite and infinite-horizon MDPs are similar and are explained in a high level in Algorithm 1. They progress in episodes, each made up of a generative phase and an exploration phase. During the $k$-th generative phase, an approximately optimal policy $\pi^{(k)}$ is computed using the algorithms from Results 1 and 2. Such policy is then employed in the exploration phase to accumulate regret for a period of time long enough to allow the computation of a better policy in the next generative phase. If the $(k-1)$-th exploration phase lasts $\tau_k$ time steps, then in the $k$-th generative phase one can make $O(\tau_k)$ calls to the classical or quantum oracles $\mathcal{C}_p$ or $\mathcal{O}_p$, which, according to Results 1 and 2, yields an $\varepsilon_k$-optimal policy where $\varepsilon_k = \widetilde{O}(\tau_k^{-1/2})$ in the classical setting and $\varepsilon_k = \widetilde{O}(\tau_k^{-1})$ in the quantum setting (the dependence on other parameters in shown in Algorithm 1). In order to set the length of the exploration phases, which determine the number of oracle calls in the generative phases, we follow a doubling trick: the length of the exploration phase is increased by a constant factor after each episode. For finite-horizon MDPs, the increase factor is $2H$, while for infinite-horizon MDPs it is 2.

Although very similar, the main difference between the finite and infinite-horizon versions of Algorithm 1 is that each exploration phase in the finite-horizon setting must be a multiple of $H$, since the agent explores the MDP in sequences of $H$ steps. This is implicit in the definition of $\mathrm{Regret}_H(T)$, where $V_1^*(x_1^{(k)}) - V_1^{\pi^{(k)}}(x_1^{(k)})$ is a difference of rewards over $H$ steps. As a consequence, the agent computes a policy $\pi^{(k)} = (\pi_1^{(k)}, \pi_2^{(k)}, \ldots, \pi_H^{(k)})$ for $H$ time steps during the generative phase (hence why we write $\pi_{t \pmod H}^{(k)}$ in Algorithm 1 to compactly represent the repeated application over all decision rules in $\pi^{(k)}$). In the infinite-horizon setting, the agent computes a stationary policy $\pi_k^\infty = (\pi_k, \pi_k, \ldots)$ during the generative phase, meaning that the same decision rule $\pi_k$ is employed throughout the next exploration phase.

We now state the regret bounds of Algorithm 1. Their proof (and a more detailed description of Algorithm 1) is left to Appendices E and F.

**Result 3** (Finite-horizon regret). *Let $M = \langle \mathcal{X}, \mathcal{A}, H, p, r \rangle$ be a finite-horizon MDP. If $\mathcal{X}$ is finite with size $S$, there are classical and quantum algorithms with regret $\mathrm{Regret}_H(T)$ upper-bounded after $T$ steps, with high probability, by*

$$\textit{Classical: } \widetilde{O}\big(\sqrt{HSAT\log(HSA)}\big),$$

$$\textit{Quantum: } \widetilde{O}\big(\min\{HSA, H^2 S\sqrt{A}\log(HS)\}\log(T/H)\log(HSA)\big),$$

*where $\widetilde{O}(\cdot)$ hides $\mathrm{poly}\log\log$ terms. If $\mathcal{X} = [0,1]^D$, assume that $p$ and $r$ are Hölder continuous with parameters $L, \alpha \geq 0$, $L$ constant. There are classical and quantum algorithms with regret $\mathrm{Regret}_H(T)$ upper-bounded after $T$ steps, with high probability, by*

$$\textit{Classical: } \widetilde{O}\big(T^{\frac{D+\alpha}{D+2\alpha}}(H + \sqrt{HA\log(HAT)})\big),$$

$$\textit{Quantum: } \widetilde{O}\big(\min\{HA, H^2\sqrt{A}\log(HT)\}T^{\frac{D}{D+\alpha}}\log(T/H)\log(HAT)\big).$$

**Result 4** (Infinite-horizon in-path regret). *Let $M = \langle \mathcal{X}, \mathcal{A}, p, r \rangle$ be an infinite-horizon (undiscounted) average-reward weakly communicating MDP with $\mathrm{sp}(h^*) \leq \Lambda$. If $\mathcal{X}$ is finite with size $S$, there are classical and quantum algorithms with in-path regret $\mathrm{Regret}_\infty^{\mathrm{path}}(T)$ upper-bounded after $T$ steps, with high probability, by*

$$\textit{Classical: } \widetilde{O}\big(\Lambda\sqrt{SAT\log T}\log(SAT)\big),$$

$$\textit{Quantum: } \widetilde{O}\big(\Lambda\sqrt{T\log T} + \Lambda S\sqrt{A}\log^2 T\log(SAT)\log(ST)\big),$$

*where $\widetilde{O}(\cdot)$ hides $\mathrm{poly}\log\log$ terms. If $\mathcal{X} = [0,1]^D$, assume that $p$ and $r$ are Hölder continuous with parameters $L, \alpha \geq 0$, $L$ constant. There are classical and quantum algorithms with in-path regret $\mathrm{Regret}_\infty^{\mathrm{path}}(T)$ upper-bounded after $T$ steps, with high probability, by*

$$\textit{Classical: } \widetilde{O}\big(\Lambda T^{\frac{D+\alpha}{D+2\alpha}}\sqrt{A\log T}\log(AT)\big),$$

$$\textit{Quantum: } \widetilde{O}\big(\Lambda\sqrt{T\log T} + \Lambda\sqrt{A}T^{\frac{D}{D+\alpha}}\log^3 T\log(AT)\big).$$

---

**Algorithm 1** Online-learning algorithms for finite and infinite-horizon MDPs

---

**Input:** Compact state space $\mathcal{X}$ with $\frac{1}{n}$-net $\mathcal{S}_n$, finite action space $\mathcal{A}$, horizon $H$ (finite-horizon), upper bound $\Lambda$ on optimal bias span (infinite-horizon), rewards $r : \mathcal{X} \times \mathcal{A} \to [0,1]$, $L, \alpha \geq 0$.

1: $t \leftarrow 1$ and $\tau_1 \leftarrow 1$
2: **for** episodes $k = 1, 2, \dots$ **do**
    **Generative phase**
3:    **Finite-horizon:** Using Result 1, choose policy $\pi^{(k)}$ such that, with high probability,

$$\|V_1^{\pi^{(k)}} - V_1^*\|_\infty \leq \begin{cases} \widetilde{O}\Big(\sqrt{\frac{H^2|\mathcal{S}_n|A}{\tau_k}} + H^2 Ln^{-\alpha}\Big) & \text{(classical)}, \\ \widetilde{O}\Big(\min\Big\{\frac{H|\mathcal{S}_n|A}{\tau_k}, \frac{H^2|\mathcal{S}_n|\sqrt{A}}{\tau_k}\Big\} + H^2 Ln^{-\alpha}\Big) & \text{(quantum)} \end{cases}$$

3:    **Infinite-horizon:** Using Result 2, choose decision rule $\pi_k$ such that, with high probability,

$$\|g^{\pi_k^\infty} - g^* e\|_\infty \leq \begin{cases} \widetilde{O}\Big(\sqrt{\frac{\Lambda^2|\mathcal{S}_n|A}{\tau_k}} + (1 + \Lambda)Ln^{-\alpha}\Big) & \text{(classical)}, \\ \widetilde{O}\Big(\frac{\Lambda|\mathcal{S}_n|\sqrt{A}}{\tau_k} + (1 + \Lambda)Ln^{-\alpha}\Big) & \text{(quantum)} \end{cases}$$

    **Exploration phase**
4:    $\tau_{k+1} \leftarrow t$ and observe random initial state $x_t$
5:    **while** $t < 2H\tau_{k+1}$ (finite-horizon) / $t < 2\tau_{k+1}$ (infinite-horizon) **do**
6:        Choose action $a_t \sim \pi_{t \pmod H}^{(k)}(x_t)$ (finite-horizon) or $a_t \sim \pi_k(x_t)$ (infinite-horizon)
7:        Obtain reward $r_t \leftarrow r(x_t, a_t)$, observe next state $x_{t+1} \sim p(\cdot|x_t, a_t)$, and $t \leftarrow t + 1$
8:    **end while**
9: **end for**

---

Several regret bounds from Results 3 and 4 considerably improve upon prior works (Auer et al., 2008; Azar et al., 2017; Zanette & Brunskill, 2019; Ortner & Ryabko, 2012; Lakshmanan et al., 2015; Zhong et al., 2024; Ganguly et al., 2023) (see Table 1 for a clear comparison). For finite-horizon MDPs, our classical bound $\widetilde{O}(\sqrt{HSAT})$ matches the ones from Azar et al. (2017); Zanette & Brunskill (2019) when $T \geq H^3 S^3 A$ and $SA \geq H$, and avoids the extra terms $\widetilde{O}(H^2 S^2 A + H\sqrt{T})$ outside this parameter range. The quantum bound $\widetilde{O}(\min\{HSA, H^2 S\sqrt{A}\})$ is quadratically better in $S$, $A$, $H$ compared to Zhong et al. (2024); Ganguly et al. (2023) and maintains the logarithmic dependence on $T$. Regarding infinite-horizon MDPs, our classical regret $\widetilde{O}(\Lambda\sqrt{SAT})$ improves upon the bound $\widetilde{O}(\Lambda\sqrt{S^2 AT})$ of Bartlett & Tewari (2009); Fruit et al. (2018), while for compact state spaces, the classical regret $\widetilde{O}(\Lambda T^{\frac{D+\alpha}{D+2\alpha}}\sqrt{A})$ is superior to the bound $\widetilde{O}(\Lambda T^{\frac{1+(D+1)\alpha}{1+(D+2)\alpha}}\sqrt{A})$ of Lakshmanan et al. (2015).[4] Our quantum regret bounds for infinite-horizon MDPs are completely novel and clearly improve upon their classical counterparts. One should keep in mind, however, that these comparisons are made between different RL models.

The secret of the improved performance of Algorithm 1 is that it avoids standard RL principles like "optimism in the face of uncertainty" (Brafman & Tennenholtz, 2002) common in several RL algorithms (Auer et al., 2008; Bartlett & Tewari, 2009; Ortner & Ryabko, 2012; Lakshmanan et al., 2015; Azar et al., 2017; Jin et al., 2018; Zanette & Brunskill, 2019; Efroni et al., 2019; Zhong et al., 2024; Ganguly et al., 2023). In these algorithms, each state-action pair is given some "optimism" such that its imagined value is as high as statistically possible and the agent chooses a policy according to such optimistic values. In other words, the estimated transition probability $\widetilde{p}$ together with its uncertainty define a set of plausible MDPs $\mathcal{M}$ that contains the true MDP with high probability. The agent then chooses an optimal policy with respect to *all* MDPs in $\mathcal{M}$ (optimism). Our algorithms, on the other hand, avoid this principle altogether since, during the generative phases, we have access to the true MDP via oracles $\mathcal{C}_p$ or $\mathcal{O}_p$. Ultimately, an optimal policy is what is needed to maintain a small regret. Therefore, there is no reason to approximate $p$ and the agent instead directly computes an approximate optimal policy using Results 1 and 2. The agent can eventually learn $p$ later on when a policy close to optimal has already been obtained.

---

[4]For $D > 1$, (Lakshmanan et al., 2015, Eq. (29)) ignores a term $\widetilde{O}(\Lambda A T^{\frac{D}{1+(D+2)\alpha}})$ which dominates the complexity for small $\alpha$, so our regret bound is better for all values of $\alpha$ and $D$.

Table 1: Summary of several known upper bounds for the regret $\mathrm{Regret}_H(T)$ of *finite-horizon* MDPs $\langle \mathcal{X}, \mathcal{A}, H, p, r \rangle$ and for the *in-path* regret $\mathrm{Regret}_\infty^{\mathrm{path}}(T)$ of *infinite-horizon* average-reward MDPs $\langle \mathcal{X}, \mathcal{A}, p, r \rangle$ with state space $\mathcal{X}$, action space $\mathcal{A}$ with size $A$, number of time steps $T$, horizon $H$ (finite-horizon), and $\mathrm{sp}(h^*) \leq \Lambda$ (infinite-horizon). The state space $\mathcal{X}$ can be finite with size $S$ or $\mathcal{X} = [0,1]^D$, in which case $p$ and $r$ are assumed to be Hölder continuous with exponent $\alpha \geq 0$. Here $\Delta \geq \Lambda$ and $t_{\mathrm{mix}}$ are the MDP's diameter and mixing time, respectively. In the last rows, we show our bounds for the *expected* regret $\mathrm{Regret}_\infty^{\mathbb{E}}(T)$. All bounds are up to $\mathrm{poly}\log$ factors in $A$, $T$, $S$ (for $\mathcal{X}$ finite), and $H$ (finite-horizon) or $\Lambda$ (infinite-horizon). Refs. marked by † have no efficient implementation for infinite-horizon MDPs. The reader should keep in mind that not all works assume the same RL model: there are model-based (Auer et al., 2008) and model-free (Jin et al., 2018) approaches, while Zhong et al. (2024); Ganguly et al. (2023) and our work assume a hybrid generative model. Some works focus on optimising other parameters besides regret.

| Finite state space $\mathcal{X}$ | Finite-horizon MDPs ($\mathrm{Regret}_H(T)$) | | Infinite-horizon MDPs ($\mathrm{Regret}_\infty^{\mathrm{path}}(T)$) | |
|---|---|---|---|---|
| | Regret (classical) | Regret (quantum) | Regret (classical) | Regret (quantum) |
| Auer et al. (2008) | $\sqrt{H^2S^2AT}+HSA$ | - | $\sqrt{\Delta^2S^2AT}+\Delta SA$ | - |
| Bartlett & Tewari (2009)† | $\sqrt{H^2S^2AT}$ | - | $\sqrt{\Lambda^2S^2AT}$ | - |
| Ouyang et al. (2017) | - | - | $\sqrt{\Lambda^2S^2AT}$ | - |
| Azar et al. (2017) | $\sqrt{HSAT}+H^2S^2A+\sqrt{H^2T}$ | - | - | - |
| Fruit et al. (2018) | - | - | $\sqrt{\Lambda^2S^2AT}+\Lambda S^2A$ | - |
| Jin et al. (2018) | $\sqrt{H^4SAT}$ $\sqrt{H^3SAT}+\sqrt{H^9S^3A^3}$ | - - | - - | - - |
| Zanette & Brunskill (2019) | $\sqrt{HSAT}+H^2S^{\frac{3}{2}}A(\sqrt{S}+\sqrt{H})$ | - | - | - |
| Efroni et al. (2019) | $\sqrt{HSAT}+H^2S^{\frac{3}{2}}A(\sqrt{S}+\sqrt{H})$ | - | - | - |
| Bai et al. (2019) | $\sqrt{H^3SAT}+\sqrt{H^9S^3A^3}$ | - | - | - |
| Zhang & Ji (2019)† | - | - | $\sqrt{\Lambda SAT}+\Lambda(S^{10}AT)^{\frac{1}{4}}$ | - |
| Zhang et al. (2020) | $\sqrt{H^2SAT}+H^8S^2A^{\frac{3}{2}}T^{\frac{1}{4}}$ | - | - | - |
| Fruit et al. (2020) | - | - | $\sqrt{\Delta S^2AT}$ | - |
| Ortner (2020)† | - | - | $\sqrt{t_{\mathrm{mix}}SAT}$ | - |
| Wei et al. (2020) | - | - | $\Lambda(SAT^2)^{\frac{1}{3}}$ | - |
| Menard et al. (2021) | $\sqrt{H^2SAT}+H^4SA$ | - | - | - |
| Li et al. (2021) | $\sqrt{H^2SAT}+H^6SA$ | - | - | - |
| Wei et al. (2021) | - - | - - | $\sqrt{\Lambda^2S^3A^3T}$ $\sqrt{\Lambda}(SAT)^{\frac{3}{4}}+(\Lambda SAT)^{\frac{2}{3}}$ | - - |
| Zhang & Xie (2023) | - | - | $\sqrt{\Lambda^2S^{10}A^4T}$ | - |
| Zhong et al. (2024) | - | $H^3S^2A$ | - | - |
| Ganguly et al. (2023) | - | $H^2S^2A$ | - | - |
| Agrawal & Agrawal (2025) | $\sqrt{H^{12}S^2AT}+H^9S^2A$ | - | - | - |
| This work | $\sqrt{HSAT}$ | $\min\{HSA, H^2S\sqrt{A}\}$ | $\sqrt{\Lambda^2SAT}$ | $\sqrt{\Lambda^2T}+\sqrt{\Lambda^2S^2A}$ |
| $\mathrm{Regret}_\infty^{\mathbb{E}}(T)$ | - | - | $\sqrt{\Lambda^2SAT}$ | $\sqrt{\Lambda^2S^2A}$ |

| State space $\mathcal{X}=[0,1]^D$ | Finite-horizon MDPs ($\mathrm{Regret}_H(T)$) | | Infinite-horizon MDPs ($\mathrm{Regret}_\infty^{\mathrm{path}}(T)$) | |
|---|---|---|---|---|
| | Regret (classical) | Regret (quantum) | Regret (classical) | Regret (quantum) |
| Ortner & Ryabko (2012)† | $\sqrt{H^2A}\,T^{\frac{2D+\alpha}{2D+2\alpha}}$ | - | $\sqrt{\Lambda^2A}\,T^{\frac{2D+\alpha}{2D+2\alpha}}$ | - |
| Lakshmanan et al. (2015)† | $\sqrt{H^2A}\,T^{\frac{1+(D+1)\alpha}{1+(D+2)\alpha}}$ | - | $\sqrt{\Lambda^2A}\,T^{\frac{1+(D+1)\alpha}{1+(D+2)\alpha}}$ | - |
| This work | $(H+\sqrt{HA})T^{\frac{D+\alpha}{D+2\alpha}}$ | $\min\{HA, H^2\sqrt{A}\}T^{\frac{D}{D+\alpha}}$ | $\sqrt{\Lambda^2A}\,T^{\frac{D+\alpha}{D+2\alpha}}$ | $\sqrt{\Lambda^2T}+\sqrt{\Lambda^2A}\,T^{\frac{D}{D+\alpha}}$ |
| $\mathrm{Regret}_\infty^{\mathbb{E}}(T)$ | - | - | $\sqrt{\Lambda^2A}\,T^{\frac{D+\alpha}{D+2\alpha}}$ | $\sqrt{\Lambda^2A}\,T^{\frac{D}{D+\alpha}}$ |

## 3.3 A NEW MEASURE OF REGRET

A striking feature of Results 3 and 4 is that, just like Zhong et al. (2024); Ganguly et al. (2023), the quantum regret bounds for finite-horizon MDPs are exponentially better in the number of steps $T$, but the same is not true for infinite-horizon MDPs, where the term $\widetilde{O}(\Lambda\sqrt{T})$ hinders such an exponential advantage. A closer look at the proof reveals that $\widetilde{O}(\Lambda\sqrt{T})$ comes from a (Azuma-

Hoeffding) concentration inequality, which hints as being an artifact of a deviation from some mean. A comparison between finite and infinite-horizon regrets $\text{Regret}_H(T)$ and $\text{Regret}_\infty^{\text{path}}(T)$ sheds further light into this issue. $\text{Regret}_H(T)$ is defined as the difference between two *expected* quantities, more precisely, the difference of expected sum of rewards of the optimal policy and the actual policy. On the other hand, $\text{Regret}_\infty^{\text{path}}(T)$ takes into account the path of observed state-action pairs and not the difference of some quantity averaged with respect to the optimal policy and the actual policy, e.g., the average reward $g^\pi(x)$. For this reason, we introduce a novel measure of regret for infinite-horizon MDPs, $\text{Regret}_\infty^{\mathbb{E}}(T)$, which we call *expected* regret, defined as

$$\text{Regret}_\infty^{\mathbb{E}}(T) := \sum_{t=1}^T \left( g^* - \min_{x \in \mathcal{X}} g^{\pi_t^\infty}(x) \right),$$

where $\pi_t^\infty = (\pi_t, \pi_t, \dots)$ is the stationary policy obtained from the current decision rule $\pi_t$. In other words, $g^{\pi_t^\infty}(x)$ measures the average reward per step if the agent were to use his or her current decision rule $\pi_t$ for the rest of the interaction. Obviously, the agent can pick a different decision rule $\pi_{t+1}$ at the next time step and thus incur a different average regret $g^* - \min_{x \in \mathcal{X}} g^{\pi_{t+1}^\infty}(x)$ at that step. The choice for minimising over $x \in \mathcal{X}$ takes into account a worst-case situation and is arbitrary in principle. One could have picked the current environment state $x_t$ and defined the average regret per step as $g^* - g^{\pi_t^\infty}(x_t)$ instead.[5]

We show that, although our classical algorithm has the same in-path and expected regrets, our quantum algorithm can achieve an expected regret $\text{Regret}_\infty^{\mathbb{E}}(T)$ that is exponentially better in $T$ compared to their classical counterparts (see also Table 1), which highlights the importance of a proper measure of regret for quantum RL algorithms that can isolate the extension of a quantum advantage from random fluctuations.

**Result 5** (Infinite-horizon expected regret). *Let $M = \langle \mathcal{X}, \mathcal{A}, p, r \rangle$ be an infinite-horizon average-reward weakly communicating MDP with $\text{sp}(h^*) \leq \Lambda$. If $\mathcal{X}$ is finite with size $S$, there is a quantum algorithm with expected regret $\text{Regret}_\infty^{\mathbb{E}}(T)$ upper-bounded after $T$ steps, with high probability, by*

$$\widetilde{O}\big(\Lambda S \sqrt{A} \log^2 T \log(SAT) \log(ST)\big),$$

*where $\widetilde{O}(\cdot)$ hides* poly log log *terms. If $\mathcal{X} = [0, 1]^D$, assume that $p$ and $r$ are Hölder continuous with parameters $L, \alpha \geq 0$, $L$ constant. There is a quantum algorithm with expected regret $\text{Regret}_\infty^{\mathbb{E}}(T)$ upper-bounded after $T$ steps, with high probability, by*

$$\widetilde{O}\big(\Lambda \sqrt{A} T^{\frac{D}{D+\alpha}} \log^3 T \log(AT)\big).$$

## 4 CONCLUSIONS AND FUTURE WORK

We proposed novel quantum algorithms for computing $\varepsilon$-optimal policies for finite-horizon and infinite-horizon MDPs, and generalised known classical algorithms to the setting of compact state spaces assuming that $r$ and $p$ are Hölder continuous. We then formalised a RL model, which we called online exploration-generative model, wherein the agent can freely interact with the environment from time to time via specific oracles. By combining our optimal policy algorithms with this online learning model, we then proposed novel classical and quantum online algorithm for learning finite and infinite-horizon MDPs which achieve better regret bounds compared to prior works. This includes the proposal of a novel measure of regret with respect to which our quantum RL algorithm has exponentially better regret in $T$ compared to its classical counterpart. Our results show the impact of having a bit of "freedom" between agent and environment and stimulates the exploration of new RL models and their comparison to standard ones.

We mention a few future directions. There is the question of proving lower bounds on the query complexity in computing $\varepsilon$-optimal policies and on regret. For the former, Luo et al. (2025) made some recent progress. Another direction is exploring variants of our RL model, e.g., allowing more or less samples to oracles $\mathcal{C}_p$ and $\mathcal{O}_p$ than $O(\tau_k)$. It is not hard to see that allowing $O(\tau_k^2)$ oracle calls during generative phases would lead to a $\text{poly} \log T$ regret also for classical algorithms, while $O(\tau_k^c)$ oracle calls for $c < 1$ would reinflate the dependence on $T$ to polynomial for quantum algorithms.

---

[5]If the MDP is *unichain* (Puterman, 2014, Section 8.3), $g^{\pi_t^\infty}(x)$ is independent of $x \in \mathcal{X}$ for any stationary policy, and so $g^* - g^{\pi_t^\infty}$ is the average regret per step.

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

## A PRELIMINARIES

For $n \in \mathbb{N} := \{1, 2, \dots\}$, let $[n] := \{1, \dots, n\}$. Given $u \in \mathbb{R}^n$, its $i$-th entry is denoted by $u(i)$, and its $\ell_1$ and $\ell_\infty$-norms are $\|u\|_1 := \sum_{i \in [n]} |u(i)|$ and $\|u\|_\infty := \max_{i \in [n]} |u(i)|$, respectively. Given a metric space $\mathcal{X}$, let $\mathscr{M}(\mathcal{X})$ be the $\sigma$-algebra of (Borel) measurable subsets of $\mathcal{X}$ and $\mathscr{P}(\mathcal{X})$ the collection of probability distributions on $\mathscr{M}(\mathcal{X})$. Let $\mathscr{B}(\mathcal{X})$ be the space of all bounded Borel measurable real-valued functions on $\mathcal{X}$. Given $u \in \mathscr{B}(\mathcal{X})$, let $\mathrm{sp}(u)$ be the span seminorm defined as $\mathrm{sp}(u) := \sup_{x \in \mathcal{X}} u(x) - \inf_{x \in \mathcal{X}} u(x)$. When $\mathcal{X}$ is finite, we interchangeably interpret $u \in \mathscr{B}(\mathcal{X})$ as a bounded real-valued function and a bounded real-valued vector $u \in \mathbb{R}^{|\mathcal{X}|}$. We use $e \in \mathscr{B}(\mathcal{X})$ to denote the all-ones function, $e(x) = 1$ for all $x \in \mathcal{X}$. Unless mentioned otherwise, $\widetilde{O}(\cdot)$ hides polylogarithmic factors, i.e., $\widetilde{O}(f(n)) = O(f(n) \operatorname{poly} \log f(n))$.

We shall need the following simple lemmas.

**Fact 1** (([Lakshmanan et al., 2015](), Lemma 2))**.** *For any sequence $z_1, \ldots, z_n \in \mathbb{R}$ with $0 \le z_k \le Z_{k-1} := \max\{1, \sum_{i=1}^{k-1} z_i\}$,*

$$\sum_{k=1}^{m} \frac{z_k}{Z_{k-1}^{1-\gamma}} \le \frac{Z_m^{\gamma}}{2^{\gamma} - 1} \quad \text{for any} \quad \gamma \in [0, 1].$$

**Lemma 2.** *For any sequence $z_1, \ldots, z_n \in \mathbb{R}$ with $0 \le z_k \le Z_{k-1} := \max\{1, \sum_{i=1}^{k-1} z_i\}$ and $Z_n \ge 4$,*

$$\sum_{k=1}^{n} \frac{z_k}{Z_{k-1}} \le 4 \log_2(Z_n/2).$$

*Proof.* We employ Fact 1 to obtain

$$\sum_{k=1}^{n} \frac{z_k}{Z_{k-1}} \le \sum_{k=1}^{n} \frac{z_k}{Z_{k-1}^{1-\gamma}} \le \frac{Z_n^{\gamma}}{2^{\gamma} - 1} \quad \text{for any} \quad \gamma \in [0, 1].$$

A simple calculation yields that the value $\gamma = \log_2 \left( \frac{\ln Z_n}{\ln(Z_n/2)} \right)$ (which is $\le 1$ since $Z_n \ge 4$) minimises the above right-hand side and thus leads to

$$\sum_{k=1}^{n} \frac{z_k}{Z_{k-1}} \le \frac{Z_n^{\log_2 \left( \frac{\ln Z_n}{\ln(Z_n/2)} \right)}}{\frac{\ln Z_n}{\ln(Z_n/2)} - 1} = \frac{\ln(Z_n/2)}{\ln 2} \left( 1 + \frac{\ln 2}{\ln(Z_n/2)} \right)^{\log_2 Z_n} \le \frac{4}{\ln 2} \ln(Z_n/2),$$

where the last inequality uses that $\left( 1 + \frac{\ln 2}{\ln(x/2)} \right)^{\log_2 x}$ is a decreasing function of $x$ and $Z_n \ge 4$. $\quad\square$

## A.1 BACKGROUND ON QUANTUM COMPUTATION

Little background on quantum computation is needed for our paper and we refer the reader to [Nielsen & Chuang](2010) for more information. The quantum state of a quantum system is described by a unit vector from a Hilbert space denoted by the ket notation $|\cdot\rangle$. A qubit, the quantum equivalent of a bit, is a quantum system described by a unit vector in $\mathbb{C}^2$, $\alpha|0\rangle + \beta|1\rangle$ with $\alpha, \beta \in \mathbb{C}$ such that $|\alpha|^2 + |\beta|^2 = 1$, while an $n$-qubit system is described by a unit vector in $\mathbb{C}^{2^n}$. The evolution of a quantum state $|\psi\rangle \in \mathbb{C}^{2^n}$ is described by a unitary operator $U \in \mathbb{C}^{2^n \times 2^n}$, $UU^\dagger = I$ where $U^\dagger$ is the Hermitian conjugate of $U$. In order to extract classical information from a quantum system, a quantum measurement is usually performed, which is a set $\{E_m\}_m$ of positive operators $E_m \succ 0$ that sum to identity, $\sum_m E_m = I$. The probability of measuring $E_m$ on $|\psi\rangle$ is $\langle \psi | E_m | \psi \rangle$. We use $\bar{0}$ to denote the all-zeros vector and $|\bar{0}\rangle$ to denote the state $|0\rangle \otimes \cdots \otimes |0\rangle$ where the number of qubits is clear from the context.

In this paper, we shall employ quantum oracles for functions $u \in \mathscr{B}(\mathcal{X})$ and probability distributions $p \in \mathscr{P}(\mathcal{X})$. We say we have quantum access to $u \in \mathscr{B}(\mathcal{X})$ if we have access to the oracle $\mathcal{O}_u : |x\rangle|\bar{0}\rangle \mapsto |x\rangle|u(x)\rangle$ and its inverse, and we say we have quantum sampling access to $p \in \mathscr{P}(\mathcal{X})$ if we have access to the oracle $\mathcal{O}_p : |\bar{0}\rangle \mapsto \int_{\mathcal{X}} \sqrt{p(\mathrm{d}x)}|x\rangle$ and its inverse. If $\mathcal{X} = \mathcal{S}$ is finite, then $\mathcal{O}_p : |\bar{0}\rangle \mapsto \sum_{s \in \mathcal{S}} \sqrt{p(s)}|s\rangle$. Quantum access to a function is usually referred to as a quantum random access memory (QRAM) ([Giovannetti et al., 2008a](); [b](); [Jaques & Rattew, 2023](); [Allcock et al., 2024]()). It is possible to build quantum access to a given $u \in \mathscr{B}(\mathcal{X})$ in $O(|\mathcal{X}|)$ time.

## A.2 CLASSICAL AND QUANTUM SUBROUTINES

In this section, we present the quantum subroutines that will be used for the rest of the paper, starting with the minimum/maximum finding algorithm by [Dürr & Høyer](1996).

**Fact 3** (Quantum max-finding ([Dürr & Høyer, 1996]()))**.** *Given quantum access to $u \in \mathscr{B}(\mathcal{X})$ via oracle $\mathcal{O}_u$, one can find $\max_{x \in \mathcal{X}} u(x)$ and $\min_{x \in \mathcal{X}} u(x)$ with probability $1 - \delta$ using $O(\sqrt{|\mathcal{X}|} \log \frac{1}{\delta})$ queries to $\mathcal{O}_u$.*

Another important quantum subroutine is approximating the mean of some random variable. Several quantum mean estimation algorithms have been proposed (Montanaro, 2015; van Apeldoorn, 2021; Hamoudi, 2021; Cornelissen et al., 2022; Kothari & O'Donnell, 2023). Here we shall employ the univariate version due to Kothari & O'Donnell (2023) and the multivariate version due to Cornelissen et al. (2022), or more specifically, the improved version due to Tang (2025).

**Fact 4** (Quantum mean estimation with variance (Kothari & O'Donnell, 2023, Theorem 1.1)).
*Let $\epsilon > 0$ and $\delta \in (0, 1)$. Assume quantum access to function $u : \mathcal{X} \to \mathbb{R}$ via oracle $\mathcal{O}_u$ and quantum sampling access to probability distribution $p \in \mathscr{P}(\mathcal{X})$ via oracle $\mathcal{O}_p$. Let $\sigma := \int_{\mathcal{X}} p(\mathrm{d}x)u(x)^2 - \left( \int_{\mathcal{X}} p(\mathrm{d}x)u(x) \right)^2$. There is a quantum algorithm that computes $\widetilde{\mu} \in \mathbb{R}$ such that $|\widetilde{\mu} - \int_{\mathcal{X}} p(\mathrm{d}x)u(x)| \leq \sqrt{\sigma}\epsilon$ with success probability $1 - \delta$ using $O\left(\frac{1}{\epsilon} \log \frac{1}{\delta}\right)$ queries to $\mathcal{O}_u, \mathcal{O}_p$ and their inverses.*

**Corollary 5** (Quantum mean estimation). *Let $\epsilon > 0$ and $\delta \in (0, 1)$. Assume quantum access to function $u : \mathcal{X} \to \mathbb{R}_+$ with known $\min_{x \in \mathcal{X}} u(x)$ and $\max_{x \in \mathcal{X}} u(x)$ via oracle $\mathcal{O}_u$ and quantum sampling access to probability distribution $p \in \mathscr{P}(\mathcal{X})$ via oracle $\mathcal{O}_p$. There is a quantum algorithm that computes $\widetilde{\mu} \in \mathbb{R}$ such that $|\widetilde{\mu} - \int_X p(\mathrm{d}x)u(x)| \leq \mathrm{sp}(u)\epsilon$ with success probability $1 - \delta$ using $O\left(\frac{1}{\epsilon} \log \frac{1}{\delta}\right)$ queries to $\mathcal{O}_u, \mathcal{O}_p$ and their inverses.*

**Fact 6** (Quantum multidimensional mean estimation with variance (Tang, 2025, Theorem I.2)). *Let $\epsilon > 0$ and $\delta \in (0, 1)$. Assume quantum access to $n$-dimensional function $f : \mathcal{X} \to \mathbb{R}^n$ via oracle $\mathcal{O}_f$ and quantum sampling access to probability distribution $p \in \mathscr{P}(\mathcal{X})$ via oracle $\mathcal{O}_p$. Let $\sigma := \sum_{i \in [n]} \int_{\mathcal{X}} p(\mathrm{d}x)f_i(x)^2 - \left( \int_{\mathcal{X}} p(\mathrm{d}x)f_i(x) \right)^2$. There is a quantum algorithm that computes $\widetilde{\mu} \in \mathbb{R}^n$ such that $\|\widetilde{\mu} - \int_X p(\mathrm{d}x)f(x)\|_\infty \leq \sqrt{\sigma}\epsilon$ with success probability $1 - \delta$ using $O\left(\frac{1}{\epsilon} \log \frac{n}{\delta}\right)$ quantum queries to $\mathcal{O}_p, \mathcal{O}_f$ and their inverses.*

Finally, we shall make use of Hoeffding and Bernstein inequalities.

**Fact 7** (Hoeffding inequality). *Let $\delta \in (0, 1)$, $u \in \mathcal{B}(\mathcal{X})$, and $x_1, \ldots, x_m \in \mathcal{X}$ samples from the distribution $p \in \mathscr{P}(\mathcal{X})$. Then*

$$\Pr\left[ \left| \frac{1}{m} \sum_{i=1}^m u(x_i) - \int_{\mathcal{X}} p(\mathrm{d}x)u(x) \right| \leq \|u\|_\infty \sqrt{\frac{2}{m} \ln \frac{2}{\delta}} \right] \geq 1 - \delta.$$

**Fact 8** (Bernstein inequality). *Let $\delta \in (0, 1)$, $u \in \mathcal{B}(\mathcal{X})$, and $x_1, \ldots, x_m \in \mathcal{X}$ samples from the distribution $p \in \mathscr{P}(\mathcal{X})$. Let $\sigma := \int_{\mathcal{X}} p(\mathrm{d}x)u(x)^2 - \left( \int_{\mathcal{X}} p(\mathrm{d}x)u(x) \right)^2$. Then*

$$\Pr\left[ \left| \frac{1}{m} \sum_{i=1}^m u(x_i) - \int_{\mathcal{X}} p(\mathrm{d}x)u(x) \right| \leq \sqrt{\frac{2\sigma}{m} \ln \frac{2}{\delta}} + \frac{2\|u\|_\infty}{3m} \ln \frac{2}{\delta} \right] \geq 1 - \delta.$$

### A.3 BACKGROUND ON MARKOV DECISION PROCESSES

In this paper, we are concerned with two types of discrete-time Markov decision processes (MDPs): *finite-horizon* and *infinite-horizon* MDPs. A finite-horizon MDP $M$ is described by a five-tuple $\langle \mathcal{X}, \mathcal{A}, H, p, r \rangle$, while an infinite-horizon MDP $M$ is described by a four-tuple $\langle \mathcal{X}, \mathcal{A}, p, r \rangle$. The Borel spaces $\mathcal{X}$ and $\mathcal{A}$ denote the *state* and *action* spaces, respectively.[6] We assume that $\mathcal{X}$ is compact and $\mathcal{A}$ is finite with size $A$. The horizon $H$ denotes the length or number of steps of the MDP. For infinite-horizon MDPs, $H = \infty$. The *reward function* $r : \mathcal{X} \times \mathcal{A} \to [0, 1]$ is a measurable function and the *stochastic kernels* $p(\cdot|x, a)$ denote the *transition probabilities* to the next state given the current state-action pair $(x, a)$.[7] For simplicity, we assume that $p(\cdot|x, a)$ is weakly continuous in $(x, a) \in \mathcal{X} \times \mathcal{A}$. In this work, we shall make a distinction between finite and infinite state spaces. For such, we reserve the letter $\mathcal{S}$ for a finite state space, while $\mathcal{X}$ can be an arbitrary compact state space.

---

[6] Some authors also include the set $\{\mathcal{A}_x : x \in \mathcal{X}\}$ of allowable actions in state $x \in \mathcal{X}$ in the definition of an MDP. Here we assume for simplicity that $\mathcal{A}_x = \mathcal{A}$ for all $x \in \mathcal{X}$.

[7] For finite-horizon MDPs, the reward function and stochastic kernels can depend on the time step as $r : \mathcal{X} \times \mathcal{A} \times [H] \to [0, 1]$ and $p : \mathcal{X} \times \mathcal{A} \times [H] \to \mathscr{P}(\mathcal{X})$. We assume for simplicity that their form is time independent.

An MDP models the interaction between an agent and the reinforcement learning environment. At any time step $t \in \mathbb{N}$, the agent in a particular state $x_t \in \mathcal{X}$ chooses an action $a_t \in \mathcal{A}$, obtains a reward $r(x_t, a_t)$, and moves to a new state $x_{t+1} \in \mathcal{X}$ with probability $p(x_{t+1}|x_t, a_t)$. For finite-horizon MDPs, this interaction is performed $H$ times, while for infinite-horizon MDPs, this interaction can be performed indefinitely. The agent chooses an action $a_t \in \mathcal{A}$ according to a *randomised Markovian decision rule* $\pi_t : \mathcal{X} \rightarrow \mathscr{P}(\mathcal{A})$, i.e., a stochastic kernel on $\mathcal{A}$ given $\mathcal{X}$. Instead of writing $\pi_t(x)(\cdot) \in \mathscr{P}(\mathcal{A})$ we employ the notation $\pi_t(\cdot|x)$. A *randomised Markovian policy* is a sequence $\pi = (\pi_t)_t$ of randomised Markovian decision rules on $\mathcal{A}$ given $\mathcal{X}$.[8] We say that a policy is *deterministic* if it is a sequence of stochastic kernels $\pi = (\pi_t)_t$ on $\mathcal{A}$ given $\mathcal{X}$ such that $\pi_t(a|x) = 1$ for some $a \in \mathcal{A}$. A policy is said to be *stationary* if it is a constant sequence $\pi = (\pi_t)_t$ of decision rules on $\mathcal{A}$ given $\mathcal{X}$ such that $\pi_t = \pi_{t'}$ for all $t, t'$. Given a decision rule $d$, we shall employ the notation $d^\infty$ for the stationary policy $\pi = (d, d, \dots)$ in infinite-horizon MDPs. The set of all randomised and deterministic decision rules are denoted by $\mathcal{D}^\mathrm{R}$ and $\mathcal{D}^\mathrm{D}$, and the set of all randomised and deterministic policies are denoted by $\Pi^\mathrm{R}$ and $\Pi^\mathrm{D}$, respectively. Given $d \in \mathcal{D}^\mathrm{R}$, define $r_d(x)$ and $p_d(x'|x)$

$$r_d(x) := \mathbb{E}_{a \sim d(\cdot|x)}[r(x,a)] \qquad \text{and} \qquad p_d(x'|x) := \mathbb{E}_{a \sim d(\cdot|x)}[p(x'|x,a)].$$

In this work, we shall consider *weakly communicating* infinite-horizon MDPs (Puterman, 2014, Section 8.3) (there is no need to constrain the class of finite-horizon MDPs). We say that an infinite-horizon MDP is weakly communicating if there is a *closed* set of states, with each state in that set accessible from every other state in that set under some deterministic stationary policy, plus a possibly empty set of states which is transient under every policy. Here, a state $x \in \mathcal{X}$ is *transient* if $\mathbb{E}[\tau_x] < \infty$, where the random variable $\tau_x$ represents the number of visits to state $x$.

A policy $\pi = (\pi_t)_t$ together with an initial distribution $\mu$ of the system state induces a probability measure $P_\mu^\pi$ on $(\Omega, \mathscr{M}(\Omega))$, where $\Omega = (\mathcal{X} \times \mathcal{A})^H$ (finite-horizon) or $\Omega = (\mathcal{X} \times \mathcal{A})^\infty$ (infinite-horizon), through

$$P_\mu^\pi(X_1 = x) = \mu(x),$$
$$P_\mu^\pi(A_t = a | X_t = x_t) = \pi_t(a|x_t),$$
$$P_\mu^\pi(X_{t+1} = x | X_t = x_t, A_t = a_t) = p(x|x_t, a_t).$$

We denote by $\mathbb{E}_\mu^\pi$ the expectation with respect to $P_\mu^\pi$. If $\mu = \delta_x$, we write simply $P_x^\pi$ and $\mathbb{E}_x^\pi$.

### A.3.1 FINITE-HORIZON MDPs

For finite-horizon MPDs with horizon $H$, the standard criterion to evaluate the performance of an agent in a RL environment is the *expected total reward* criterion. The expected total reward $V_1^\pi(x)$ over the decision making horizon $H$ when executing policy $\pi$ with initial state $x \in \mathcal{X}$ is

$$V_1^\pi(x) := \mathbb{E}_x^\pi \left[ \sum_{t=1}^{H} r(x_t, a_t) \right].$$

More generally, for $t \in [H]$, we define the *value function $V_t^\pi : \mathcal{X} \rightarrow \mathbb{R}$* as the total expected reward obtained by using policy $\pi \in \Pi^\mathrm{R}$ at time steps $t, t+1, \dots, H$, i.e.,

$$V_t^\pi(x) := \mathbb{E}_{x_t=x}^\pi \left[ \sum_{t'=t}^{H} r(x_{t'}, a_{t'}) \right] \qquad \forall x \in \mathcal{X}.$$

We say that a policy $\pi_\varepsilon \in \Pi^\mathrm{R}$ is $\varepsilon$-optimal for $\varepsilon \geq 0$ if

$$V_1^{\pi_\varepsilon}(x) \geq V_1^\pi(x) - \varepsilon \qquad \forall x \in \mathcal{X}, \pi \in \Pi^\mathrm{R},$$

and a policy $\pi^* \in \Pi^\mathrm{R}$ is *optimal* if $V_1^{\pi^*}(x) \geq V_1^\pi(x)$ for all $x \in \mathcal{X}, \pi \in \Pi^\mathrm{R}$. The optimal total expected reward (or optimal value) $V_t^*(x)$, for $t \in [H]$, on initial state $x \in \mathcal{X}$ is defined as

$$V_t^*(x) := \sup_{\pi \in \Pi^\mathrm{R}} V_t^\pi(x).$$

---

[8]It is possible to consider history-dependent policies. The choice for Markovian policies is without loss of generality since we can construct a randomised Markov policy from a history-dependent policy with the same probability distribution of states and actions (Puterman, 2014, Theorem 5.5.1).

We let $\mathcal{L}_d : \mathscr{B}(\mathcal{X}) \to \mathscr{B}(\mathcal{X})$ be the Bellman operator associated with decision rule $d \in \mathcal{D}^{\mathrm{R}}$ and $\mathcal{L} : \mathscr{B}(\mathcal{X}) \to \mathscr{B}(\mathcal{X})$ be the optimal Bellman operator defined, $\forall u \in \mathscr{B}(\mathcal{X}), x \in \mathcal{X}$, as

$$(\mathcal{L}_d u)(x) := r_d(x) + \int_{\mathcal{X}} p_d(\mathrm{d}x'|x)u(x') \quad \text{and} \quad (\mathcal{L}u)(x) := \max_{d \in \mathcal{D}^{\mathrm{D}}}\{(\mathcal{L}_d u)(x)\}.$$

We also write $\mathcal{L}_d = \mathcal{L}_a$ for a decision rule $d(x) = a \in \mathcal{A}$. Note that $\mathcal{L}u = \max_{d \in \mathcal{D}^{\mathrm{D}}} \mathcal{L}_d u = \max_{a \in \mathcal{A}} \mathcal{L}_a u$. The operator $\mathcal{L}$ is monotonic, i.e., $u \le v \implies \mathcal{L}u \le \mathcal{L}v$, and is non-expansive, i.e., $\mathrm{sp}(\mathcal{L}u - \mathcal{L}v) \le \mathrm{sp}(u - v)$ and $\|\mathcal{L}u - \mathcal{L}v\|_\infty \le \|u - v\|_\infty$ for all $u, v \in \mathscr{B}(\mathcal{X})$ (Puterman, 2014, Proposition 6.2.4). In order to derive convergence results, one usually assumes that $\mathcal{L}$ is also a span contraction.

**Definition 9** (*J*-stage *ν*-span contraction)**.** *Given $J \in \mathbb{N}$ and $\nu \in [0, 1)$, we say that an operator $\mathcal{N} : \mathscr{B}(\mathcal{X}) \to \mathscr{B}(\mathcal{X})$ is a J-stage ν-span contraction if* $\mathrm{sp}(\mathcal{N}^J u - \mathcal{N}^J v) \le \nu\,\mathrm{sp}(u - v) \ \forall u, v \in \mathscr{B}(\mathcal{X})$.

Under some conditions on the stochastic kernel $p$, one can show that $\mathcal{L}$ is a $J$-stage span contraction, see, e.g., (Puterman, 2014, Theorems 8.5.2 & 8.5.3).

The equations

$$u_{H+1} \equiv 0 \qquad \text{and} \qquad u_t = \mathcal{L}u_{t+1}, \quad t \in [H],$$

are called *optimality equations*. It is known (Puterman, 2014, Theorem 4.5.1) that any set of solutions $\{u_t\}_{t \in [H]} \subset \mathscr{B}(\mathcal{X})$ to the optimality equations are such that $u_t(x) = V_t^*(x) \ \forall x \in \mathcal{X}, t \in [H]$.

### A.3.2 Infinite-horizon MDPs

For infinite-horizon MDPs, the chosen criterion in this paper to evaluate the performance of an agent in a RL environment is the *average reward* criterion. The average reward (or gain) $g^\pi(x)$ when executing policy $\pi$ with initial state $x \in \mathcal{X}$ is defined by[9]

$$g^\pi(x) := \lim_{T \to \infty} \mathbb{E}_x^\pi\left[\frac{1}{T}\sum_{t=1}^T r(x_t, a_t)\right].$$

We say that a policy $\pi^\varepsilon \in \Pi^{\mathrm{R}}$ is $\varepsilon$-optimal for $\varepsilon \ge 0$ if

$$g^{\pi^\varepsilon}(x) \ge g^\pi(x) - \varepsilon \quad \forall x \in \mathcal{X}, \pi \in \Pi^{\mathrm{R}}.$$

If $\varepsilon = 0$, the policy $\pi^* \in \Pi^{\mathrm{R}}$ is said to be *optimal* if $g^{\pi^*}(x) \ge g^\pi(x)$ for all $x \in \mathcal{X}, \pi \in \Pi^{\mathrm{R}}$. The *optimal average reward* $g^*(x)$ on initial state $x \in \mathcal{X}$ is defined as

$$g^*(x) := \sup_{\pi \in \Pi^{\mathrm{R}}} g^\pi(x).$$

An optimal policy $\pi^* \in \Pi^{\mathrm{R}}$ exists whenever $g^{\pi^*}(x) = g^*(x)$ for all $x \in \mathcal{X}$. The *bias* $h^{d^\infty}(x)$ of a stationary policy $d^\infty, d \in \mathcal{D}^{\mathrm{R}}$, is the expected total difference between the reward and the stationary average reward, i.e.,[10]

$$h^{d^\infty}(x) := \lim_{T \to \infty} \mathbb{E}_x^{d^\infty}\left[\sum_{t=1}^T r_d(x_t) - g^{d^\infty}(x_t)\right] \qquad \forall x \in \mathcal{X}. \tag{2}$$

The Bellman operator $\mathcal{L}_d : \mathscr{B}(\mathcal{X}) \to \mathscr{B}(\mathcal{X})$ associated with decision rule $d \in \mathcal{D}^{\mathrm{R}}$ and the optimal Bellman operator $\mathcal{L} : \mathscr{B}(\mathcal{X}) \to \mathscr{B}(\mathcal{X})$ are similarly defined as for finite-horizon MDPs.

Under suitable conditions on the kernels $p(\cdot|x, a)$, the optimal gain $g^*(x)$ is state independent, i.e., $g^*(x) = g^*$ for all $x \in \mathcal{X}$. Moreover, for any measurable stationary policy $d^\infty, d \in \mathcal{D}^{\mathrm{R}}$, the gain $g^{d^\infty}$ and bias $h^{d^\infty}$ satisfy the system of evaluation equations

$$g^{d^\infty}(x) = \int_{\mathcal{X}} p_d(\mathrm{d}y|x)g^{d^\infty}(y) \qquad \text{and} \qquad h^{d^\infty} = \mathcal{L}_d h^{d^\infty} - g^{d^\infty}.$$

---

[9]More generally, the average reward is defined as $g^\pi(x) = \limsup_{T \to \infty} \frac{1}{T}\mathbb{E}_x^\pi\left[\sum_{t=1}^T r(x_t, a_t)\right]$.

[10]This is valid for *aperiodic* chains. For periodic chains, Eq. (2) holds in the Cesaro limit sense.

Finally, there is an optimal stationary deterministic policy $(d^*)^\infty$, $d \in \mathcal{D}^\mathrm{D}$, for which $(g^*, h^*) = (g^{(d^*)^\infty}, h^{(d^*)^\infty})$ satisfy the *average cost optimality equation*

$$h^* = \mathcal{L}h^* - g^*e.$$

In the finite-state-space setting, weakly communicating MDPs naturally satisfy all of the above, plus the fact that any optimal policy $\pi^* \in \arg\max_{\pi \in \Pi^\mathrm{R}} g^\pi(s)$ has constant gain. In the continuous-state-space setting with compact $\mathcal{X}$, extra requirements are needed on the stochastic kernels $p(\cdot|x, a)$, e.g., geometric convergence to an invariant probability measure. We refer the reader to (Saldi et al., 2017, Theorem 2.5), (Hernández-Lerma & Lasserre, 2012, Theorem 5.5.4), (Hernandez-Lerma, 2001, Chapter 3), (Gordienko & Hernández-Lerma, 1995, Theorem 2.6 & Lemma 3.4), (Jaśkiewicz & Nowak, 2006, Theorem 3), (Vega-Amaya, 2003, Theorem 3.3), and (Hernandez-Lerma & Lasserre, 1999, Chapter 10) for more information. From now on, we shall not worry about such conditions and assume the above properties hold.

## A.4 Discretization of state space

We briefly review a procedure to obtain finite-state models which will be used on continuous state-space MDPs. See Saldi et al. (2017) for more information. Consider a continuous state space $\mathcal{X}$ and a metric $m_\mathcal{X}$ on $\mathcal{X}$. By assumption, $\mathcal{X}$ is compact and hence totally bounded. Therefore, there exists a sequence $(\{s_{n,i}\}_{i \in [k_n]})_{n \geq 1}$ of finite sets such that, for all $n \in \mathbb{N}$,

$$\min_{i \in [k_n]} m_\mathcal{X}(x, s_{n,i}) < \frac{1}{n} \quad \text{for all } x \in \mathcal{X}.$$

The finite grid $\mathcal{S}_n := \{s_{n,i}\}_{i \in [k_n]}$ is called a $\frac{1}{n}$-net in $\mathcal{X}$. Define the function

$$Q_{n,\mathcal{X}} : \mathcal{X} \to \mathcal{S}_n \quad \text{as} \quad Q_{n,\mathcal{X}}(x) = \arg\min_{s \in \mathcal{S}_n} m_\mathcal{X}(x, s),$$

where ties are broken so that $Q_{n,\mathcal{X}}$ is measurable. The map $Q_{n,\mathcal{X}}$ is often called a nearest neighbour quantizer with respect to distortion measure $m_\mathcal{X}$ (Gray & Neuhoff, 1998). For each $n$, this mapping induces a partition $\{\mathcal{X}_{n,i}\}_{i \in [k_n]}$ of $\mathcal{X}$ given by

$$\mathcal{X}_{n,i} = \{x \in \mathcal{X} : Q_{n,\mathcal{X}}(x) = s_{n,i}\},$$

with diameter $\mathrm{diam}(\mathcal{X}_{n,i}) = \sup_{x,x' \in \mathcal{X}_{n,i}} m_\mathcal{X}(x, x') < \frac{2}{n}$. Given $s \in \mathcal{S}$, we also use the notation $\mathcal{X}_n(s)$ to denote $\{x \in \mathcal{X} : Q_{n,\mathcal{X}}(x) = s\}$.

As an example, consider the 1-dimensional setting where the $\mathcal{X} = [0, 1]$. The $\frac{1}{n}$-net $\mathcal{S}_n$ partitions $\mathcal{X}$ into $n$ intervals $\mathcal{X}_{n,0}, \ldots, \mathcal{X}_{n,n-1}$, where

$$\mathcal{X}_{n,0} = \left[0, \frac{1}{n}\right] \quad \text{and} \quad \mathcal{X}_{n,i} = \left(\frac{i}{n}, \frac{i+1}{n}\right] \text{ for } i = 1, \ldots, n-1.$$

Each interval $\mathcal{X}_{n,i}$ is represented by a state $s_{n,i} \in \mathcal{S}_n$. For the rest of our paper, for simplicity we assume an underlying Euclidean topology for $\mathcal{X}$, in which case $m_\mathcal{X}$ is simply the $\ell_2$-norm $\|\cdot\|_2$.

We introduce natural assumptions for the transition probabilities and rewards in close states, which will be used throughout the paper. Similar assumptions have been considered in Saldi et al. (2017); Ortner & Ryabko (2012); Kara & Yuksel (2023).

**Assumption 10** (Hölder continuity). *The reward function $r : \mathcal{X} \times \mathcal{A} \to [0, 1]$ and stochastic kernel $p : \mathcal{X} \times \mathcal{A} \to \mathscr{P}(\mathcal{X})$ are Hölder continuous, i.e., there are constants $L, \alpha \geq 0$ such that*

$$|r(x, a) - r(x', a)| \leq L\|x - x'\|_2^\alpha \quad \text{for all} \quad (x, x', a) \in \mathcal{X} \times \mathcal{X} \times \mathcal{A},$$

$$\|p(\cdot|x, a) - p(\cdot|x', a)\|_\mathrm{tvd} \leq L\|x - x'\|_2^\alpha \quad \text{for all} \quad (x, x', a) \in \mathcal{X} \times \mathcal{X} \times \mathcal{A}.$$

# B The reinforcement learning model of Zhong et al.

Zhong et al. (2024) proposed quantum RL algorithms for finite-horizon finite-state-space MDPs. In order to do so, the authors employed an exploration-generative learning model that shares several similarities to ours. Although not explicitly stated in their paper, their RL model alternates between

exploration phases that accumulate rewards and thus regret, and *free* generative phases where no rewards are obtained. Their exploration phase is virtually the same as ours, during which the environment's state $x_t$ jumps to $x_{t+1} \sim p(\cdot|x_t, a_t)$ after the agent chooses an action $a_t \sim \pi_t(\cdot|x_t)$ and receives reward $r(x_t, a_t)$. Even though Zhong et al. (2024) phrased such interaction in a quantum fashion using the oracle $\mathcal{O}_p$ plus an action oracle $\mathcal{O}_{\pi_t}$ for the decision rule $\pi_t \in \mathcal{D}^R$ defined as

$$\mathcal{O}_{\pi_t} : |x\rangle|\bar{0}\rangle \mapsto \sum_{a \in \mathcal{A}} \sqrt{\pi_t(a|x)}|x\rangle|a\rangle \qquad \forall x \in \mathcal{X},$$

there is absolutely no need for such and the agent-environment interaction can be done entirely classically. Indeed, Zhong et al. (2024) model the agent-environment interaction during an exploration phase as repeated applications of $\mathcal{O}_p \mathcal{O}_{\pi_t}$ followed by a measurement, which returns $x_{t+1}$ from the distribution $p(\cdot|x_t, a_t)$, just as in our case.

Regarding their free generative steps, Zhong et al. (2024) allow the application of $\mathcal{O}_p$ or $\mathcal{O}_p^\dagger$ to a register *belonging* to the environment which contains a quantum state $|x_t\rangle_\mathcal{X}$ corresponding to the state $x_t$ of the environment which has been visited in an exploration step. Then, quantum mean estimation is applied and, in (Zhong et al., 2024, Algorithm 1), they indicate that it can be implemented using the stored states $\mathcal{O}_p|x_t\rangle_\mathcal{X}$ and $\mathcal{O}_p^\dagger|x_t\rangle_\mathcal{X}$. This claim is not actually correct because quantum mean estimation requires being able to apply oracles to quantum states created during the run of the mean estimation. This issue can be fixed by allowing quantum mean estimation to query $\mathcal{O}_p$ on superpositions of $|x_t\rangle_\mathcal{X}$. This effectively results in a model similar to ours, with groups of exploration steps being exploration phases and runs of quantum mean estimation being generative phases. The main difference between both models, however, is the choice for the degree of control over the environment: either the agent can manipulate the environment's state during generative phases and apply $\mathcal{O}_p$ and $\mathcal{O}_p^\dagger$ onto states that the agent prepares (our model), or the environment's state $|x_t\rangle_\mathcal{X}$ is fixed from the end of the previous exploration phase and applications of $\mathcal{O}_p$ and $\mathcal{O}_p^\dagger$ are done on such state $|x_t\rangle_\mathcal{X}$ only. Finally, the length of generative phases in Zhong et al. (2024) is also proportional to the previous exploration period $\tau_k$. This is implicit when they use quantum multi-dimensional amplitude estimation (van Apeldoorn, 2021) with $O(n(x, a))$ iterations for each state-action pair $(x, a) \in \mathcal{X} \times \mathcal{A}$, where $n(x, a)$ is the number of times $(x, a)$ has been observed in the last exploration phase.

## C    COMPUTING OPTIMAL POLICIES FOR FINITE-HORIZON MDPS

Several RL algorithms (Auer & Ortner, 2006; Bartlett & Tewari, 2009; Auer et al., 2008; Ortner & Ryabko, 2012; Lakshmanan et al., 2015; Fruit et al., 2018) compute approximate optimal policies for the target unknown MDP $M$ given empirical estimates of the true stochastic kernels $p$. As explained in Section 3, our RL algorithms also require approximate optimal policies for $M$, therefore, in this section we study the problem of finding approximate optimal policies for $M$ if one is given access to oracle $\mathcal{O}_p$ and its inverse. We start by considering finite-horizon MDPs $M = \langle \mathcal{X}, \mathcal{A}, H, p, r \rangle$. As briefly mentioned in Appendix A.3, a simple backward induction algorithm can be used to find $V_t^*(x)$ for all $t \in [H]$: For $t = H, H-1, \ldots, 1$, compute

$$\forall x \in \mathcal{X} : \qquad u_t(x) = (\mathcal{L}u_{t+1})(x) \qquad \text{and} \qquad \pi_t(x) \in \arg\max_{a \in \mathcal{A}}\{(\mathcal{L}_a u_{t+1})(x)\}.$$

Consider the policy $\pi = (\pi_t)_{t \in [H]} \in \Pi^D$. Then one can prove that $u_t(x) = V_t^\pi(x) = V_t^*(x)$ for all $x \in \mathcal{X}$, meaning that $u_t$ is the optimal total expected reward from time $t$ onward and $\pi$ is an optimal policy (Puterman, 2014, Theorem 4.5.1). If $\mathcal{X} = \mathcal{S}$ is finite with size $S$, then this simple backward induction algorithm has time complexity $O(HS^2 A)$. However, it is possible to improve this complexity within a generative model with access to oracle $\mathcal{C}_p$, as we shall see next.

### C.1    CLASSICAL APPROXIMATE BACKWARD INDUCTION ALGORITHM

The classical complexity of computing $\varepsilon$-approximate optimal policies for finite-horizon MDPs with generative model has been completely characterised by Sidford et al. (2018a). By leveraging several techniques, the authors proposed a modern version of the above backward induction algorithm with sample complexity $\widetilde{O}\left(\frac{H^3 SA}{\varepsilon^2}\right)$, which is optimal up to logarithmic factors (Sidford et al., 2018a,

Corollary F.7). In this section, we review their algorithm, which is described in Algorithm 2, already generalised to compact state spaces.

The backward induction algorithm from Sidford et al. (2018a) works in *epochs*. By starting at the beginning of the $k$-th epoch with some initial functions $u_1^{(k-1)}, u_2^{(k-1)}, \ldots, u_H^{(k-1)} : \mathcal{X} \to \mathbb{R}$ such that $0 \leq V_t^*(x) - u_t^{(k-1)}(x) \leq 2\epsilon_k$ for all $x \in \mathcal{X}$ and $t \in [H]$, by the end of the $k$-th epoch (which is the beginning of the $(k+1)$-th epoch), their algorithm produces $u_1^{(k)}, u_2^{(k)}, \ldots, u_H^{(k)} : \mathcal{X} \to \mathbb{R}$ such that $0 \leq V_t^*(x) - u_t^{(k)}(x) \leq \epsilon_k$, thus halving the initial error. By starting with zero functions $u_1^{(0)}, \ldots, u_H^{(0)} \equiv 0$ at the first epoch, and noticing that $V_t^*(x) \leq H$, then one only needs to iterate $O(\log(H/\varepsilon))$ times in order to obtain a final $\varepsilon$-approximation.

Now let us focus on a single epoch. The algorithm from Sidford et al. (2018a) uses three crucial techniques: *monotonicity*, *variance reduction*, and *total-variance* techniques. The monotonicity technique means maintaining the monotonicity condition $u_t^{(k)}(x) \leq (\mathcal{L}_{\pi_t^{(k)}} u_{t+1}^{(k)})(x)$ throughout the epoch, which guarantees that $u_t^{(k)}(x) \leq V_t^{\pi^{(k)}}(x) \leq V_t^*(x)$ by the monotonicity of the Bellman operator. Therefore, an $\epsilon$-optimal value function $u_t^{(k)}(x)$ satisfying such monotonicity condition yields an $\epsilon$-optimal policy, which is not true otherwise. In general, an $\epsilon$-optimal value function yields an $2\epsilon/H$-optimal greedy policy in the worst case (Singh & Yee, 1994; Bertsekas, 2012).

Naively, at each time step $t \in [H]$ of each epoch $k$, we need to estimate the quantity $\int_{\mathcal{X}} p(\mathrm{d}x'|x,a) u_t^{(k)}(x')$ up to additive error $\epsilon_k/H$ in order to achieve the new value functions $u_1^{(k)}, \ldots, u_H^{(k)}$ with error $\epsilon_k$. Since $\|u_t^{(k)}\|_\infty \leq H$, by a Hoeffding bound $\widetilde{O}(H^4/\epsilon_k^2)$ samples would suffice for each time step $t \in [H]$, leading to a total of $\widetilde{O}(H^5/\epsilon_k^2)$ samples for all time steps. The variance reduction technique rewrites the standard backward induction iteration as

$$u_t^{(k)}(x) \leftarrow \max_{a \in \mathcal{A}} \left\{ r(x,a) + \int_{\mathcal{X}} p(\mathrm{d}x'|x,a)(u_{t+1}^{(k)}(x') - u_{t+1}^{(k-1)}(x')) + \int_{\mathcal{X}} p(\mathrm{d}x'|x,a) u_{t+1}^{(k-1)}(x') \right\}.$$

The main idea is that the quantities $\int_{\mathcal{X}} p(\mathrm{d}x'|x,a) u_t^{(k-1)}(x')$ for all $t \in [H]$ can be computed at the beginning of the epoch using the same batch of $\widetilde{O}(H^4/\epsilon^2)$ samples, which saves a factor of $H$. Regarding the other quantity $\int_{\mathcal{X}} p(\mathrm{d}x'|x,a)(u_{t+1}^{(k)}(x') - u_{t+1}^{(k-1)}(x'))$, since $\|u_{t+1}^{(k)} - u_{t+1}^{(k-1)}\|_\infty \leq \epsilon_k$ by the monotonicity condition, it can be approximated up to error $\epsilon_k/2H$ using only $\widetilde{O}(H^2)$ samples for each time step $t \in [H]$, leading to a total of $\widetilde{O}(H^3)$ samples over all time steps.

Finally, in order to reduce the sample complexity dependence on $H$ from $O(H^4)$ down to $O(H^3)$, the final technique, total variance, is employed, the main idea being that the true error accumulates much less than the naive sum of estimation errors at each time step. This means that one does not require to set the update error to be $\epsilon_k/H$ at each time step. It can be shown (Sidford et al., 2018a) (see Fact 13 below) that the total accumulation error is $\sqrt{H^3/m}$, thus picking $m = O(H^3/\epsilon_k^2)$ samples is sufficient to obtain a total error equal to $\epsilon_k$.

By putting all the three aforementioned techniques together, one arrives at the result of Sidford et al. (2018a). For completeness, we include a proof of correctness for Algorithm 2, already generalised to compact state spaces. We start with the empirical estimation error for the means and variances on Lines 7 and 8 in Algorithm 2.

**Lemma 11.** *Let $\mathcal{X}$ a compact set with $\frac{1}{n}$-net $\mathcal{S}_n$, $\mathcal{A}$ a finite set, and functions $\{u_t \in \mathscr{B}(\mathcal{X})\}_{t \in [H]}$. For each $(s,a) \in \mathcal{S}_n \times \mathcal{A}$, consider samples $x_{s,a}^{(1)}, \ldots, x_{s,a}^{(m)} \in \mathcal{X}$ from $p(\cdot|s,a)$. For $t \in [H]$, let $\mu_t(x,a) := \int_{\mathcal{X}} p(\mathrm{d}x'|x,a) u_t(x')$ and $\sigma_t(x,a) := \int_{\mathcal{X}} p(\mathrm{d}x'|x,a) u_t(x')^2 - \left( \int_{\mathcal{X}} p(\mathrm{d}x'|x,a) u_t(x') \right)^2$. Let the empirical estimates be*

$$\widetilde{\mu}_t(s,a) := \frac{1}{m} \sum_{i=1}^m u_t(x_{s,a}^{(i)}),$$

$$\forall (s,a,t) \in \mathcal{S}_n \times \mathcal{A} \times [H]:$$

$$\widetilde{\sigma}_t(s,a) := \frac{1}{m} \sum_{i=1}^m u_t(x_{s,a}^{(i)})^2 - \left( \frac{1}{m} \sum_{i=1}^m u_t(x_{s,a}^{(i)}) \right)^2.$$

*Then, with probability at least $1 - \delta$, for all $(s, a, t) \in \mathcal{S}_n \times \mathcal{A} \times [H]$,*

$$|\widetilde{\mu}_t(s,a) - \mu_t(s,a)| \leq \sqrt{\frac{2\sigma_t(s,a)}{m} \ln \frac{8H|\mathcal{S}_n|A}{\delta}} + \frac{2\|u_t\|_\infty}{3m} \ln \frac{8H|\mathcal{S}_n|A}{\delta},$$

$$|\widetilde{\sigma}_t(s,a) - \sigma_t(s,a)| \leq 4\sqrt{\frac{2}{m} \ln \frac{8H|\mathcal{S}_n|A}{\delta}} \|u_t\|_\infty^2.$$

*Moreover, $|\mu_t(x,a) - \mu_t(s,a)| \leq Ln^{-\alpha}\|u_t\|_\infty$ and $|\sigma_t(x,a) - \sigma_t(s,a)| \leq 4Ln^{-\alpha}\|u_t\|_\infty^2$ under Assumption 10 for all $s \in \mathcal{S}_n$, $x \in \mathcal{X}(s)$, $a \in \mathcal{A}$, and $t \in [H]$.*

*Proof.* By Bernstein's inequality (Fact 8) and a union bound over $\mathcal{S}_n \times \mathcal{A} \times [H]$, then, with probability at least $1 - \frac{\delta}{4}$, for all $(s, a, t) \in \mathcal{S}_n \times \mathcal{A} \times [H]$,

$$|\widetilde{\mu}_t(s,a) - \mu_t(s,a)| \leq \sqrt{\frac{2\sigma_t(s,a)}{m} \ln \frac{8H|\mathcal{S}_n|A}{\delta}} + \frac{2\|u_t\|_\infty}{3m} \ln \frac{8H|\mathcal{S}_n|A}{\delta},$$

given the first inequality. Regarding the second inequality, by Hoeffding's inequality (Fact 7) and a union bound over $\mathcal{S}_n \times \mathcal{A} \times [H]$, with probability at least $1 - \frac{\delta}{2}$, for all $(s, a, t) \in \mathcal{S}_n \times \mathcal{A} \times [H]$,

$$|\widetilde{\mu}_t(s,a) - \mu_t(s,a)| \leq \sqrt{\frac{2}{m} \ln \frac{8H|\mathcal{S}_n|A}{\delta}} \|u_t\|_\infty,$$

$$\left| \frac{1}{m} \sum_{i=1}^m u_t(x_{s,a}^{(i)})^2 - \int_{\mathcal{X}} p(\mathrm{d}x'|s,a) u_t(x')^2 \right| \leq \sqrt{\frac{2}{m} \ln \frac{8H|\mathcal{S}_n|A}{\delta}} \|u_t\|_\infty^2.$$

Call $\theta := \frac{1}{m} \ln \frac{8H|\mathcal{S}_n|A}{\delta}$. Now notice that, conditioned on the above,

$$\begin{aligned}
|\widetilde{\mu}_t(s,a)^2 - \mu_t(s,a)^2| &= |\widetilde{\mu}_t(s,a) + \mu_t(s,a)||\widetilde{\mu}_t(s,a) - \mu_t(s,a)| \\
&\leq (2\mu_t(s,a) + \sqrt{2\theta}\|u_t\|_\infty)\sqrt{2\theta}\|u_t\|_\infty \\
&\leq (2\sqrt{2\theta} + 2\theta)\|u_t\|_\infty^2 \qquad\qquad (\mu_t(s,a) \leq \|u_t\|_\infty) \\
&\leq 3\sqrt{2\theta}\|u_t\|_\infty^2,
\end{aligned}$$

assuming that $m$ is large enough so that $2\theta \leq 1$. This means that, with probability at least $1 - \frac{\delta}{2}$, for all $(s, a, t) \in \mathcal{S}_n \times \mathcal{A} \times [H]$,

$$|\widetilde{\sigma}_t(s,a) - \sigma_t(s,a)| \leq |\widetilde{\mu}_t(s,a)^2 - \mu_t(s,a)^2| + \left| \frac{1}{m}\sum_{i=1}^m u_t(x_{s,a}^{(i)})^2 - \int_{\mathcal{X}} p(\mathrm{d}x'|x,a)u_t(x')^2 \right|$$

$$\leq 4\sqrt{\frac{2}{m} \ln \frac{8H|\mathcal{S}_n|A}{\delta}}\|u_t\|_\infty^2,$$

as required. Finally, by Assumption 10, for all $s \in \mathcal{S}_n$, $x \in \mathcal{X}(s)$, $a \in \mathcal{A}$, and $t \in [H]$,

$$|\mu_t(x,a) - \mu_t(s,a)| \leq \int_{\mathcal{X}} |p(\mathrm{d}x'|x,a) - p(\mathrm{d}x'|s,a)||u_t(x')| \leq Ln^{-\alpha}\|u_t\|_\infty,$$

$$\left| \int_{\mathcal{X}} p(\mathrm{d}x'|x,a)u_t(x')^2 - \int_{\mathcal{X}} p(\mathrm{d}x'|s,a)u_t(x')^2 \right| \leq Ln^{-\alpha}\|u_t\|_\infty^2,$$

and following a similar argument as above, $|\sigma_t(x,a) - \sigma_t(s,a)| \leq 4Ln^{-\alpha}\|u_t\|_\infty^2$. $\qquad\square$

We now analyse the empirical estimation error of $\int_{\mathcal{X}} p(\mathrm{d}x'|x,a)(u_k^{(t+1)}(x') - u_{k-1}^{(t+1)}(x'))$ from Line 15 in Algorithm 2.

**Lemma 12.** *Let $\mathcal{X}$ be a compact set with $\frac{1}{n}$-net $\mathcal{S}_n$ and $\mathcal{A}$ a finite set. For each $(s, a) \in \mathcal{S}_n \times \mathcal{A}$, consider samples $x_{s,a}^{(1)}, \ldots, x_{s,a}^{(\ell)} \in \mathcal{X}$ from the distribution $p(\cdot|s,a)$, where $\ell := \lceil 2^9 H^2 \ln \frac{2H|\mathcal{S}_n|A}{\delta} \rceil$. Let $u, v \in \mathcal{B}(\mathcal{X})$ such that $\|u - v\|_\infty \leq \epsilon$ for $\epsilon \geq 0$. Let the empirical estimates*

$$\forall s \in \mathcal{S}_n, x \in \mathcal{X}(s), a \in \mathcal{A}: \qquad \widehat{\beta}(x,a) := \widehat{\beta}(s,a) := \frac{1}{\ell}\sum_{i=1}^\ell u(x_{s,a}^{(i)}) - v(x_{s,a}^{(i)}) - \frac{\epsilon}{16H} - Ln^{-\alpha}\epsilon.$$

---

**Algorithm 2** Classical approximate backward induction algorithm

---

**Input:** Compact state space $\mathcal{X}$ with $\frac{1}{n}$-net $\mathcal{S}_n$, finite action space $\mathcal{A}$, horizon $H$, classical sampling access to probability kernels $p$, failure probability $\delta \in (0,1)$, error $\varepsilon > 0$.

**Output:** $(\varepsilon + 12Ln^{-\alpha}H^2)$-optimal policy $\pi^{(K)} \in \Pi^{\mathrm{D}}$.

1: Initialise $u_t^{(0)} \equiv 0$ for all $t \in [H]$ and $\pi^{(0)} \in \Pi^{\mathrm{D}}$ arbitrary
2: **for** $k = 1$ to $K := \lceil \log_2(H/\varepsilon) \rceil$ **do**
3:      $\epsilon_k \leftarrow H/2^k$
4:      $m_k \leftarrow \lceil \frac{128H^3}{\min\{\epsilon_k^2,1\}} \ln \frac{16H|\mathcal{S}_n|AK}{\delta} \rceil$, $\ell_k \leftarrow \lceil 512H^2 \ln \frac{4H|\mathcal{S}_n|AK}{\delta} \rceil$, $\theta_k \leftarrow \frac{1}{m_k} \ln \frac{16H|\mathcal{S}_n|AK}{\delta}$
5:      For each $(s,a) \in \mathcal{S}_n \times \mathcal{A}$, sample $x_{s,a}^{(1)}, x_{s,a}^{(2)}, \ldots, x_{s,a}^{(m_k)} \in \mathcal{X}$
6:      **for** $(s,a,t) \in \mathcal{S}_n \times \mathcal{A} \times [H]$ **do**
7:          $\widetilde{\sigma}_t^{(k)}(s,a) \leftarrow \frac{1}{m_k} \sum_{i=1}^{m_k} (u_t^{(k-1)}(x_{s,a}^{(i)}))^2 - \left( \frac{1}{m_k} \sum_{i=1}^{m_k} u_t^{(k)}(x_{s,a}^{(i)}) \right)^2$
8:          $\widehat{\mu}_t^{(k)}(s,a) \leftarrow \frac{1}{m_k} \sum_{i=1}^{m_k} u_t^{(k-1)}(x_{s,a}^{(i)}) - \sqrt{2\theta_k \widetilde{\sigma}_t^{(k)}(s,a)} - (\frac{2}{3}\theta_k + 2(2\theta_k)^{3/4} + Ln^{-\alpha})H$
9:      **end for**
10:     $\widehat{\mu}_t^{(k)}(x,a) \leftarrow \widehat{\mu}_t^{(k)}(s,a)$ for all $s \in \mathcal{S}_n$, $x \in \mathcal{X}(s)$, $a \in \mathcal{A}$, and $t \in [H]$
11:     $u_H^{(k)}(x) \leftarrow u_H^{(k-1)}(x)$ for all $x \in \mathcal{X}$ and $\pi_H^{(k)} \leftarrow \pi_H^{(k-1)}$
12:     **for** $t = H-1, H-2, \ldots, 1$ **do**
13:        **for** $(s,a) \in \mathcal{S}_n \times \mathcal{A}$ **do**
14:          Sample $\bar{x}_{s,a}^{(1)}, \bar{x}_{s,a}^{(2)}, \ldots, \bar{x}_{s,a}^{(\ell_k)} \in \mathcal{X}$
15:          $\widehat{\beta}_{t+1}^{(k)}(s,a) \leftarrow \frac{1}{\ell_k} \sum_{i=1}^{\ell_k} \left( u_{t+1}^{(k)}(\bar{x}_{s,a}^{(i)}) - u_{t+1}^{(k-1)}(\bar{x}_{s,a}^{(i)}) \right) - \frac{\epsilon_k}{4H} - \frac{3}{2}Ln^{-\alpha}H$
16:        **end for**
17:        **for** $s \in \mathcal{S}_n$ **do**
18:          $u_t^{(k)}(s) \leftarrow \max_{a \in \mathcal{A}} \{ r(s,a) + \widehat{\mu}_{t+1}^{(k)}(s,a) + \widehat{\beta}_{t+1}^{(k)}(s,a) \} - Ln^{-\alpha}$
19:          $\pi_t^{(k)}(s) \leftarrow \arg\max_{a \in \mathcal{A}} \{ r(s,a) + \widehat{\mu}_{t+1}^{(k)}(s,a) + \widehat{\beta}_{t+1}^{(k)}(s,a) \}$
20:          If $u_t^{(k)}(s) \leq u_t^{(k-1)}(s)$, then $u_t^{(k)}(s) \leftarrow u_t^{(k-1)}(s)$ and $\pi_t^{(k)}(s) \leftarrow \pi_t^{(k-1)}(s) \ \forall x \in \mathcal{X}(s)$
21:        **end for**
22:      **end for**
23: **end for**
24: **return** $\pi^{(K)} \in \Pi^{\mathrm{D}}$

---

Then, with probability at least $1 - \frac{\delta}{H}$, for all $(x,a) \in \mathcal{X} \times \mathcal{A}$,

$$\int_{\mathcal{X}} p(\mathrm{d}x'|x,a)(u(x') - v(x')) - \frac{\epsilon}{8H} - 2Ln^{-\alpha}\epsilon \leq \widehat{\beta}(x,a) \leq \int_{\mathcal{X}} p(\mathrm{d}x'|x,a)(u(x') - v(x')).$$

*Proof.* By a Hoeffding's inequality (Fact 7), Assumption 10, and a union bound over $\mathcal{S}_n \times \mathcal{A}$, with probability at least $1 - \frac{\delta}{H}$, for all $s \in \mathcal{S}_n$, $x \in \mathcal{X}(s)$, and $a \in \mathcal{A}$,

$$\left| \frac{1}{\ell} \sum_{i=1}^{\ell} u(x_{s,a}^{(i)}) - v(x_{s,a}^{(i)}) - \int_{\mathcal{X}} p(\mathrm{d}x'|x,a)(u(x') - v(x')) \right| \leq \left( \sqrt{\frac{2}{\ell} \ln \frac{2H|\mathcal{S}_n|A}{\delta}} + Ln^{-\alpha} \right) \|u - v\|_{\infty}$$

$$\leq \frac{\epsilon}{16H} + Ln^{-\alpha}\epsilon.$$

By shifting the estimate to have one-sided error, we obtain the one-sided error $\frac{\epsilon}{8H} + 2Ln^{-\alpha}\epsilon$. $\qquad\square$

We are almost ready to prove the correctness of Algorithm 2 and its complexity. Before that, we state a useful upper bound on the variance.

**Fact 13** ((Sidford et al., 2018a, Lemma F.4)). *Let* $\sigma_t^{\pi}(x,a) := \int_{\mathcal{X}} p(\mathrm{d}x'|x,a)V_t^{\pi}(x')^2 - \left( \int_{\mathcal{X}} p(\mathrm{d}x'|x,a)V_t^{\pi}(x') \right)^2$ *for a given policy* $\pi$. *For any policy* $\pi$ *and* $(x,t) \in \mathcal{X} \times [H]$,[11]

$$\sum_{t'=t}^{H-1} \int \cdots \int_{\mathcal{X}^{t'-t}} p_{\pi_t}(\mathrm{d}x_{t+1}|x)p_{\pi_{t+1}}(\mathrm{d}x_{t+2}|x_{t+1}) \cdots p_{\pi_{t'-1}}(\mathrm{d}x_{t'}|x_{t'-1})\sqrt{\sigma_{t'+1}^{\pi}(x_{t'}, \pi_{t'}(x_{t'}))} \leq H^{\frac{3}{2}},$$

---

[11]We note the typo in (Sidford et al., 2018a, Lemma F.4) where $\| \cdot \|_{\infty}^2$ should be $\| \cdot \|_{\infty}$.

where $\sigma_{t+1}^\pi(x_t, \pi_t(x_t)) = \sigma_{t+1}^\pi(x, \pi_t(x))$ for $t' = t$.

**Theorem 14** (Classical finite-horizon generative algorithm). *Let $M = \langle \mathcal{X}, \mathcal{A}, H, p, r \rangle$ be a finite-horizon MDP and $\mathcal{S}_n$ a $\frac{1}{n}$-net for $\mathcal{X}$. Let $K := \lceil \log_2(H/\varepsilon) \rceil$ for a given $\varepsilon > 0$ and $\epsilon_k := H/2^k$ for all $k \in [K]$. Under Assumption 10 with $Ln^{-\alpha} \le \frac{1}{16H}$, Algorithm 2 computes functions $\{u_t^{(k)} \in \mathcal{B}(\mathcal{X})\}_{t \in [H], k \in [K]}$ and policies $\{\pi^{(k)} \in \Pi^\mathrm{D}\}_{k \in [K]}$ such that, with probability at least $1 - \delta$,*

$$V_t^*(x) - \epsilon_k - 12Ln^{-\alpha}H^2 \le u_t^{(k)}(x) \le V_t^{\pi^{(k)}}(x) \le V_t^*(x) \qquad \forall(x, t, k) \in \mathcal{X} \times [H] \times [K].$$

*In particular, $\pi^{(K)} \in \Pi^\mathrm{D}$ is such that $V_1^*(x) - \varepsilon - 12Ln^{-\alpha}H^2 \le V_1^{\pi^{(K)}}(x) \le V_1^*(x)$ for all $x \in \mathcal{X}$. The $\mathcal{C}_p$-query complexity is*

$$O\left( \frac{H^3|\mathcal{S}_n|A}{\varepsilon^2} \log\left( \frac{H|\mathcal{S}_n|A}{\delta} \log \frac{H}{\varepsilon} \right) \right).$$

*Proof.* The proof is by induction on $k = 0, 1, \ldots, K$. The base case $k = 0$ is trivial since $\epsilon_0 = H$, $u_t^{(0)} \equiv 0$, and $V_t^*(x) \le H + 1 - t$ for all $t \in [H]$. Assume then that Algorithm 2 has properly computed functions and policies such that

$$V_t^*(x) - \epsilon_{k'} - 12Ln^{-\alpha}H^2 \le u_t^{(k')}(x) \le (\mathcal{L}_{\pi_t^{(k')}} u_{t+1}^{(k')})(x) \le V_t^*(x) \quad \forall k' \in [k-1], t \in [H], \quad (3)$$

and consider now epoch $k \in [K]$. We first analyse the approximate quantities $\widehat{\mu}_t^{(k)}$ and $\widetilde{\sigma}_t^{(k)}$ from Lines 7 and 8. Define $\widetilde{\mu}_t^{(k)}(s, a) := \frac{1}{m_k} \sum_{i=1}^{m_k} u_t^{(k-1)}(x_{s,a}^{(i)})$ for all $(s, a, t) \in \mathcal{S}_n \times \mathcal{A} \times [H]$. Define $\mu_t^{(k)}(x, a) := \int_\mathcal{X} p(\mathrm{d}x'|x, a) u_t^{(k-1)}(x')$ and $\sigma_t^{(k)}(x, a) := \int_\mathcal{X} p(\mathrm{d}x'|x, a) u_t^{(k-1)}(x')^2 - \left( \int_\mathcal{X} p(\mathrm{d}x'|x, a) u_t^{(k-1)}(x') \right)^2$. Then, according to Lemma 11 and already using that $\|u_t^{(k-1)}\|_\infty \le \|V_t^*\|_\infty \le H$ by the induction hypothesis, with probability at least $1 - \frac{\delta}{2K}$, for all $(s, a, t) \in \mathcal{S}_n \times \mathcal{A} \times [H]$, (let $\theta_k := \frac{1}{m_k} \ln \frac{16H|\mathcal{S}_n|AK}{\delta} \le \frac{1}{128H^3} \min\{\epsilon_k^2, 1\}$)

$$|\widetilde{\mu}_t^{(k)}(s, a) - \mu_t^{(k)}(s, a)| \le \sqrt{2\theta_k \sigma_t^{(k)}(s, a)} + \frac{2}{3}\theta_k H, \tag{4a}$$

$$|\widetilde{\sigma}_t^{(k)}(s, a) - \sigma_t^{(k)}(s, a)| \le 4\sqrt{2\theta_k} H^2, \tag{4b}$$

which we condition on. By using Eq. (4b) onto Eq. (4a), we get that

$$|\widetilde{\mu}_t^{(k)}(s, a) - \mu_t^{(k)}(s, a)| \le \sqrt{2\theta_k \widetilde{\sigma}_t^{(k)}(s, a)} + \left( \frac{2}{3}\theta_k + 2(2\theta_k)^{3/4} \right) H,$$

from which we define $\widehat{\mu}_t^{(k)}(x, a)$, for all $s \in \mathcal{S}_n$, $x \in \mathcal{X}(s)$, $a \in \mathcal{A}$, as

$$\widehat{\mu}_t^{(k)}(x, a) := \widetilde{\mu}_t^{(k)}(s, a) - \sqrt{2\theta_k \widetilde{\sigma}_t^{(k)}(s, a)} - \left( \frac{2}{3}\theta_k + 2(2\theta_k)^{3/4} + Ln^{-\alpha} \right) H.$$

The quantity $\widehat{\mu}_t^{(k)}(x, a)$ has one-sided error: for all $s \in \mathcal{S}_n$, $x \in \mathcal{X}(s)$, $a \in \mathcal{A}$,

$$\mu_t^{(k)}(x, a) \ge \widehat{\mu}_t^{(k)}(x, a) \ge \mu_t^{(k)}(x, a) - 2\sqrt{2\theta_k \widetilde{\sigma}_t^{(k)}(s, a)} - \left( \frac{4}{3}\theta_k + 4(2\theta_k)^{3/4} + 2Ln^{-\alpha} \right) H.$$

We can express the above inequality using the variance $\sigma_t^*(x, a) := \int_\mathcal{X} p(\mathrm{d}x'|x, a) V_t^*(x')^2 - \left( \int_\mathcal{X} p(\mathrm{d}x'|x, a) V_t^*(x') \right)^2$ since, for $s \in \mathcal{S}_n$, $x \in \mathcal{X}(s)$, and $a \in \mathcal{A}$,

$$\sqrt{\widetilde{\sigma}_t^{(k)}(s, a)} \le \sqrt{\sigma_t^{(k)}(s, a)} + 2(2\theta_k)^{1/4} H \qquad \text{(by Eq. (4b))}$$

$$\le \sqrt{\sigma_t^{(k)}(x, a)} + 2(\sqrt{Ln^{-\alpha}} + (2\theta_k)^{1/4}) H$$

$$\text{(by } |\sigma_t^{(k)}(x, a) - \sigma_t^{(k)}(s, a)| \le 4Ln^{-\alpha}H^2)$$

$$\le \sqrt{\sigma_t^*(x, a)} + 2\epsilon_k + 12Ln^{-\alpha}H^2 + 2(\sqrt{Ln^{-\alpha}} + (2\theta_k)^{1/4}) H,$$

where we used that $\mathrm{Var}[u_t^{(k-1)}] \leq \mathrm{Var}[V_t^*] + \mathrm{Var}[V_t^* - u_t^{(k-1)}]$ and that $\mathrm{Var}[V_t^* - u_t^{(k-1)}] \leq (2\epsilon_k + 12Ln^{-\alpha}H^2)^2$ if $\|V_t^* - u_t^{(k-1)}\|_\infty \leq 2\epsilon_k + 12Ln^{-\alpha}H^2$ according to the induction hypothesis. This means that, for all $(x, a, t) \in \mathcal{X} \times \mathcal{A} \times [H]$, with probability at least $1 - \frac{\delta}{2K}$,

$$\widehat{\mu}_t^{(k)}(x, a) \leq \mu_t^{(k)}(x, a), \tag{5a}$$

$$\widehat{\mu}_t^{(k)}(x, a) \geq \mu_t^{(k)}(x, a) - \sqrt{8\theta_k \sigma_t^*(x, a)} - \sqrt{8\theta_k}(2\epsilon_k + 12Ln^{-\alpha}H^2) - \left(\frac{16}{3}\theta_k + 8(2\theta_k)^{3/4} + 4Ln^{-\alpha}\right)H. \tag{5b}$$

Given the above inequalities, which we condition on, we now analyse the backward iteration that happens within epoch $k \in [K]$ when calculating $u_t^{(k)} \in \mathscr{B}(\mathcal{X})$ for $t = H, H-1, \ldots, 1$. First let us focus on proving that $u_t^{(k)}(x) \leq V_t^{\pi^{(k)}}(x) \leq V_t^*(x)$. For such, assume by induction that

$$u_{t'}^{(k)}(x) \leq (\mathcal{L}_{\pi_{t'}^{(k)}} u_{t'+1}^{(k)})(x) \leq V_{t'}^{\pi^{(k)}}(x) \leq V_{t'}^*(x) \qquad \forall x \in \mathcal{X}, t' = t+1, t+2, \ldots, H+1.$$

The case $t = H+1$ is trivial since $V_{H+1}^*, u_{H+1}^{(k)} \equiv 0$. For $t \leq H$, consider, $\forall s \in \mathcal{S}_n, x \in \mathcal{X}(s), a \in \mathcal{A}$, the quantities from Line 15,

$$\widehat{\beta}_{t+1}^{(k)}(x, a) := \frac{1}{\ell_k} \sum_{i=1}^{\ell_k} \left(u_{t+1}^{(k)}(\bar{x}_{s,a}^{(i)}) - u_{t+1}^{(k-1)}(\bar{x}_{s,a}^{(i)})\right) - \frac{\epsilon_k}{4H} - \frac{3}{2}Ln^{-\alpha}H.$$

Note that $u_{t+1}^{(k-1)}(x) \leq u_{t+1}^{(k)}(x)$ holds, since it is enforced by design (Line 20). Due to that and by the induction hypothesis (both in $t$ and $k$), then $\|u_{t+1}^{(k)} - u_{t+1}^{(k-1)}\|_\infty \leq 2\epsilon_k + 12Ln^{-\alpha}H^2$. Moreover, $\left(\frac{1}{16H} + Ln^{-\alpha}\right)(2\epsilon_k + 12Ln^{-\alpha}H^2) \leq \frac{\epsilon_k}{4H} + \frac{3}{2}Ln^{-\alpha}H$ using that $Ln^{-\alpha} \leq \frac{1}{16H}$. Hence, according to Lemma 12 with $\ell_k = \lceil 2^9 H^2 \ln \frac{4H|\mathcal{S}_n|AK}{\delta} \rceil$, with probability at least $1 - \frac{\delta}{2KH}$, for all $(x, a) \in \mathcal{X} \times \mathcal{A}$,

$$\widehat{\beta}_{t+1}^{(k)}(x, a) \leq \int_{\mathcal{X}} p(\mathrm{d}x'|x, a)\left(u_t^{(k)}(x') - u_{t+1}^{(k-1)}(x')\right), \tag{6a}$$

$$\widehat{\beta}_{t+1}^{(k)}(x, a) \geq \int_{\mathcal{X}} p(\mathrm{d}x'|x, a)\left(u_{t+1}^{(k)}(x') - u_{t+1}^{(k-1)}(x')\right) - \frac{\epsilon_k}{2H} - 3Ln^{-\alpha}H. \tag{6b}$$

Conditioned on the above, there are two cases to consider. **Case 1:** $\pi_t^{(k)} \neq \pi_t^{(k-1)}$. If $\pi_t^{(k)}(s) \neq \pi_t^{(k-1)}(s)$ for some $s \in \mathcal{S}_n$, then it means that, for all $x \in \mathcal{X}(s)$,

$$\begin{aligned} u_t^{(k)}(x) := u_t^{(k)}(s) &= r(s, \pi_t^{(k)}(s)) - Ln^{-\alpha} + \widehat{\mu}_{t+1}^{(k)}(s, \pi_t^{(k)}(s)) + \widehat{\beta}_{t+1}^{(k)}(s, \pi_t^{(k)}(s)) \\ &\leq r(x, \pi_t^{(k)}(x)) + \widehat{\mu}_{t+1}^{(k)}(x, \pi_t^{(k)}(x)) + \widehat{\beta}_{t+1}^{(k)}(x, \pi_t^{(k)}(x)) \\ &\leq r(x, \pi_t^{(k)}(x)) + \int_{\mathcal{X}} p(\mathrm{d}x'|x, \pi_t^{(k)}(x))u_{t+1}^{(k)}(x') \qquad \text{(by Eqs. (5a) and (6a))} \\ &= (\mathcal{L}_{\pi_t^{(k)}} u_{t+1}^{(k)})(x). \end{aligned}$$

**Case 2:** $\pi_t^{(k)} = \pi_t^{(k-1)}$. If both policies are equal, then, for all $x \in \mathcal{X}$,

$$u_t^{(k)}(x) = u_t^{(k-1)}(x) \leq (\mathcal{L}_{\pi_t^{(k-1)}} u_{t+1}^{(k-1)})(x) \leq (\mathcal{L}_{\pi_t^{(k-1)}} u_{t+1}^{(k)})(x) = (\mathcal{L}_{\pi_t^{(k)}} u_{t+1}^{(k)})(x),$$

where we used the induction hypothesis to argue that $u_t^{(k-1)}(x) \leq (\mathcal{L}_{\pi_t^{(k-1)}} u_{t+1}^{(k-1)})(x)$. From the above, we can readily see that $u_t^{(k)}(x) \leq V_t^*(x)$, which completes the induction on $t$.

We now move on to proving that $u_t^{(k)}(x) \geq V_t^*(x) - \epsilon_k - 12Ln^{-\alpha}H^2$ (it is implicitly assumed that all $u_{t'+1}^{(k)}$ for $t' = t, \ldots, H-1$ have already been computed by the backward iteration). For such,

first note that

$$V_t^*(x) - u_t^{(k)}(x) = (\mathcal{L}V_{t+1}^*)(x) - u_t^{(k)}(x)$$

$$\leq \max_{a \in \mathcal{A}} \left\{ r(x,a) + \int_{\mathcal{X}} p(\mathrm{d}x'|x,a)V_{t+1}^*(x') \right\}$$

$$- \max_{a \in \mathcal{A}} \left\{ r(x,a) - 2Ln^{-\alpha} + \widehat{\mu}_{t+1}^{(k)}(x,a) + \widehat{\beta}_{t+1}^{(k)}(x,a) \right\}$$

$$(u_t^{(k)}(x) \geq u_t^{(k)}(s) - Ln^{-\alpha})$$

$$\leq \int_{\mathcal{X}} p(\mathrm{d}x'|x,\pi_t^*(x))\left(V_{t+1}^*(x') - u_{t+1}^{(k)}(x')\right) + \xi_{t+1}^{(k)}(x),$$

(by Eqs. (5b) and (6b))

where we defined

$$\xi_{t+1}^{(k)}(x) := \sqrt{8\theta_k \sigma_{t+1}^*(x,\pi_t^*(x))} + \frac{\epsilon_k}{2H} + \sqrt{8\theta_k}(2\epsilon_k + 12Ln^{-\alpha}H^2) + \left(\frac{16}{3}\theta_k + 8(2\theta_k)^{3/4} + 9Ln^{-\alpha}\right)H.$$

Solving the recursion, we obtain that

$$V_t^*(x) - u_t^{(k)}(x) \leq \sum_{t'=t}^{H-1} \int \cdots \int_{\mathcal{X}^{t'-t}} p_{\pi_t^*}(\mathrm{d}x_{t+1}|x)p_{\pi_{t+1}^*}(\mathrm{d}x_{t+2}|x_{t+1}) \cdots p_{\pi_{t'-1}^*}(\mathrm{d}x_{t'}|x_{t'-1})\xi_{t'+1}^{(k)}(x_{t'}).$$

By employing that $\| \sum_{t'=t}^{H-1} \int \cdots \int_{\mathcal{X}^{t'-t}} p_{\pi_t^*}(\mathrm{d}x_{t+1}|x) \cdots p_{\pi_{t'-1}^*}(\mathrm{d}x_{t'}|x_{t'-1})\|_\infty \leq H - t + 1$ and

$$\left\| \sum_{t'=t}^{H-1} \int \cdots \int_{\mathcal{X}^{t'-t}} p_{\pi_t^*}(\mathrm{d}x_{t+1}|x) \cdots p_{\pi_{t'-1}^*}(\mathrm{d}x_{t'}|x_{t'-1})\sqrt{\sigma_{t'+1}^*(x_{t'},\pi_{t'}^*(x_{t'}))} \right\|_\infty \leq H^{3/2},$$

(by Fact 13)

we finally obtain that

$$V_t^*(x) - u_t^{(k)}(x) \leq \frac{\epsilon_k}{2} + 2\sqrt{2\theta_k}H^{3/2} + 4\sqrt{2\theta_k}\epsilon_k H + \left(\frac{16}{3}\theta_k + 8(2\theta_k)^{3/4}\right)H^2$$

$$+ 24\sqrt{2\theta_k}Ln^{-\alpha}H^3 + 9Ln^{-\alpha}H^2$$

$$\leq \epsilon_k \left(\frac{1}{2} + \frac{2\sqrt{2}}{\sqrt{128}} + \frac{4\sqrt{2}}{\sqrt{128}} + \frac{16}{3 \cdot 128} + \frac{8 \cdot 2^{3/4}}{128^{3/4}}\right)$$

$$+ \frac{24\sqrt{2}}{\sqrt{128}}Ln^{-\alpha}H^{3/2} + 9Ln^{-\alpha}H^2 \qquad (\theta_k \leq \frac{1}{128H^3}\min\{\epsilon_k^2, 1\})$$

$$< \epsilon_k + 12Ln^{-\alpha}H^2,$$

This concludes the proof of Eq. (3) for epoch $k \in [K]$ and thus for all epochs by induction.

Regarding the failure probability, Eq. (5) holds with probability $1 - \frac{\delta}{2K}$ for a single epoch, while Eq. (6) holds with probability $1 - \frac{\delta}{2HK}$ for a single epoch and time step. Across all epochs and time steps, the success probability is at least $1 - \delta$, as required. Finally, the total number of samples is

$$O\left(\sum_{k=1}^{K}(m_k + H\ell_k)|\mathcal{S}_n|A\right) = O\left(\frac{H^3|\mathcal{S}_n|A}{\varepsilon^2}\log\left(\frac{H|\mathcal{S}_n|A}{\delta}\log\frac{H}{\varepsilon}\right)\right). \qquad \square$$

## C.2 QUANTUM BACKWARD INDUCTION ALGORITHM

In this section, we propose two quantum algorithms that output an $\varepsilon$-optimal policy given sampling access to the probability kernel via the oracle $\mathcal{O}_p$. The first quantum algorithm is analogous to the classical approximate backward induction algorithm from the previous section while the second quantum algorithm is based on a much simpler backward induction algorithm. For both algorithms, the main idea is to estimate $\int_{\mathcal{X}} p(\mathrm{d}x'|x,a)u_t^{(k-1)}(x')$ and $\int_{\mathcal{X}} p(\mathrm{d}x'|x,a)\left(u_t^{(k)}(x') - u_t^{(k-1)}(x')\right)$ via quantum mean estimation subroutines (Cornelissen et al., 2022; Kothari & O'Donnell, 2023).

Wang et al. (2021) considered the case of infinite-horizon *discounted* MDPs with a generative model, which shares similarities with the finite-horizon case. The following analysis thus takes inspiration from the work of Wang et al. (2021), who also employed quantum mean estimation subroutines to speed-up the corresponding classical value iteration algorithm for infinite-horizon discounted MDPs from Sidford et al. (2018a). Still, there are a few subtle differences, e.g., requiring the estimation of $H$ different means $\int_{\mathcal{X}} p(\mathrm{d}x'|x,a)u_t^{(k-1)}(x')$ at the beginning of each epoch instead of just one as for infinite-horizon discounted MDPs. To get around this issue, we employ the quantum multivariate mean estimation subroutine from Cornelissen et al. (2022) to estimate these $H$ average quantities at once with only an $O(\sqrt{H})$ overhead. This is, however, the best one can hope for: there is no concept of "reusing" quantum samples like in classical algorithms, which ultimately hinders a full quadratic speed-up in terms of $H$ compared to the classical complexity. The quantities $\int_{\mathcal{X}} p(\mathrm{d}x'|x,a)\big(u_t^{(k)}(x') - u_t^{(k-1)}(x')\big)$ during the backward recursion are approximated with the quantum mean estimation of Kothari & O'Donnell (2023).

Our first quantum algorithm (Algorithm 3) is based on the modern backward induction algorithm from the previous section and relies on monotonicity, variance reduction, and total-variance techniques. Its overall correctness proof is very similar to the one for Algorithm 2, hence we skip parts in which both proofs are virtually the same. We shall need a slightly different version of Fact 13.

**Lemma 15.** *Let $\sigma_t^\pi(x,a) := \int_{\mathcal{X}} p(\mathrm{d}x'|x,a)V_t^\pi(x')^2 - \big(\int_{\mathcal{X}} p(\mathrm{d}x'|x,a)V_t^\pi(x')\big)^2$ for some policy $\pi$. For any policy $\pi$ and $(x,t) \in \mathcal{X} \times [H]$,*

$$\sum_{k=1}^{H}\sum_{t'=t}^{H-1} \int \cdots \int_{\mathcal{X}^{t'-t}} p_{\pi_t}(\mathrm{d}x_{t+1}|x)p_{\pi_{t+1}}(\mathrm{d}x_{t+2}|x_{t+1})\cdots p_{\pi_{t'-1}}(\mathrm{d}x_{t'}|x_{t'-1})\sigma_k^\pi(x_{t'},\pi_{t'}(x_{t'})) \leq 4H^3,$$

*where $\sigma_k^\pi(x_t,\pi_t(x_t)) = \sigma_k^\pi(x,\pi_t(x))$ for $t' = t$.*

*Proof.* Given $u \in \mathscr{B}(\mathcal{X})$, define $P_t^\pi u \in \mathscr{B}(\mathcal{X})$ as $(P_t^\pi u)(x) = \int_{\mathcal{X}} p_{\pi_t}(\mathrm{d}x'|x)u(x')$ for simplicity. Then $V_t^\pi = r_t^\pi + P_t^\pi V_{t+1}^\pi$ for $t \in [H]$ and $V_{H+1}^\pi \equiv 0$, where $r_t^\pi(x) = r(x,\pi_t(x))$. Moreover, write $\mathrm{Var}_{P_t^\pi}(V_{t'}^\pi) = \sigma_{t'}^\pi(x,\pi_t(x)) = \int_{\mathcal{X}} p(\mathrm{d}x'|x,\pi_t(x))V_{t'}^\pi(x')^2 - \big(\int_{\mathcal{X}} p(\mathrm{d}x'|x,\pi_t(x))V_{t'}^\pi(x')\big)^2$. If we let $(u \circ v)(x) = u(x)v(x)$ denote the Hadamard product for $u,v \in \mathscr{B}(\mathcal{X})$, then we can write

$$\mathrm{Var}_{P_t^\pi}(V_{k+1}^\pi) = P_t^\pi(V_{k+1}^\pi \circ V_{k+1}^\pi) - (P_t^\pi V_{k+1}^\pi) \circ (P_t^\pi V_{k+1}^\pi)$$
$$= P_t^\pi(V_{k+1}^\pi \circ V_{k+1}^\pi) - (V_k^\pi - r_k^\pi) \circ (V_k^\pi - r_k^\pi) \leq P_t^\pi(V_{k+1}^\pi \circ V_{k+1}^\pi) - V_k^\pi \circ V_k^\pi + 2V_k^\pi \circ r_k^\pi.$$

Notice that the left-hand side of our sought-after statement can be rewritten more compactly as $\sum_{k=1}^{H}\sum_{j=t}^{H-1}\prod_{i=t}^{j-1}P_i^\pi \mathrm{Var}_{P_j^\pi}(V_k^\pi)$. We bound it as

$$\sum_{k=1}^{H}\sum_{j=t}^{H-1}\prod_{i=t}^{j-1}P_i^\pi \mathrm{Var}_{P_j^\pi}(V_k^\pi)$$

$$\leq \sum_{j=t}^{H-1}\prod_{i=t}^{j-1}P_i^\pi \mathrm{Var}_{P_j^\pi}(V_1^\pi) + \sum_{k=1}^{H-1}\sum_{j=t}^{H-1}\prod_{i=t}^{j-1}P_i^\pi\big(P_j^\pi(V_{k+1}^\pi \circ V_{k+1}^\pi) - V_k^\pi \circ V_k^\pi + 2V_k^\pi \circ r_k^\pi\big)$$

$$\leq 3H^3 + \sum_{k=1}^{H-1}\sum_{j=t}^{H-1}\left(\prod_{i=t}^{j}P_i^\pi(V_{k+1}^\pi \circ V_{k+1}^\pi) - \prod_{i=t}^{j-1}P_i^\pi(V_k^\pi \circ V_k^\pi)\right)$$

$$\text{(since } \|\mathrm{Var}_{P_j^\pi}(V_1^\pi)\|_\infty \leq H^2, \|V_k^\pi\|_\infty \leq H, \|\textstyle\sum_{k=1}^{H-1}r_k^\pi\|_\infty \leq H)$$

$$= 3H^3 + \sum_{k=2}^{H}\sum_{j=t}^{H-1}\prod_{i=t}^{j}P_i^\pi(V_k^\pi \circ V_k^\pi) - \sum_{k=1}^{H-1}\sum_{j=t}^{H-2}\prod_{i=t}^{j}P_i^\pi(V_k^\pi \circ V_k^\pi) - \sum_{k=1}^{H-1}(V_k^\pi \circ V_k^\pi)$$

$$= 3H^3 + \sum_{k=2}^{H}\prod_{i=t}^{H-1}P_i^\pi(V_k^\pi \circ V_k^\pi) + \sum_{j=t}^{H-2}\prod_{i=t}^{j}P_i^\pi(V_H^\pi \circ V_H^\pi) - \sum_{j=t}^{H-2}\prod_{i=t}^{j}P_i^\pi(V_1^\pi \circ V_1^\pi) - \sum_{k=1}^{H-1}(V_k^\pi \circ V_k^\pi)$$

$$\leq 3H^3 + \sum_{j=t}^{H-1}\prod_{i=t}^{j}P_i^\pi(V_H^\pi \circ V_H^\pi) - \sum_{j=t}^{H-2}\prod_{i=t}^{j}P_i^\pi(V_1^\pi \circ V_1^\pi) \qquad (\|\textstyle\prod_{i=t}^{H-1}P_i^\pi\|_1 \leq 1)$$

$$\leq 4H^3. \qquad\qquad\qquad\qquad\qquad\qquad\qquad\qquad (\|V_H^\pi\|_\infty \leq 1) \quad \square$$

---

**Algorithm 3** Modern quantum backward induction algorithm

---

**Input:** Compact state space $\mathcal{X}$ with $\frac{1}{n}$-net $\mathcal{S}_n$, finite action space $\mathcal{A}$, horizon $H$, quantum sampling access to probability kernels $p$, failure probability $\delta \in (0,1)$, error $\varepsilon > 0$.

**Output:** $(\varepsilon + 8Ln^{-\alpha}H^2)$-optimal policy $\pi^{(K)} \in \Pi^{\mathrm{D}}$.

1: Initialise $u_t^{(0)} \equiv 0$ for all $t \in [H]$ and $\pi^{(0)} \in \Pi^{\mathrm{D}}$ arbitrary
2: **for** $k = 1$ to $K := \lceil \log_2(H/\varepsilon) \rceil$ **do**
3: $\quad$ $\epsilon_k \leftarrow H/2^k$ and $\theta_k \leftarrow \frac{\min\{\epsilon_k, 1\}}{20H^{3/2}}$
4: $\quad$ **for** $(s,a,t) \in \mathcal{S}_n \times \mathcal{A} \times [H]$ **do**
5: $\quad\quad$ Obtain $\widetilde{\sigma}_t^{(k)}(s,a)$ such that $\left| \widetilde{\sigma}_t^{(k)}(s,a) - \sigma_t^{(k)}(s,a) \right| \leq \theta_k H^2$ by using Fact 6, where
$\quad\quad$ $\sigma_t^{(k)}(s,a) := \int_{\mathcal{X}} p(\mathrm{d}x'|s,a) u_t^{(k-1)}(x')^2 - \left( \int_{\mathcal{X}} p(\mathrm{d}x'|s,a) u_t^{(k-1)}(x') \right)^2$
6: $\quad\quad$ Obtain $\widetilde{\mu}_t^{(k)}(s,a)$ such that $\left| \widetilde{\mu}_t^{(k)}(s,a) - \mu_t^{(k)}(s,a) \right| \leq \theta_k \sqrt{\frac{1}{H}\sum_{t'=1}^H \sigma_{t'}^{(k)}(s,a)}$ by using
$\quad\quad$ Fact 6, where $\mu_t^{(k)}(s,a) := \int_{\mathcal{X}} p(\mathrm{d}x'|s,a) u_t^{(k-1)}(x')$
7: $\quad$ **end for**
8: $\quad$ $\widehat{\mu}_t^{(k)}(x,a) \leftarrow \widetilde{\mu}_t^{(k)}(s,a) - \theta_k \sqrt{\frac{1}{H}\sum_{t'=1}^H \widetilde{\sigma}_{t'}^{(k)}(s,a)} - (\theta_k^{3/2} + Ln^{-\alpha})H$ for all $(s,a,t) \in$
$\quad$ $\mathcal{S}_n \times \mathcal{A} \times [H]$, $x \in \mathcal{X}(s)$
9: $\quad$ $u_H^{(k)}(x) \leftarrow u_H^{(k-1)}(x)$ for all $x \in \mathcal{X}$ and $\pi_H^{(k)} \leftarrow \pi_H^{(k-1)}$
10: $\quad$ **for** $t = H-1, H-2, \ldots, 1$ **do**
11: $\quad\quad$ **for** $(s,a) \in \mathcal{S}_n \times \mathcal{A}$ **do**
12: $\quad\quad\quad$ Use Corollary 5 to get $\widetilde{\beta}_{t+1}^{(k)}(s,a)$ such that $\left| \widetilde{\beta}_{t+1}^{(k)}(s,a) - \beta_{t+1}^{(k)}(s,a) \right| \leq \frac{\epsilon_k}{4H} - Ln^{-\alpha}H$,
$\quad\quad\quad$ where $\beta_{t+1}^{(k)}(s,a) := \int_{\mathcal{X}} p(\mathrm{d}x'|s,a)\left(u_{t+1}^{(k)}(x') - u_{t+1}^{(k-1)}(x')\right)$
13: $\quad\quad\quad$ $\widehat{\beta}_{t+1}^{(k)}(s,a) \leftarrow \widetilde{\beta}_{t+1}^{(k)}(s,a) - \frac{\epsilon_k}{4H} - Ln^{-\alpha}H$
14: $\quad\quad$ **end for**
15: $\quad\quad$ **for** $s \in \mathcal{S}_n$ **do**
16: $\quad\quad\quad$ $u_t^{(k)}(s) \leftarrow \max_{a \in \mathcal{A}}\{r(s,a) + \widehat{\mu}_{t+1}^{(k)}(s,a) + \widehat{\beta}_{t+1}^{(k)}(s,a)\} - Ln^{-\alpha}$
17: $\quad\quad\quad$ $\pi_t^{(k)}(s) \leftarrow \arg\max_{a \in \mathcal{A}}\{r(s,a) + \widehat{\mu}_{t+1}^{(k)}(s,a) + \widehat{\beta}_{t+1}^{(k)}(s,a)\}$
18: $\quad\quad\quad$ If $u_t^{(k)}(s) \leq u_t^{(k-1)}(s)$, then $u_t^{(k)}(x) \leftarrow u_t^{(k-1)}(x)$ and $\pi_t^{(k)}(x) \leftarrow \pi_t^{(k-1)}(x)$ $\forall x \in \mathcal{X}(s)$
19: $\quad\quad$ **end for**
20: $\quad$ **end for**
21: **end for**
22: **return** $\pi^{(K)} \in \Pi^{\mathrm{D}}$

---

**Theorem 16** (Modern quantum finite-horizon generative algorithm)**.** *Let $M = \langle \mathcal{X}, \mathcal{A}, H, p, r \rangle$ be a finite-horizon MDP and let $\mathcal{S}_n$ be a $\frac{1}{n}$-net for $\mathcal{X}$. Let $K := \lceil \log_2(H/\varepsilon) \rceil$ for a given $\varepsilon > 0$ and $\epsilon_k := H/2^k$ for all $k \in [K]$. Under Assumption 10 with $Ln^{-\alpha} \leq \frac{1}{16H}$, Algorithm 3 computes functions $\{u_t^{(k)} \in \mathscr{B}(\mathcal{X})\}_{t \in [H], k \in [K]}$ and policies $\{\pi^{(k)} \in \Pi^{\mathrm{D}}\}_{k \in [K]}$ such that, with probability $1 - \delta$,*

$$V_t^*(x) - \epsilon_k - 8Ln^{-\alpha}H^2 \leq u_t^{(k)}(x) \leq V_t^{\pi^{(k)}}(x) \leq V_t^*(x) \qquad \forall(x,t,k) \in \mathcal{X} \times [H] \times [K].$$

*In particular, $\pi^{(K)} \in \Pi^{\mathrm{D}}$ is such that $V_1^*(x) - \varepsilon - 8Ln^{-\alpha}H^2 \leq V_1^{\pi^{(K)}}(x) \leq V_1^*(x)$ for all $x \in \mathcal{X}$. The $\mathcal{O}_p, \mathcal{O}_p^\dagger$-query complexity is*

$$O\left( \frac{H^2|\mathcal{S}_n|A}{\varepsilon} \log\left( \frac{H|\mathcal{S}_n|A}{\delta} \log\frac{H}{\varepsilon} \right) \right).$$

*Proof.* Again the proof is by induction on the epoch $k = 0, 1, \ldots, K$, the base case $k = 0$ being trivial. Assume then that Algorithm 3 has properly computed functions and policies such that

$$V_t^*(x) - \epsilon_{k'} - 8Ln^{-\alpha}H^b \leq u_t^{(k')}(x) \leq (\mathcal{L}_{\pi_t^{(k')}} u_{t+1}^{(k')})(x) \leq V_t^*(x) \quad \forall k' \in [k-1], t \in [H], \quad (7)$$

and consider epoch $k \in [K]$. Let $\theta_k := \frac{1}{20H^{3/2}} \min\{\epsilon_k, 1\}$. We start the analysis with the quantities $\widetilde{\mu}_t^{(k)}$ and $\widetilde{\sigma}_t^{(k)}$ from Lines 5 and 6. Define once again the true quantities $\mu_t^{(k)}(x,a) := \int_{\mathcal{X}} p(\mathrm{d}x'|x,a) u_t^{(k-1)}(x')$ and $\sigma_t^{(k)}(x,a) := \int_{\mathcal{X}} p(\mathrm{d}x'|x,a) u_t^{(k-1)}(x')^2 -$

$\left( \int_{\mathcal{X}} p(\mathrm{d}x'|x,a) u_t^{(k-1)}(x') \right)^2$. For all $(s,a) \in \mathcal{S}_n \times \mathcal{A}$, we employ the quantum multivariate mean estimation from Fact 6 with

$$m_k := O\left( \frac{\sqrt{H}}{\theta_k} \log \frac{H|\mathcal{S}_n|AK}{\delta} \right) = O\left( \frac{H^2}{\min\{\epsilon_k, 1\}} \log \frac{H|\mathcal{S}_n|AK}{\delta} \right)$$

queries to $\mathcal{O}_p$ to obtain $(\widetilde{\sigma}_1^{(k)}(s,a), \ldots, \widetilde{\sigma}_H^{(k)}(s,a)) \in \mathbb{R}^H$ (the vector $(\sigma_1^{(k)}(s,a), \ldots, \sigma_H^{(k)}(s,a))$ has standard deviation at most $H^{5/2}$). Likewise, for all $(s,a) \in \mathcal{S}_n \times \mathcal{A}$, we employ the quantum mean estimation subroutine from Fact 6 with

$$n_k := O\left( \frac{\sqrt{H}}{\theta_k} \log \frac{H|\mathcal{S}_n|AK}{\delta} \right) = O\left( \frac{H^2}{\min\{\epsilon_k, 1\}} \log \frac{H|\mathcal{S}_n|AK}{\delta} \right)$$

queries $\mathcal{O}_p$ to obtain $(\widetilde{\mu}_1^{(k)}(s,a), \ldots, \widetilde{\mu}_H^{(k)}(s,a)) \in \mathbb{R}^H$. These quantities are such that, with probability $1 - \frac{\delta}{2K}$, for all $(s,a,t) \in \mathcal{S}_n \times \mathcal{A} \times [H]$,

$$|\widetilde{\mu}_t^{(k)}(s,a) - \mu_t^{(k)}(s,a)| \leq \theta_k \sqrt{\frac{1}{H} \sum_{t'=1}^{H} \sigma_{t'}^{(k)}(s,a)}, \tag{8a}$$

$$|\widetilde{\sigma}_t^{(k)}(s,a) - \sigma_t^{(k)}(s,a)| \leq \theta_k H^2, \tag{8b}$$

similarly to the classical case, and which we condition on. By using Eq. (8b) onto Eq. (8a), then

$$|\widetilde{\mu}_t^{(k)}(s,a) - \mu_t^{(k)}(s,a)| \leq \theta_k \sqrt{\frac{1}{H} \sum_{t'=1}^{H} \widetilde{\sigma}_{t'}^{(k)}(s,a)} + \theta_k^{3/2} H,$$

from which we define $\widetilde{\mu}_t^{(k)}(x,a)$, for all $s \in \mathcal{S}_n$, $x \in \mathcal{X}(s)$, $a \in \mathcal{A}$, as

$$\widehat{\mu}_t^{(k)}(x,a) := \widetilde{\mu}_t^{(k)}(s,a) - \theta_k \sqrt{\frac{1}{H} \sum_{t'=1}^{H} \widetilde{\sigma}_{t'}^{(k)}(s,a)} - \theta_k^{3/2} H - Ln^{-\alpha} H,$$

which has one-sided error. We can express the above quantity using the variance $\sigma_t^*(x,a) := \int_{\mathcal{X}} p(\mathrm{d}x'|x,a) V_t^*(x')^2 - \left( \int_{\mathcal{X}} p(\mathrm{d}x'|x,a) V_t^*(x') \right)^2$, since, for all $s \in \mathcal{S}_n$, $x \in \mathcal{X}(s)$, and $a \in \mathcal{A}$,

$$\sqrt{\widetilde{\sigma}_t^{(k)}(s,a)} \leq \sqrt{\sigma_t^{(k)}(s,a)} + \sqrt{\theta_k} H \qquad \text{(by Eq. (8b))}$$

$$\leq \sqrt{\sigma_t^{(k)}(x,a)} + \sqrt{\theta_k} H + 2\sqrt{Ln^{-\alpha}} H \quad \text{(by } |\sigma_t^{(k)}(x,a) - \sigma_t^{(k)}(s,a)| \leq 4Ln^{-\alpha} H^2)$$

$$\leq \sqrt{\sigma_t^*(x,a)} + 2\epsilon_k + 8Ln^{-\alpha} H^2 + \sqrt{\theta_k} H + 2\sqrt{Ln^{-\alpha}} H$$

using that $\mathrm{Var}[u_t^{(k-1)}] \leq \mathrm{Var}[V_t^*] + \mathrm{Var}[V_t^* - u_t^{(k-1)}]$ and $\mathrm{Var}[V_t^* - u_t^{(k-1)}] \leq (2\epsilon_k + 8Ln^{-\alpha} H^2)^2$ if $\|V_t^* - u_t^{(k-1)}\|_\infty \leq 2\epsilon_k + 8Ln^{-\alpha} H^2$ according to the induction hypothesis. This means that, for all $(x,a,t) \in \mathcal{X} \times \mathcal{A} \times [H]$, with probability at least $1 - \frac{\delta}{2K}$,

$$\widehat{\mu}_t^{(k)}(x,a) \leq \mu_t^{(k)}(x,a), \tag{9a}$$

$$\widehat{\mu}_t^{(k)}(x,a) \geq \mu_t^{(k)}(x,a) - 2\theta_k \sqrt{\frac{1}{H} \sum_{t'=1}^{H} \sigma_{t'}^*(x,a)} - 2\theta_k(2\epsilon_k + 8Ln^{-\alpha} H^2) - 6\theta_k^{3/2} H - 4Ln^{-\alpha} H. \tag{9b}$$

We now proceed to the backward iteration that happens within epoch $k \in [K]$. Proving that $u_t^{(k)}(x) \leq V_t^{\pi^{(k)}}(x) \leq V_t^*(x)$ is very similar to Theorem 14. Assume by induction that

$$u_{t'}^{(k)}(x) \leq (\mathcal{L}_{\pi_{t'}^{(k)}} u_{t'+1}^{(k)})(x) \leq V_{t'}^{\pi^{(k)}}(x) \leq V_{t'}^*(x) \qquad \forall x \in \mathcal{X}, t' = t+1, t+2, \ldots, H+1.$$

Once $u_{t'+1}^{(k)} \in \mathscr{B}(\mathcal{X})$ has been computed for all $t' = t, t+1, \ldots, H-1$, we can compute the quantities $\widetilde{\beta}_{t+1}^{(k)}$ from Line 12 at time step $t \in [H]$. For all $(s, a) \in \mathcal{S}_n \times \mathcal{A}$, we employ the quantum mean estimation subroutine from Corollary 5 with $\ell_k := O\big(H \log \frac{H|\mathcal{S}_n|AK}{\delta}\big)$ queries to $\mathcal{O}_p$ to obtain $\widehat{\beta}_{t+1}^{(k)}(s, a)$ such that, with probability $1 - \frac{\delta}{2HK}$, for all $s \in \mathcal{S}_n$, $x \in \mathcal{X}(s)$, $a \in \mathcal{A}$,

$$\left| \widetilde{\beta}_{t+1}^{(k)}(s, a) - \int_{\mathcal{X}} p(\mathrm{d}x'|x, a)\big(u_{t+1}^{(k)}(x') - u_{t+1}^{(k-1)}(x')\big) \right| \leq \left( \frac{1}{16H} + Ln^{-\alpha} \right)(2\epsilon_k + 8Ln^{-\alpha}H^2)$$

(using that $\|u_{t+1}^{(k)} - u_{t+1}^{(k-1)}\|_\infty \leq 2\epsilon_k + 8Ln^{-\alpha}H^2$ due to $u_{t+1}^{(k-1)}(x) \leq u_{t+1}^{(k)}(x)$ and the induction hypothesis both in $k$ and $t$). Using that $\big(\frac{1}{16H} + Ln^{-\alpha}\big)(2\epsilon_k + 8Ln^{-\alpha}H^2) \leq \frac{\epsilon_k}{4H} + Ln^{-\alpha}H$ since $Ln^{-\alpha} \leq \frac{1}{16H}$, we define, for all $s \in \mathcal{S}_n$, $x \in \mathcal{X}(s)$, $a \in \mathcal{A}$,

$$\widehat{\beta}_{t+1}^{(k)}(x, a) := \widetilde{\beta}_{t+1}^{(k)}(s, a) - \frac{\epsilon_k}{4H} - Ln^{-\alpha}H,$$

from which, with probability at least $1 - \frac{\delta}{2KH}$, for all $(x, a) \in \mathcal{X} \times \mathcal{A}$, it holds that

$$\widehat{\beta}_{t+1}^{(k)}(x, a) \leq \int_{\mathcal{X}} p(\mathrm{d}x'|x, a)\big(u_{t+1}^{(k)}(x') - u_{t+1}^{(k-1)}(x')\big), \tag{10a}$$

$$\widehat{\beta}_{t+1}^{(k)}(x, a) \geq \int_{\mathcal{X}} p(\mathrm{d}x'|x, a)\big(u_{t+1}^{(k)}(x') - u_{t+1}^{(k-1)}(x')\big) - \frac{\epsilon_k}{2H} - 2Ln^{-\alpha}H. \tag{10b}$$

From here, the proof of $u_t^{(k)}(x) \leq V_t^{\pi^{(k)}}(x) \leq V_t^*(x)$ is the same as Theorem 14, so we omit it and move on to proving that $u_t^{(k)}(x) \geq V_t^*(x) - \epsilon_k - 8Ln^{-\alpha}H$ (assume that all $u_{t'+1}^{(k)}$ for $t' = t, \ldots, H-1$ have already been computed by the backward iteration). Again,

$$V_t^*(x) - u_t^{(k)}(x) \leq \max_{a \in \mathcal{A}} \left\{ r(x, a) + \int_{\mathcal{X}} p(\mathrm{d}x'|x, a)V_{t+1}^*(x') \right\}$$

$$- \max_{a \in \mathcal{A}} \left\{ r(x, a) - 2Ln^{-\alpha} + \widehat{\mu}_{t+1}^{(k)}(x, a) + \widehat{\beta}_{t+1}^{(k)}(x, a) \right\}$$

$$(u_t^{(k)}(x) \geq u_t^{(k)}(s) - Ln^{-\alpha})$$

$$\leq \int_{\mathcal{X}} p(\mathrm{d}x'|x, \pi_t^*(x))\big(V_{t+1}^*(x') - u_{t+1}^{(k)}(x')\big) + \xi_{t+1}^{(k)}(x),$$

(by Eqs. (9b) and (10b))

where we defined

$$\xi_{t+1}^{(k)}(x) := 2\theta_k \sqrt{\frac{1}{H} \sum_{t'=1}^{H} \sigma_{t'}^*(x, \pi_t^*(x))} + \frac{\epsilon_k}{2H} + 2\theta_k(2\epsilon_k + 8Ln^{-\alpha}H^2) + 6\theta_k^{3/2}H + 6Ln^{-\alpha}H.$$

Solving the above recursion, using $\|\sum_{t'=t}^{H-1} \int \cdots \int_{\mathcal{X}^{t'-t}} p_{\pi_t^*}(\mathrm{d}x_{t+1}|x) \cdots p_{\pi_{t'-1}^*}(\mathrm{d}x_{t'}|x_{t'-1})\|_\infty \leq H$, and

$$\left\| \sum_{t'=t}^{H-1} \int \cdots \int_{\mathcal{X}^{t'-t}} p_{\pi_t^*}(\mathrm{d}x_{t+1}|x) \cdots p_{\pi_{t'}^*}(\mathrm{d}x_{t'-1}|x_{t'-1}) \sqrt{\frac{1}{H} \sum_{t''=1}^{H} \sigma_{t''}^*(x_{t'}, \pi_{t'}^*(x_{t'}))} \right\|_\infty$$

$$\leq \left\| \sqrt{\sum_{t'=t}^{H-1} \int \cdots \int_{\mathcal{X}^{t'-t}} p_{\pi_t^*}(\mathrm{d}x_{t+1}|x) \cdots p_{\pi_{t'-1}^*}(\mathrm{d}x_{t'}|x_{t'-1}) \sum_{t''=1}^{H} \sigma_{t''}^*(x_{t'}, \pi_{t'}^*(x_{t'}))} \right\|_\infty \leq 2H^{3/2},$$

(by Cauchy–Schwarz inequality and Lemma 15)

we finally get that

$$V_t^*(x) - u_t^{(k)}(x) \leq 4\theta_k H^{3/2} + \frac{\epsilon_k}{2} + 2\theta_k(2\epsilon_k + 8Ln^{-\alpha}H^2)H + 6\theta_k^{3/2}H^2 + 6Ln^{-\alpha}H^2$$

$$\leq \epsilon_k \left( \frac{1}{2} + \frac{4}{20} + \frac{4}{20H^{3/2}} + \frac{6}{20^{3/2}H^{1/4}} \right) + \left( \frac{2 \cdot 8}{20\sqrt{H}} + 6 \right)Ln^{-\alpha}H^2$$

$$(\theta_k = \frac{1}{20H^{3/2}} \min\{\epsilon_k, 1\})$$

$$\leq \epsilon_k + 8Ln^{-\alpha}H^2,$$

---

**Algorithm 4** Simple quantum backward induction algorithm

---

**Input:** Compact state space $\mathcal{X}$ with $\frac{1}{n}$-net $\mathcal{S}_n$, finite action space $\mathcal{A}$, horizon $H$, quantum sampling access to probability kernels $p$, failure probability $\delta \in (0,1)$, error $\varepsilon > 0$.

**Output:** $(\varepsilon + (1+H)HLn^{-\alpha})$-optimal policy $\pi \in \Pi^{\mathrm{D}}$.

1: $\widehat{u}_H(s) \leftarrow \min_{a \in \mathcal{A}}\{r(s,a)\}$ and $\pi_H(s) \leftarrow \arg\min_{a \in \mathcal{A}}\{r(s,a)\}$ with probability $1 - \frac{\delta}{H|\mathcal{S}_n|}$ for all $s \in \mathcal{S}_n$ (Fact 3)

2: $u_H(x) \leftarrow \widehat{u}_H(s) - Ln^{-\alpha}$ and $\pi_H(x) \leftarrow \pi_H(s)$ for all $s \in \mathcal{S}_n$ and $x \in \mathcal{X}(s)$

3: **for** $t = H-1, H-2, \ldots, 1$ **do**

4:     **for** $s \in \mathcal{S}_n$ **do**

5:         Use Corollary 5 to obtain a unitary $\mathcal{U}_s^{(t+1)} : |a\rangle|\bar{0}\rangle \mapsto |a\rangle|\mu_{t+1}(s,a)\rangle$ for all $a \in \mathcal{A}$, where $|\mu_{t+1}(s,a) - \int_{\mathcal{X}} p(\mathrm{d}x'|s,a)u_{t+1}(x')| \leq \frac{\varepsilon}{2H}$ with high probability

6:         Use quantum maximum finding with unitary $\mathcal{U}_s^{(t+1)}$ (Fact 3 with $\mathcal{U}_s^{(t+1)}$) to obtain $\widehat{u}_t(s)$ and $a_t(s)$ such that, with probability $1 - \frac{\delta}{H|\mathcal{S}_n|}$,

$$\widehat{u}_t(s) = \max_{a \in \mathcal{A}}\{r(s,a) + \mu_{t+1}(s,a)\} \quad \text{and} \quad \pi_t(s) = \arg\max_{a \in \mathcal{A}}\{r(s,a) + \mu_{t+1}(s,a)\}$$

7:         $u_t(x) \leftarrow \widehat{u}_t(s) - \frac{\varepsilon}{2H} - (1+H)Ln^{-\alpha}$ and $\pi_t(x) \leftarrow \pi_t(s)$ for all $x \in \mathcal{X}$

8:         If $u_t(s) \leq u_{t+1}(s)$, then $u_t(x) \leftarrow u_{t+1}(x)$ and $\pi_t(x) \leftarrow \pi_{t+1}(x)$ for all $x \in \mathcal{X}(s)$

9:     **end for**

10: **end for**

11: **return** $\pi = (\pi_1, \ldots, \pi_H) \in \Pi^{\mathrm{D}}$

---

This concludes the proof of Eq. (7) for epoch $k \in [K]$ and thus for all epochs by induction.

Regarding the failure probability, Eq. (9) holds with probability $1 - \frac{\delta}{2K}$ for a single epoch, while Eq. (10) holds with probability $1 - \frac{\delta}{2HK}$ for a single epoch and time step. Therefore, across all epochs and time steps, the success probability is at least $1 - \delta$, as required. Finally, the total number of samples used is

$$O\left(\sum_{k=1}^{K}(m_k + n_k + H\ell_k)|\mathcal{S}_n|A\right) = O\left(\frac{H^2|\mathcal{S}_n|A}{\varepsilon}\log\left(\frac{H|\mathcal{S}_n|A}{\delta}\log\frac{H}{\varepsilon}\right)\right). \qquad \square$$

Our second quantum algorithm (Algorithm 4) is based on a simpler backward induction algorithm without variance reduction and total-variance techniques: we simply compute $u_t(x) = (\mathcal{L}u_{t+1})(x)$ for $t = H, H-1, \ldots, 1$, i.e., in a backward fashion, using quantum subroutines. Although this will inevitably lead to a worse sample complexity on the horizon $H$, it is possible now to employ quantum minimum finding (Dürr & Høyer, 1996) together with quantum mean estimation, which brings down the sample complexity on the action space size from $O(A)$ down to $O(\sqrt{A})$.

**Theorem 17** (Simple quantum finite-horizon generative algorithm). *Let $M = \langle \mathcal{X}, \mathcal{A}, H, p, r \rangle$ be a finite-horizon MDP and let $\mathcal{S}_n$ be a $\frac{1}{n}$-net for $\mathcal{X}$. Let $\varepsilon > 0$ and $\delta \in (0,1)$. Under Assumption 10 with parameters $L, \alpha \geq 0$, Algorithm 4 computes functions $\{u_t \in \mathscr{B}(\mathcal{X})\}_{t \in [H]}$ and policy $\pi \in \Pi^{\mathrm{D}}$ such that, with probability $1 - \delta$,*

$$V_t^*(x) - \varepsilon - 2(1+H)HLn^{-\alpha} \leq u_t(x) \leq V_t^\pi(x) \leq V_t^*(x) \qquad \forall(x,t) \in \mathcal{X} \times [H].$$

*The $\mathcal{O}_p, \mathcal{O}_p^\dagger$-query complexity is*

$$O\left(\frac{H^3|\mathcal{S}_n|\sqrt{A}}{\varepsilon}\log\left(\frac{H|\mathcal{S}_n|A}{\delta}\right)\log\left(\frac{H|\mathcal{S}_n|}{\delta}\right)\right).$$

*Proof.* The proof is by induction on $t \in [H]$. We start by analysing the quantity $\widehat{u}_t$ in Line 6 at time step $t \in [H]$. First notice that, for all $s \in \mathcal{S}_n$ and $x \in \mathcal{X}(s)$,

$$|(\mathcal{L}u_{t+1})(s) - (\mathcal{L}u_{t+1})(x)| \leq |r(s,a) - r(x,a)| + \left|\int_{\mathcal{X}}(p(\mathrm{d}x'|s,a) - p(\mathrm{d}x'|x,a))u_{t+1}(x')\right|$$

$$\leq (1+H)Ln^{-\alpha}, \qquad\qquad (\|u_{t+1}\|_\infty \leq H \text{ by induction})$$

where $a = \arg\max_{a' \in \mathcal{A}}(\mathcal{L}_{a'} u_{t+1})(s)$. With probability $1 - \frac{\delta}{H|\mathcal{S}_n|}$, $|\widehat{u}_t(s) - (\mathcal{L} u_{t+1})(s)| \leq \frac{\varepsilon}{2H}$ and thus $u_t(x) := \widehat{u}_t(s) - \frac{\varepsilon}{2H} - (1+H)Ln^{-\alpha}$ for $s \in \mathcal{S}_n$ and $x \in \mathcal{X}(s)$ is such that

$$u_t(x) \leq (\mathcal{L} u_{t+1})(x) \quad \text{and} \quad u_t(x) \geq (\mathcal{L} u_{t+1})(x) - \frac{\varepsilon}{H} - 2(1+H)Ln^{-\alpha}. \tag{11}$$

From here, proving the inequality $u_t(x) \leq V_t^\pi(x)$ is done exactly as in Theorems 14 and 16 as long as $u_t(x) \leq (\mathcal{L} u_{t+1})(x)$ (Eq. (11)) and $u_{t+1}(x) \leq u_t(x)$, which is enforced by design (Line 8). To prove that $V_t^*(x) - \varepsilon - (1+H)HLn^{-\alpha} \leq u_t(x)$, we follow the proof of Theorems 14 and 16,

$$V_t^*(x) - u_t(x) \leq (\mathcal{L} V_{t+1}^*)(x) - (\mathcal{L} u_{t+1})(x) + \frac{\varepsilon}{H} + 2(1+H)Ln^{-\alpha}$$

$$\leq \int_{\mathcal{X}} p(\mathrm{d}x'|x, \pi_t^*(x))\big(V_{t+1}^*(x') - u_{t+1}(x')\big) + \frac{\varepsilon}{H} + 2(1+H)Ln^{-\alpha}.$$

Solving the above recursion and using $\|\sum_{t'=t}^{H-1} \int \cdots \int_{\mathcal{X}^{t'-t}} p_{\pi_t^*}(\mathrm{d}x_{t+1}|x) \cdots p_{\pi_{t'-1}^*}(\mathrm{d}x_{t'}|x_{t'-1})\|_\infty \leq H$, we get that

$$V_t^*(x) - u_t(x) \leq \varepsilon + 2(1+H)HLn^{-\alpha},$$

which concludes the proof of correctness.

We now analyse the success probability of Algorithm 4. To do so, we must analyse how quantum oracles fail (see (Chen & de Wolf, 2023, Appendix A) for a similar argument as follows). Ideally, we would like to implement the unitary $\mathcal{U}_{s,\mathrm{ideal}}^{(t+1)} : |a\rangle|\bar{0}\rangle \mapsto |a\rangle|\mu_{t+1}(s,a)\rangle$ where $|\mu_{t+1}(s,a)\rangle$ contains the approximation $|\mu_{t+1}(s,a) - \int_{\mathcal{X}} p(\mathrm{d}x'|s,a)u_{t+1}(x')| \leq \frac{\varepsilon}{2H}$. In practice, however, we implement the unitary $\mathcal{U}_s^{(t+1)} : |a\rangle|\bar{0}\rangle \mapsto |a\rangle(\sqrt{1-\delta_2}|\mu_{t+1}(s,a)\rangle + \sqrt{\delta_2}|\phi_a\rangle)$ for some $\delta_2 \in (0,1)$, where the register $|\mu_{t+1}(s,a)\rangle$ holds the desired approximation $\mu_{t+1}(s,a)$ and $|\phi_a\rangle$ is a normalised quantum state orthogonal to $|\mu_{t+1}(s,a)\rangle$. Notice that

$$\forall a \in \mathcal{A}: \quad \|(\mathcal{U}_{s,\mathrm{ideal}}^{(t+1)} - \mathcal{U}_s^{(t+1)})|a\rangle|\bar{0}\rangle\| = \sqrt{(1-\sqrt{1-\delta_2})^2 + \delta_2} = \sqrt{2 - 2\sqrt{1-\delta_2}} \leq \sqrt{2\delta_2},$$

using that $\sqrt{1-\delta_2} \geq 1 - \delta_2$. Since quantum minimum finding does not take into account the action of $\mathcal{U}_s^{(t+1)}$ onto states of the form $|a\rangle|\bar{0}^\perp\rangle$ for $|\bar{0}^\perp\rangle$ orthogonal to $|\bar{0}\rangle$, we can, without of loss of generality, assume that $\|\mathcal{U}_{s,\mathrm{ideal}}^{(t+1)} - \mathcal{U}_s^{(t+1)}\| \leq \sqrt{2\delta_2}$. The success probability of quantum minimum finding (Fact 3) is $1 - \delta_1$ when employing $\mathcal{U}_{s,\mathrm{ideal}}^{(t+1)}$, for some $\delta_1 \in (0,1)$. However, since it employs $\mathcal{U}_s^{(t+1)}$ instead, the success probability decreases by at most the spectral norm of the difference between the "real" and the "ideal" total unitaries. To be more precise, the "ideal" quantum minimum finding is a sequence of gates $\mathcal{A} = U_1 E_1 U_2 E_2 \cdots U_N E_N$, where $U_i \in \{\mathcal{U}_{s,\mathrm{ideal}}^{(t+1)}, \mathcal{U}_{s,\mathrm{ideal}}^{(t+1)\dagger}\}$, $E_i$ is a circuit of elementary gates, and $N = c\sqrt{A}\log\frac{1}{\delta_1}$ with $c$ constant is the number of queries to $\mathcal{U}_{s,\mathrm{ideal}}^{(t+1)}$. The "real" implementation, on the other hand, is $\widetilde{\mathcal{A}} = \widetilde{U}_1 E_1 \widetilde{U}_2 E_2 \cdots \widetilde{U}_N E_N$, where $\widetilde{U}_i \in \{\mathcal{U}_s^{(t+1)}, \mathcal{U}_s^{(t+1)\dagger}\}$. Then $\|\mathcal{A} - \widetilde{\mathcal{A}}\| \leq c\sqrt{A}\log\big(\frac{1}{\delta_1}\big)\|\mathcal{U}_{s,\mathrm{ideal}}^{(t+1)} - \mathcal{U}_s^{(t+1)}\| \leq c\sqrt{A}\log\big(\frac{1}{\delta_1}\big)\sqrt{2\delta_2}$ and the failure probability is $\delta_1 + c\sqrt{A}\log\frac{1}{\delta_1}\sqrt{2\delta_2}$. By taking $\delta_1 = O\big(\frac{\delta}{H|\mathcal{S}_n|}\big)$ and $\delta_2 = O\big(\frac{\delta_1^2}{A\log^2(1/\delta_1)}\big)$, the failure probability in outputting $\max_{a \in \mathcal{A}}\{r(s,a) + \mu_{t+1}(s,a)\}$ is at most $\frac{\delta}{H|\mathcal{S}_n|}$. By a usual union bound over all $s \in \mathcal{S}_n$ and $t \in [H]$, the failure probability is at most $\delta$.[12]

Regarding the query complexity, for each $s \in \mathcal{S}_n$, one call to the unitary $\mathcal{U}_s^{(t+1)}$ uses $O\big(\frac{\|u_{t+1}\|_\infty}{\varepsilon/H}\log\frac{1}{\delta_2}\big) = O\big(\frac{H^2}{\varepsilon}\log\frac{H|\mathcal{S}_n|A}{\delta}\big)$ queries to $\mathcal{O}_p$, while quantum maximum finding makes $O\big(\sqrt{A}\log\frac{1}{\delta_1}\big) = O\big(\sqrt{A}\log\frac{H|\mathcal{S}_n|}{\delta}\big)$ queries to $\mathcal{U}_s^{(t+1)}$. Summing over all $\mathcal{S}_n \times [H]$, the total query complexity is

$$O\bigg(\frac{H^3|\mathcal{S}_n|\sqrt{A}}{\varepsilon}\log\frac{H|\mathcal{S}_n|A}{\delta}\log\frac{H|\mathcal{S}_n|}{\delta}\bigg). \qquad \square$$

---

[12](Wang et al., 2021, Theorem 6) employs a similar but slightly incorrect argument for the failure probability. More precisely, they ignore the failure probability $\delta_1$ coming from quantum minimum finding, or rather, assume it to be constant. Their runtime incorrectly ignores the factor $\log\frac{1}{\delta_1}$ (Wang et al., 2021, Theorem 7).

# D    COMPUTING OPTIMAL POLICIES FOR INFINITE-HORIZON MDPS

In this section, we focus on finding approximate optimal policies for infinite-horizon weakly communicating MDPs $M = \langle \mathcal{X}, \mathcal{A}, p, r \rangle$ such that $\mathrm{sp}(h^*) \leq \Lambda$. Several classical algorithms have been proposed for such task, one of them main ones being value iteration (Kearns & Singh, 1998b; Lutter et al., 2021; Hartmanns & Kaminski, 2020; Bertsekas, 1998; Quatmann & Katoen, 2018; Shani et al., 2007; Weng & Zanuttini, 2013), which is quite similar to the backward induction algorithm covered in the last section: starting from some initial function $u_0 \in \mathcal{B}(\mathcal{X})$, normally $u_0 \equiv 0$, generate a sequence of functions $(u_t)_{t \in \mathbb{N}}$ according to the update rule $u_{t+1} = \mathcal{L}u_t$. It is known that when $\mathrm{sp}(u_{t+1} - u_t) \leq \varepsilon$, the greedy policy with respect to $u_t$ is $\varepsilon$-optimal (Puterman, 2014, Theorem 9.4.5).

Here we are interested in robust versions of value iteration wherein errors in computing $\mathcal{L}u_t$ are taken into account. Approximate versions of value iteration have been studied and are used in various settings (Farahmand et al., 2010; Mann et al., 2015; Ernst et al., 2005; Munos, 2007; De Farias & Van Roy, 2000; Van Roy, 2006), and here we consider a robust analogue of value iteration which differs from its standard implementation by generating a sequence of functions $(u_t)_{t \in \mathbb{N}}$ such that

$$\|u_{t+1} - \mathcal{L}u_t\|_\infty \leq \varepsilon_u \quad \text{for a given } \varepsilon_u \geq 0, \quad \text{where } t \in \mathbb{N}. \tag{12}$$

The error $\varepsilon_u$ could come from, e.g., approximating $\int_{\mathcal{X}} p(\mathrm{d}x'|x, a)u_t(x')$ or the maximum over $a \in \mathcal{A}$. In the following, we prove a few convergence results for the robust value iteration, starting by proving that any sequence of vectors generated as in Eq. (12) using some span contraction have bounded span.

**Lemma 18.** *Let $\epsilon \geq 0$ and $\mathcal{N} : \mathcal{B}(\mathcal{X}) \to \mathcal{B}(\mathcal{X})$ a 1-stage $\nu$-span contraction for some $\nu \in [0, 1)$. Let $(u_t)_{t \in \mathbb{N}}$ be a sequence of functions such that $\|u_{t+1} - \mathcal{N}u_t\|_\infty \leq \epsilon \; \forall t \in \mathbb{N}$. Then*

$$\mathrm{sp}(u_{t+1} - u_t) \leq \nu^t(\mathrm{sp}(\mathcal{N}u_0 - u_0) + 2\epsilon) + 4\epsilon \frac{1 - \nu^t}{1 - \nu} \qquad \text{for all } t \in \mathbb{N}.$$

*Proof.* We prove the main result by induction on $t$. For $t = 0$, $\mathrm{sp}(u_1 - u_0) \leq 2\epsilon + \mathrm{sp}(\mathcal{N}u_0 - u_0)$ using that $\mathrm{sp}(v) \leq 2\|v\|_\infty$ for any $v \in \mathcal{B}(\mathcal{X})$. For $t > 0$,

$$\mathrm{sp}(u_{t+1} - u_t) \leq \mathrm{sp}(u_{t+1} - \mathcal{N}u_t) + \mathrm{sp}(u_t - \mathcal{N}u_{t-1}) + \mathrm{sp}(\mathcal{N}u_t - \mathcal{N}u_{t-1})$$

$$\text{(triangle inequality)}$$

$$\leq 4\epsilon + \nu \, \mathrm{sp}(u_t - u_{t-1}) \quad \text{(1-stage $\nu$-span contraction and } \|u_{t+1} - \mathcal{N}u_t\|_\infty \leq \epsilon)$$

$$\leq 4\epsilon + \nu \left( \nu^{t-1}(\mathrm{sp}(\mathcal{N}u_0 - u_0) + 2\epsilon) + 4\epsilon \frac{1 - \nu^{t-1}}{1 - \nu} \right) \quad \text{(induction hypothesis)}$$

$$= \nu^t(\mathrm{sp}(\mathcal{N}u_0 - u_0) + 2\epsilon) + 4\epsilon \frac{1 - \nu^t}{1 - \nu}. \qquad \square$$

If the error per iteration $\varepsilon_u$ is small enough, the above result tells us that after several iterations, robust value iteration obtains a pair of vectors $u_{t+1}$ and $u_t$ such that $\mathrm{sp}(u_{t+1} - u_t)$ is sufficiently small. It remains to show that, once a stopping criteria of the form

$$\mathrm{sp}(u_{t+1} - u_t) \leq \varepsilon_s \quad \text{for some } \varepsilon_s > 0$$

is achieved, a corresponding policy $d_t^\infty$ associated with $u_{t+1}$ is close to optimal. The next result generalises (Puterman, 2014, Theorem 8.5.6).

**Theorem 19.** *Let $\varepsilon_u, \varepsilon_s > 0$. Let $(u_t)_{t \in \mathbb{N}}$ be any sequence of vectors such that $\|u_{t+1} - \mathcal{L}u_t\|_\infty \leq \varepsilon_u$ for all $t \in \mathbb{N}$. Assume that $\mathrm{sp}(u_{t+1} - u_t) \leq \varepsilon_s$ for some $\varepsilon_s > 0$ and $t \geq t_\varepsilon \in \mathbb{N}$. Let $d_\varepsilon \in \mathcal{D}^D$ such that $\|u_{t_\varepsilon+1} - \mathcal{L}_{d_\varepsilon}u_{t_\varepsilon}\|_\infty \leq \varepsilon_u$. Define the quantity*

$$g^\varepsilon := \frac{1}{2} \left( \max_{x \in \mathcal{X}}\{u_{t_\varepsilon+1}(x) - u_{t_\varepsilon}(x)\} + \min_{x \in \mathcal{X}}\{u_{t_\varepsilon+1}(x) - u_{t_\varepsilon}(x)\} \right).$$

*Then $\|g^{d_\varepsilon^\infty} - g^\varepsilon e\|_\infty \leq \varepsilon_u + \frac{\varepsilon_s}{2}$ and $|g^\varepsilon - g^*| \leq \varepsilon_u + \frac{\varepsilon_s}{2}$, and so $d_\varepsilon^\infty$ is a $(2\varepsilon_u + \varepsilon_s)$-optimal policy.*

*Proof.* By the definition of $g^{d_\varepsilon^\infty}$, the stochastic kernel $p_{d_\varepsilon}(\cdot|x)$ is positive Harris recurrent with unique invariant probability measure $\mu_x^{d_\varepsilon}$ such that $g^{d_\varepsilon^\infty} = \int_\mathcal{X} r_{d_\varepsilon}(x')\mu_x^{d_\varepsilon}(\mathrm{d}x')$ (Saldi et al., 2017, Theorem 2.5). Furthermore, since $\int_\mathcal{X} \mu_x^{d_\varepsilon}(\mathrm{d}x')p_{d_\varepsilon}(\cdot|x) = \mu_x^{d_\varepsilon}(\cdot)$, then

$$g^{d_\varepsilon^\infty}(x) = \int_\mathcal{X} \mu_x^{d_\varepsilon}(\mathrm{d}x')\left(r_{d_\varepsilon}(x') + \int_\mathcal{X} p_{d_\varepsilon}(\mathrm{d}x''|x')u_{t_\varepsilon}(x'') - u_{t_\varepsilon}(x')\right)$$

$$= \int_\mathcal{X} \mu_x^{d_\varepsilon}(\mathrm{d}x')\big((\mathcal{L}_{d_\varepsilon} u_{t_\varepsilon})(x') - u_{t_\varepsilon}(x')\big).$$

Since $\inf_{x'\in\mathcal{X}}\{u_{t_\varepsilon+1}(x') - u_{t_\varepsilon}(x')\} \le u_{t_\varepsilon+1}(x) - u_{t_\varepsilon}(x) \le \sup_{x'\in\mathcal{X}}\{u_{t_\varepsilon+1}(x') - u_{t_\varepsilon+1}(x')\}$ for all $x \in \mathcal{X}$, then

$$\inf_{x'\in\mathcal{X}}\{u_{t_\varepsilon+1}(x') - u_{t_\varepsilon}(x')\} \le \int_\mathcal{X} \mu_x^{d_\varepsilon}(\mathrm{d}x')\big(u_{t_\varepsilon+1}(x') - u_{t_\varepsilon}(x')\big) \le \sup_{x'\in\mathcal{X}}\{u_{t_\varepsilon+1}(x') - u_{t_\varepsilon+1}(x')\},$$

from which it follows that

$$\left|\int_\mathcal{X} \mu_x^{d_\varepsilon}(\mathrm{d}x')\big(u_{t_\varepsilon+1}(x') - u_{t_\varepsilon}(x')\big) - g^\varepsilon\right| \le \frac{1}{2}\mathrm{sp}(u_{t_\varepsilon+1} - u_{t_\varepsilon}) \le \frac{\varepsilon_s}{2} \implies \|g^{d_\varepsilon^\infty} - g^\varepsilon e\|_\infty \le \varepsilon_u + \frac{\varepsilon_s}{2},$$

using that $\|u_{t_\varepsilon+1} - \mathcal{L}_{d_\varepsilon} u_{t_\varepsilon}\|_\infty \le \varepsilon_u$. By repeating the same procedure but with the optimal policy $(d^*)^\infty$ instead of $d_\varepsilon^\infty$, then $|g^* - g^\varepsilon| \le \varepsilon_u + \frac{\varepsilon_s}{2}$ (this time using that $\|u_{t+1} - \mathcal{L}u_t\|_\infty \le \varepsilon_u$). □

### D.1 CLASSICAL VALUE ITERATION ALGORITHM UNDER A GENERATIVE MODEL

A few different classical algorithms for computing optimal policies under a generative model for infinite-horizon MDPs have been proposed (Wang, 2017a; Jin & Sidford, 2020; 2021; Wang et al., 2022; Zhang & Xie, 2023; Li et al., 2024), one of the best complexities being $\widetilde{O}\big(\frac{\Lambda^2 SA}{\varepsilon^2}\big)$ due to Zhang & Xie (2023). Here we provide, for completeness, a simple approximate version of standard value iteration (Algorithm 5), already generalised for compact state spaces, wherein the quantities $\int_\mathcal{X} p(\mathrm{d}x'|x,a)u_t(x')$ are approximated using classical samples via a Hoeffding bound, similarly to the case of finite-horizon MDPs from the previous section or infinite-horizon discounted MDPs from Sidford et al. (2018a); Li et al. (2020).

**Theorem 20** (Classical infinite-horizon generative algorithm). *Let $M = \langle \mathcal{X}, \mathcal{A}, p, r\rangle$ be an infinite-horizon weakly communicating MDP with $\mathrm{sp}(h^*) \le \Lambda$. Let $\mathcal{S}_n$ be a $\frac{1}{n}$-net for $\mathcal{X}$. Assume the optimal Bellman operator $\mathcal{L}$ of $M$ is a 1-stage $\nu$-span contraction for $\nu \in [0,1)$. Assume classical sampling access to $p$ via oracle $\mathcal{C}_p$. Let $\delta \in (0,1)$ and $\varepsilon \in (0, \frac{2}{\nu}]$. Under Assumption 10 with parameters $L, \alpha \ge 0$ such that $Ln^{-\alpha} \le \frac{1-\nu}{\nu}$, Algorithm 5 outputs an $\big(2\varepsilon + \frac{26(1+\Lambda)Ln^{-\alpha}}{1-\nu}\big)$-optimal policy $\widetilde{d}^\infty$ and $g^\varepsilon$ such that $|g^\varepsilon - g^*| < \varepsilon + \frac{13(1+\Lambda)Ln^{-\alpha}}{1-\nu}$ with probability $1-\delta$. Its $\mathcal{C}_p$-query complexity is (up to $\mathrm{poly}\log\log$ factors in $1/\varepsilon$ and $\nu$)*

$$\widetilde{O}\left(\min\left\{(1+\Lambda)^2, \frac{\Lambda^2}{(1-\nu)^2}\right\}\frac{|\mathcal{S}_n|A}{(1-\nu)^2\varepsilon^2}\frac{\log\frac{1}{\varepsilon}}{\log\frac{1}{\nu}}\log\frac{|\mathcal{S}_n|A}{\delta}\right).$$

*Proof.* In the following, let $\varepsilon_u, \bar\varepsilon_u > 0$ be such that $\varepsilon_u := \frac{1}{4}(1-\nu)\varepsilon$ and $\bar\varepsilon_u := \varepsilon_u + (1+\Lambda)Ln^{-\alpha}$. We start by proving that Algorithm 5 generates a sequence of functions $(u_t)_{t\in\mathbb{N}}$ such that $\|u_{t+1} - \mathcal{L}u_t\|_\infty \le \varepsilon_u + 4(1+\Lambda)Ln^{-\alpha}$, $\mathrm{sp}(u_{t+1}) \le \mathrm{sp}(\mathcal{L}u_t)$, $\mathrm{sp}(u_{t+1}) \le \frac{2\Lambda}{1-\nu}$, and $\mathrm{sp}(u_t - h^*) \le \nu^t\Lambda + \frac{2\bar\varepsilon_u}{1-\nu}$, which also implies that

$$\mathrm{sp}(u_{t+1}) \le \mathrm{sp}(\mathcal{L}u_t) \le \mathrm{sp}(h^*) + \mathrm{sp}(\mathcal{L}u_t - \mathcal{L}h^*) \le (1+\nu^{t+1})\Lambda + \frac{2\nu\bar\varepsilon_u}{1-\nu} \le 4\Lambda + 3,$$

assuming that $Ln^{-\alpha} \le \frac{1-\nu}{\nu}$ and $\varepsilon_u \le \frac{1-\nu}{2\nu}$. For $t = 0$, $u_1(x) = \max_{a\in\mathcal{A}}\{r(s,a)\}$ for all $s \in \mathcal{S}_n$ and $x \in \mathcal{X}(s)$ and then $\|u_1 - \mathcal{L}0\|_\infty = \max_{x\in\mathcal{X}}\{u_1(x) - \max_{a\in\mathcal{A}} r(x,a)\} \le Ln^{-\alpha}$. Moreover,

$$\mathrm{sp}(u_1 - h^*) \le \mathrm{sp}(\mathcal{L}0 - \mathcal{L}h^*) + \mathrm{sp}(u_1 - \mathcal{L}0) \le \nu\Lambda + 2Ln^{-\alpha} \le \nu\Lambda + \frac{2\bar\varepsilon_u}{1-\nu},$$

$$\mathrm{sp}(u_1) \le \mathrm{sp}(\mathcal{L}0) \le \mathrm{sp}(h^*) + \mathrm{sp}(\mathcal{L}0 - \mathcal{L}h^*) \le (1+\nu)\Lambda \le \frac{2\Lambda}{1-\nu},$$

---

**Algorithm 5** Classical approximate value iteration algorithm

---

**Input:** Compact state space $\mathcal{X}$ with $\frac{1}{n}$-net $\mathcal{S}_n$, finite action space $\mathcal{A}$, classical sampling access to probability kernels $p$, failure probability $\delta \in (0,1)$, parameters $\Lambda > 0$, $\nu \in [0,1)$, $L, \alpha \geq 0$, errors $\varepsilon, \varepsilon_u > 0$ such that $\varepsilon_u = \frac{1}{4}(1-\nu)\varepsilon$.

**Output:** $\left(2\varepsilon + \frac{26(1+\Lambda)Ln^{-\alpha}}{1-\nu}\right)$-optimal stationary policy $\widetilde{d}^\infty$.

1: $t \leftarrow 1$ and initialise $u_0 \leftarrow 0$ and $u_1(x) \leftarrow \max_{a \in \mathcal{A}}\{r(s,a)\}$ for all $s \in \mathcal{S}_n$ and $x \in \mathcal{X}(s)$

2: **while** $\mathrm{sp}(u_t - u_{t-1}) > \frac{3\varepsilon}{2} + \frac{18(1+\Lambda)Ln^{-\alpha}}{1-\nu}$ **do**

3: $\quad m_t \leftarrow \left\lceil \frac{512}{(1-\nu)^2\varepsilon^2}\min\{4(1+\Lambda)^2, \frac{\Lambda^2}{(1-\nu)^2}\} \ln \frac{\pi^2 t^2 |\mathcal{S}_n| A}{6\delta}\right\rceil$

4: $\quad$ For $(s,a) \in \mathcal{S}_n \times \mathcal{A}$, sample $x_{s,a}^{(1)}, x_{s,a}^{(2)}, \ldots, x_{s,a}^{(m_t)} \in \mathcal{X}$

5: $\quad$ For all $s \in \mathcal{S}_n$ and $x \in \mathcal{X}(s)$, $a_{t+1}(x) \leftarrow \arg\max_{a \in \mathcal{A}}\{r(s,a) + \frac{1}{m_t}\sum_{i=1}^{m_t} u_t(x_{s,a}^{(i)})\}$,

$\quad$ $\widetilde{u}_{t+1}(s) \leftarrow \max_{a \in \mathcal{A}}\{r(s,a) + \frac{1}{m_t}\sum_{i=1}^{m_t} u_t(x_{s,a}^{(i)})\}$, and

$$u_{t+1}(x) \leftarrow \begin{cases} \max\left\{\max_{s' \in \mathcal{S}_n} \widetilde{u}_{t+1}(s') - \frac{\varepsilon_u}{2}, \min_{s' \in \mathcal{S}_n} \widetilde{u}_{t+1}(s')\right\} & \text{if } \widetilde{u}_{t+1}(s) \geq \max_{s' \in \mathcal{S}_n} \widetilde{u}_{t+1}(s') - \frac{\varepsilon_u}{2}, \\[2mm] \min\left\{\min_{s' \in \mathcal{S}_n} \widetilde{u}_{t+1}(s') + \frac{\varepsilon_u}{2}, \max_{s' \in \mathcal{S}_n} \widetilde{u}_{t+1}(s')\right\} & \text{if } \widetilde{u}_{t+1}(s) \leq \min_{s' \in \mathcal{S}_n} \widetilde{u}_{t+1}(s') + \frac{\varepsilon_u}{2}, \\[2mm] \widetilde{u}_{t+1}(s) & \text{otherwise.} \end{cases}$$

6: $\quad t \leftarrow t + 1$

7: **end while**

8: **return** $\widetilde{d} \in \mathcal{D}^{\mathrm{D}}$ where $\widetilde{d}(x) := a_t(x)$ for all $x \in \mathcal{X}$

---

where we used that $\mathcal{L}h^* = h^* + g^* e$ and the span-contractive property of $\mathcal{L}$.

Fix now $t \geq 1$ and assume by induction that $(u_{t'})_{t' \in [t]}$ have been computed. By a Hoeffding bound (Fact 7) we have that, for all $(s,a) \in \mathcal{S}_n \times \mathcal{A}$, with probability $1 - \frac{6\delta}{\pi^2 t^2 |\mathcal{S}_n| A}$,

$$\left|\mu_{t+1}(s,a) - \int_{\mathcal{X}} p(\mathrm{d}x'|s,a)u_t(x')\right| \leq \sqrt{\frac{2}{m_t}\ln\frac{\pi^2 t^2 |\mathcal{S}_n| A}{6\delta}}\,\mathrm{sp}(u_t) \leq \frac{\varepsilon_u}{2},$$

using that $m_t := \left\lceil\frac{32}{\varepsilon_u^2}\min\{4(1+\Lambda)^2, \frac{\Lambda^2}{(1-\nu)^2}\}\ln\frac{\pi^2 t^2 |\mathcal{S}_n| A}{6\delta}\right\rceil$ since $\mathrm{sp}(u_t) \leq \min\{4(1+\Lambda), \frac{2\Lambda}{1-\nu}\}$ by induction (we can subtract $\min_{x' \in \mathcal{X}} u_t(x')$ from $u_t(x)$ so that $\|u_t - \min_{x' \in \mathcal{X}}\{u_t(x')\}e\|_\infty = \mathrm{sp}(u_t)$ since $u_t(x) \geq 0$). For $s \in \mathcal{S}_n$, define $\widetilde{u}_{t+1}(s) := \max_{a \in \mathcal{A}}\{r(s,a) + \mu_{t+1}(s,a)\}$ and

$$u_{t+1}(s) = \begin{cases} \max\{\max_{s' \in \mathcal{S}_n} \widetilde{u}_{t+1}(s') - \frac{\varepsilon_u}{2}, \min_{s' \in \mathcal{S}_n} \widetilde{u}_{t+1}(s')\} & \text{if } \widetilde{u}_{t+1}(s) \geq \max_{s' \in \mathcal{S}_n} \widetilde{u}_{t+1}(s') - \frac{\varepsilon_u}{2}, \\ \min\{\min_{s' \in \mathcal{S}_n} \widetilde{u}_{t+1}(s') + \frac{\varepsilon_u}{2}, \max_{s' \in \mathcal{S}_n} \widetilde{u}_{t+1}(s')\} & \text{if } \widetilde{u}_{t+1}(s) \leq \min_{s' \in \mathcal{S}_n} \widetilde{u}_{t+1}(s') + \frac{\varepsilon_u}{2}, \\ \widetilde{u}_{t+1}(s) & \text{otherwise.} \end{cases}$$

In other words, the smaller entries of $\widetilde{u}_{t+1}$ are increases by $\frac{\varepsilon_u}{2}$, while its larger entries are decreased by $\frac{\varepsilon_u}{2}$. This means that $\mathrm{sp}(u_{t+1}) \leq \mathrm{sp}(\mathcal{L}u_t)$. Moreover, $|u_{t+1}(s) - (\mathcal{L}u_t)(s)| \leq \varepsilon_u$ for all $s \in \mathcal{S}_n$. Observe that, for all $s \in \mathcal{S}_n$ and $x \in \mathcal{X}(s)$,

$$|(\mathcal{L}u_t)(s) - (\mathcal{L}u_t)(x)| \leq |r(s,a) - r(x,a)| + \left|\int_{\mathcal{X}}(p(\mathrm{d}x'|s,a) - p(\mathrm{d}x'|x,a))\left(u_t(x') - \min_{x'' \in \mathcal{X}} u_t(x'')\right)\right|$$

$$\leq 4(1+\Lambda)Ln^{-\alpha}, \qquad\qquad (\mathrm{sp}(u_t) \leq 4\Lambda + 3 \text{ by induction})$$

where $a = \arg\max_{a' \in \mathcal{A}}(\mathcal{L}_{a'}u_t)(s)$. By defining $u_{t+1}(x) = u_{t+1}(s)$ for all $s \in \mathcal{S}_n$ and $x \in \mathcal{X}(s)$, then $\|u_{t+1} - \mathcal{L}u_t\|_\infty \leq \varepsilon_u + 4(1+\Lambda)Ln^{-\alpha}$. Consider now the operator $\bar{\mathcal{L}} : \mathscr{B}(\mathcal{X}) \to \mathscr{B}(\mathcal{X})$ such that $(\bar{\mathcal{L}}v)(x) = (\mathcal{L}v)(s)$ for all $s \in \mathcal{S}_n$ and $x \in \mathcal{X}(s)$. Notice that $\|\mathcal{L}h^* - \bar{\mathcal{L}}h^*\|_\infty \leq (1+\Lambda)Ln^{-\alpha}$

similarly to above. Therefore,

$$\mathrm{sp}(u_{t+1} - h^*) \leq \mathrm{sp}(u_{t+1} - \bar{\mathcal{L}}u_t) + \mathrm{sp}(\bar{\mathcal{L}}u_t - \bar{\mathcal{L}}h^*) + \mathrm{sp}(\bar{\mathcal{L}}h^* - \mathcal{L}h^*) \quad \text{(triangle inequality)}$$

$$\leq 2\varepsilon_u + \nu\,\mathrm{sp}(u_t - h^*) + 2(1+\Lambda)Ln^{-\alpha} \quad (\mathrm{sp}(v) \leq 2\|v\|_\infty \text{ and } \mathcal{L} \text{ contraction})$$

$$\leq 2\bar{\varepsilon}_u + \nu^{t+1}\Lambda + \frac{2\nu}{1-\nu}\bar{\varepsilon}_u \quad \text{(induction hypothesis)}$$

$$= \nu^{t+1}\Lambda + \frac{2\bar{\varepsilon}_u}{1-\nu}.$$

On the other hand, it is also true that

$$\mathrm{sp}(u_{t+1}) \leq \mathrm{sp}(\mathcal{L}u_t) \leq \mathrm{sp}(h^*) + \mathrm{sp}(\mathcal{L}u_t - \mathcal{L}h^*) \leq \Lambda + \nu\frac{2\Lambda}{1-\nu} \leq \frac{2\Lambda}{1-\nu}.$$

This concludes the proof by induction that $\|u_{t+1} - \mathcal{L}u_t\|_\infty \leq \varepsilon_u + 4(1+\Lambda)Ln^{-\alpha}$ and $\mathrm{sp}(u_{t+1}) \leq \min\{4\Lambda + 3, \frac{2\Lambda}{1-\nu}\}$.

We now note that, since $\|u_{t+1} - \mathcal{L}u_t\|_\infty \leq \varepsilon_u + 4(1+\Lambda)Ln^{-\alpha}$, Lemma 18 yields that

$$\mathrm{sp}(u_{t+1} - u_t) \leq \nu^t(1 + 2\varepsilon_u + 8(1+\Lambda)Ln^{-\alpha}) + \frac{4\varepsilon_u + 16(1+\Lambda)Ln^{-\alpha}}{1-\nu} \quad \forall t \in \mathbb{N},$$

using that $\mathrm{sp}(\mathcal{L}0) \leq 1$. Therefore, $\mathrm{sp}(u_{t+1} - u_t) \leq \frac{6\varepsilon_u + 18(1+\Lambda)Ln^{-\alpha}}{1-\nu}$ if

$$t \geq t^* := \left\lceil \frac{\log \frac{(1-\nu)(1+2\varepsilon_u + 8(1+\Lambda)Ln^{-\alpha})}{2\varepsilon_u + 2(1+\Lambda)Ln^{-\alpha}}}{\log \frac{1}{\nu}} \right\rceil = O\left(\frac{\log \frac{1}{\varepsilon}}{\log \frac{1}{\nu}}\right).$$

We employ Theorem 19 to argue that, once $\mathrm{sp}(u_{t+1} - u_t) \leq \frac{6\varepsilon_u + 18(1+\Lambda)Ln^{-\alpha}}{1-\nu}$ for $t \geq t^*$, then

$$g^\varepsilon = \frac{1}{2}\left(\max_{x\in\mathcal{X}}\{u_{t^*+1}(x) - u_{t^*}(x)\} + \min_{x\in\mathcal{X}}\{u_{t^*+1}(x) - u_{t^*}(x)\}\right)$$

is such that $|g^\varepsilon - g^*| \leq \varepsilon_u + 4(1+\Lambda)Ln^{-\alpha} + \frac{3\varepsilon_u + 9(1+\Lambda)Ln^{-\alpha}}{1-\nu} < \varepsilon + \frac{13(1+\Lambda)Ln^{-\alpha}}{1-\nu}$, while the policy $\widetilde{d}^\infty \in \Pi^D$ with $\widetilde{d}(x) := \arg\max_{a\in\mathcal{A}}\{u_{t^*}(x)\}$ for all $x \in \mathcal{X}$ is $\left(2\varepsilon + \frac{26(1+\Lambda)Ln^{-\alpha}}{1-\nu}\right)$-optimal.

We now analyse the query complexity of Algorithm 5. Computing $u_{t+1}$ for a fixed step $t \in \mathbb{N}$ requires $m_t|\mathcal{S}_n|A$ queries to $\mathcal{C}_p$, i.e., $O\left(\min\{(1+\Lambda)^2, \frac{\Lambda^2}{(1-\nu)^2}\}\frac{|\mathcal{S}_n|A}{(1-\nu)^2\varepsilon^2}\log\frac{t|\mathcal{S}_n|A}{\delta}\right)$ samples. The total query complexity is then $O\left(t^*\min\{(1+\Lambda)^2, \frac{\Lambda^2}{(1-\nu)^2}\}\frac{|\mathcal{S}_n|A}{(1-\nu)^2\varepsilon^2}\log\frac{t|\mathcal{S}_n|A}{\delta}\right)$. Finally, the success probability comes from a simple union bound and the identity $\sum_{t=1}^\infty \frac{1}{t^2} = \frac{\pi^2}{6}$. $\qquad\square$

## D.2 QUANTUM VALUE ITERATION ALGORITHM UNDER A GENERATIVE MODEL

In the same way that we quantised the finite-horizon classical backward iteration algorithm in Appendix C by approximating the quantities $\int_\mathcal{X} p(dx'|x,a)u_t(x')$ using quantum mean estimation, a similar procedure can be performed for the aforementioned classical value iteration algorithm on infinite-horizon MDPs. The resulting algorithm, shown in Algorithm 6, achieves a better query complexity dependence on the action space size $A$, error parameter $\varepsilon$, and bias span upper bound $\Lambda$.

**Theorem 21** (Quantum infinite-horizon generative algorithm). *Let $M = \langle \mathcal{X}, \mathcal{A}, p, r \rangle$ be an infinite-horizon weakly communicating MDP with $\mathrm{sp}(h^*) \leq \Lambda$. Let $\mathcal{S}_n$ be a $\frac{1}{n}$-net for $\mathcal{X}$. Assume the optimal Bellman operator $\mathcal{L}$ of $M$ is a 1-stage $\nu$-span contraction for $\nu \in [0,1)$. Assume quantum quantum sampling access to $p$ via oracle $\mathcal{O}_p$. Let $\delta \in (0,1)$ and $\varepsilon \in (0, \frac{2}{\nu}]$. Under Assumption 10 with parameters $L, \alpha \geq 0$ such that $Ln^{-\alpha} \leq \frac{1-\nu}{\nu}$, Algorithm 6 outputs an $\left(2\varepsilon + \frac{26(1+\Lambda)Ln^{-\alpha}}{1-\nu}\right)$-optimal policy $\widetilde{d}^\infty$ and $g^\varepsilon$ such that $|g^\varepsilon - g^*| < \varepsilon + \frac{13(1+\Lambda)Ln^{-\alpha}}{1-\nu}$ with probability $1-\delta$. Its $\mathcal{O}_p, \mathcal{O}_p^\dagger$-query complexity is (up to $\mathrm{poly}\log\log$ factors in $1/\varepsilon$ and $\nu$)*

$$\widetilde{O}\left(\min\left\{1 + \Lambda, \frac{\Lambda}{1-\nu}\right\}\frac{|\mathcal{S}_n|\sqrt{A}}{(1-\nu)\varepsilon}\frac{\log\frac{1}{\varepsilon}}{\log\frac{1}{\nu}}\log\frac{|\mathcal{S}_n|A}{\delta}\log\frac{|\mathcal{S}_n|}{\delta}\right).$$

---

**Algorithm 6** Quantum value iteration algorithm

---

**Input:** Compact state space $\mathcal{X}$ with $\frac{1}{n}$-net $\mathcal{S}_n$, finite action space $\mathcal{A}$, quantum sampling access to probability kernels $p$, failure probability $\delta \in (0, 1)$, parameters $\Lambda > 0$, $\nu \in [0, 1)$, $L, \alpha \geq 0$, errors $\varepsilon_u, \varepsilon > 0$ such that $\varepsilon_u = \frac{1}{4}(1 - \nu)\varepsilon$.

**Output:** $\left(2\varepsilon + \frac{26(1+\Lambda)Ln^{-\alpha}}{1-\nu}\right)$-optimal stationary policy $\widetilde{d}^{\infty}$.

1: $t \leftarrow 1$ and initialise $u_0 \leftarrow 0$

2: $u_1(x) \leftarrow \max_{a \in \mathcal{A}}\{r(s, a)\}$ for all $s \in \mathcal{S}_n$ and $x \in \mathcal{X}(s)$ with probability $1 - \frac{6\delta}{\pi^2|\mathcal{S}_n|}$ (Fact 3)

3: **while** $\mathrm{sp}(u_t - u_{t-1}) > \frac{3\varepsilon}{2} + \frac{18(1+\Lambda)Ln^{-\alpha}}{1-\nu}$ **do**

4:      Build quantum access to $u_t$

5:      **for** $s \in \mathcal{S}_n$ **do**

6:          Use Corollary 5 to obtain a unitary $\mathcal{U}_s^{(t)} : |a\rangle|\bar{0}\rangle \mapsto |a\rangle|\mu_t(s, a)\rangle$ for all $a \in \mathcal{A}$, where $|\mu_t(s, a) - \int_{\mathcal{X}} p(\mathrm{d}x'|s, a)u_t(x')| \leq \frac{\varepsilon_u}{2}$ with high probability

7:          Use quantum maximum finding with unitary $\mathcal{U}_s^{(t)}$ (Fact 3 with $\mathcal{U}_s^{(t)}$) to get $\widetilde{u}_{t+1}(s)$ and $a_{t+1}(s)$ such that, with probability $1 - \frac{6\delta}{\pi^2 t^2|\mathcal{S}_n|}$,

$$\widetilde{u}_{t+1}(s) = \max_{a \in \mathcal{A}}\{r(s, a) + \mu_{t+1}(s, a)\} \quad \text{and} \quad a_{t+1}(s) = \arg\max_{a \in \mathcal{A}}\{r(s, a) + \mu_{t+1}(s, a)\}$$

8:      **end for**

9:      For all $s \in \mathcal{S}_n$ and $x \in \mathcal{X}(s)$, $a_{t+1}(x) \leftarrow a_{t+1}(s)$ and

$$u_{t+1}(x) \leftarrow \begin{cases} \max\left\{\max_{s' \in \mathcal{S}_n} \widetilde{u}_{t+1}(s') - \dfrac{\varepsilon_u}{2}, \min_{s' \in \mathcal{S}_n} \widetilde{u}_{t+1}(s')\right\} & \text{if } \widetilde{u}_{t+1}(s) \geq \max_{s' \in \mathcal{S}_n} \widetilde{u}_{t+1}(s') - \dfrac{\varepsilon_u}{2}, \\[3mm] \min\left\{\min_{s' \in \mathcal{S}_n} \widetilde{u}_{t+1}(s') + \dfrac{\varepsilon_u}{2}, \max_{s' \in \mathcal{S}_n} \widetilde{u}_{t+1}(s')\right\} & \text{if } \widetilde{u}_{t+1}(s) \leq \min_{s' \in \mathcal{S}_n} \widetilde{u}_{t+1}(s') + \dfrac{\varepsilon_u}{2}, \\[3mm] \widetilde{u}_{t+1}(s) & \text{otherwise.} \end{cases}$$

10:      $t \leftarrow t + 1$

11: **end while**

12: **return** $\widetilde{d} \in \mathcal{D}^{\mathrm{D}}$ where $\widetilde{d}(x) := a_t(x)$ for all $x \in \mathcal{X}$

---

*Proof.* In the following, let $\varepsilon_u, \bar{\varepsilon}_u > 0$ such that $\varepsilon_u := \frac{1}{4}(1 - \nu)\varepsilon$ and $\bar{\varepsilon}_u := \varepsilon_u + (1 + \Lambda)Ln^{-\alpha}$. The proof of correctness is basically the same as Theorem 20, the main difference being the computation of $\mu_{t+1}(s, a)$ and thus $u_{t+1}$. Again by induction on $t$ we would like to prove that Algorithm 6 outputs a sequence of functions $(u_t)_{t \in \mathbb{N}}$ such that $\|u_{t+1} - \mathcal{L}u_t\|_{\infty} \leq \varepsilon_u + 4(1 + \Lambda)Ln^{-\alpha}$, $\mathrm{sp}(u_{t+1}) \leq \mathrm{sp}(\mathcal{L}u_t)$, $\mathrm{sp}(u_{t+1}) \leq \min\{4\Lambda + 3, \frac{2\Lambda}{1-\nu}\}$, and $\mathrm{sp}(u_t - h^*) \leq \nu^t \Lambda + \frac{2\bar{\varepsilon}_u}{1-\nu}$.

The case $t = 0$ is the same as in Theorem 20. For $t \geq 1$, we must show how to approximate $\int_{\mathcal{X}} p(\mathrm{d}x'|s, a)u_t(x')$ for all $(s, a) \in \mathcal{S}_n \times \mathcal{A}$. We claim that Corollary 5 can be adapted to perform all its steps in superposition without any need for intermediary measurements. This means that, given an initial state $\sum_{a \in \mathcal{A}} \alpha_a|s, a\rangle|\bar{0}\rangle$ for fixed $s \in \mathcal{S}_n$ and some $\{\alpha_a\}_{a \in \mathcal{A}} \subset \mathbb{C}$, Corollary 5 can generate the mapping

$$\sum_{a \in \mathcal{A}} \alpha_a|s, a\rangle|\bar{0}\rangle \mapsto \sum_{a \in \mathcal{A}} \alpha_a|s, a\rangle|\mu_{t+1}(s, a)\rangle|\operatorname{garbage}(a)\rangle,$$

where $\{|\operatorname{garbage}(a)\rangle\}_{a \in \mathcal{A}}$ are "garbage" unit complex vectors accumulated through the computation that we ignore and

$$\left|\mu_{t+1}(s, a) - \int_{\mathcal{X}} p(\mathrm{d}x'|s, a)u_t(x')\right| \leq \frac{\varepsilon_u}{2} \qquad \forall (s, a) \in \mathcal{S}_n \times \mathcal{A}.$$

It is then possible to effectively construct a black-box unitary $\mathcal{U}_s^{(t)} : |a\rangle|\bar{0}\rangle \mapsto |a\rangle|\mu_{t+1}(s, a)\rangle$ (up to garbage states) using oracle $\mathcal{O}_p$. The maximum over $a \in \mathcal{A}$ of $r(s, a) + \mu_{t+1}(s, a)$ can thus be found by using the quantum max-finding subroutine (Fact 3) with unitary $\mathcal{U}_s^{(t)}$. This leads to $\widetilde{u}_{t+1}(s) = \max_{a \in \mathcal{A}}\{r(s, a) + \mu_{t+1}(s, a)\}$ and $a_{t+1}(s) = \arg\max_{a \in \mathcal{A}}\{r(s, a) + \mu_{t+1}(s, a)\}$ such that $|\widetilde{u}_{t+1}(s) - (\mathcal{L}u_t)(s)| \leq \frac{\varepsilon_u}{2}$ for all $s \in \mathcal{S}_n$. Again, by subtracting $\frac{\varepsilon_u}{2}$ from the larger entries of

$\widetilde{u}_{t+1}$ and adding $\frac{\varepsilon_u}{2}$ to the smaller entries of $\widetilde{u}_{t+1}$, we define, for all $s \in \mathcal{S}_n$ and $\mathcal{X}(s)$,

$$u_{t+1}(x) = \begin{cases} \max\{\max_{s' \in \mathcal{S}_n} \widetilde{u}_{t+1}(s') - \frac{\varepsilon_u}{2}, \min_{s' \in \mathcal{S}_n} \widetilde{u}_{t+1}(s')\} & \text{if } \widetilde{u}_{t+1}(s) \geq \max_{s' \in \mathcal{S}_n} \widetilde{u}_{t+1}(s') - \frac{\varepsilon_u}{2}, \\ \min\{\min_{s' \in \mathcal{S}_n} \widetilde{u}_{t+1}(s') + \frac{\varepsilon_u}{2}, \max_{s' \in \mathcal{S}_n} \widetilde{u}_{t+1}(s')\} & \text{if } \widetilde{u}_{t+1}(s) \leq \min_{s' \in \mathcal{S}_n} \widetilde{u}_{t+1}(s') + \frac{\varepsilon_u}{2}, \\ \widetilde{u}_{t+1}(s) & \text{otherwise.} \end{cases}$$

Using that $|(\mathcal{L}u_t)(s) - (\mathcal{L}u_t)(x)| \leq 4(1+\Lambda)Ln^{-\alpha}$ for all $s \in \mathcal{S}_n$ and $x \in \mathcal{X}(s)$ (since $\mathrm{sp}(u_t) \leq 4\Lambda + 3$ by induction), we once again conclude that $\|u_{t+1} - \mathcal{L}u_t\|_\infty \leq \varepsilon_u + 4(1+\Lambda)Ln^{-\alpha}$. Proving that $\mathrm{sp}(u_{t+1}) \leq \min\{4\Lambda + 3, \frac{2\Lambda}{1-\nu}\}$ is done exactly the same as in Theorem 20.

By Lemma 18, $\mathrm{sp}(u_{t+1} - u_t) \leq \frac{6\varepsilon_u + 18(1+\Lambda)Ln^{-\alpha}}{1-\nu}$ if $t \geq t^* = O(\log(1/\varepsilon)/\log(1/\nu))$. We employ Theorem 19 to argue that, once $\mathrm{sp}(u_{t+1} - u_t) \leq \frac{6\varepsilon_u + 18(1+\Lambda)Ln^{-\alpha}}{1-\nu}$ for $t \geq t^*$, the output

$$g^\varepsilon = \frac{1}{2}\left(\max_{x \in \mathcal{X}}\{u_{t^*+1}(x) - u_{t^*}(x)\} + \min_{x \in \mathcal{X}}\{u_{t^*+1}(x) - u_{t^*}(x)\}\right)$$

is such that $|g^\varepsilon - g^*| \leq \varepsilon_u + 4(1+\Lambda)Ln^{-\alpha} + \frac{3\varepsilon_u + 9(1+\Lambda)Ln^{-\alpha}}{1-\nu} < \varepsilon + \frac{13(1+\Lambda)Ln^{-\alpha}}{1-\nu}$, while the policy $\widetilde{d}^\infty \in \Pi^D$ with $\widetilde{d}(x) := \arg\max_{a \in \mathcal{A}}\{u_{t^*}(x)\}$ for all $x \in \mathcal{X}$ is $\left(2\varepsilon + \frac{26(1+\Lambda)Ln^{-\alpha}}{1-\nu}\right)$-optimal.

The analysis of the success probability of Algorithm 6 is very similar to the one in Theorem 17. Taking into account the oracle failure probabilities, outputting $\max_{a \in \mathcal{A}}\{r(s,a) + \mu_{t+1}(s,a)\}$ fails with probability at most $\frac{6\delta}{\pi^2 t^2 |\mathcal{S}_n|}$. By a usual union bound over all $s \in \mathcal{S}_n$ and $t \in \mathbb{N}$, the failure probability is at most $\delta$, since $\sum_{t=1}^\infty \frac{1}{t^2} = \frac{\pi^2}{6}$.

Finally, for the query complexity of Algorithm 6, we start from obtaining $u_{t+1}$ given $u_t$. For every $s \in \mathcal{S}_n$, each call to the unitary $\mathcal{U}_s^{(t)}$ uses $O\left(\frac{\mathrm{sp}(u_t)}{\varepsilon_u}\log\frac{t|\mathcal{S}_n|A}{\delta}\right)$ queries to $\mathcal{O}_p$, while quantum maximum finding makes $O\left(\sqrt{A}\log\frac{t|\mathcal{S}_n|}{\delta}\right)$ queries to $\mathcal{U}_s^{(t)}$. Thus the query complexity of computing $u_{t+1}(s)$ (and $a_{t+1}(s)$) is $O\left(\min\{1+\Lambda, \frac{\Lambda}{1-\nu}\}\frac{\sqrt{A}}{\varepsilon_u}\log\frac{t|\mathcal{S}_n|A}{\delta}\log\frac{t|\mathcal{S}_n|}{\delta}\right)$, already using that $\mathrm{sp}(u_t) \leq \min\{4\Lambda + 3, \frac{2\Lambda}{1-\nu}\}$. Summing over all $s \in \mathcal{S}_n$ and $t \leq t^*$, the total query complexity of Algorithm 6 is simply $O\left(t^*\min\{1+\Lambda, \frac{\Lambda}{1-\nu}\}\frac{|\mathcal{S}_n|\sqrt{A}}{(1-\nu)\varepsilon}\log\frac{t^*|\mathcal{S}_n|A}{\delta}\log\frac{t^*|\mathcal{S}_n|}{\delta}\right)$. □

# E  ALGORITHMS FOR ONLINE LEARNING OF FINITE-HORIZON MDPS

In this section, we give quantum and classical online algorithms for learning finite-horizon MDPs under the generative-exploration model of Appendix B. Thanks to the freedom in interacting with the environment during generative episodes using oracles $\mathcal{O}_p$ or $\mathcal{C}_p$, our algorithms achieve improved regret bounds.

Both our quantum and classical algorithms, presented together in Algorithm 7, are schematically simple. Like several classical RL algorithms for finite-horizon MDPs (Kearns & Singh, 2002; Brafman & Tennenholtz, 2002; Strehl et al., 2006; Auer et al., 2008; Dann et al., 2017), they proceed in episodes. Each episode is split into a generative phase and an exploration phase, which were explained in Appendix B. During a generative phase, an approximate optimal policy is chosen for the subsequent exploration phases, while during an exploration phase, the agent gets to interact using the environment in a classical manner through the previously chosen policy and thus to accumulate rewards. We stress that each exploration phase has a fixed duration equal to the horizon $H$, i.e., the online agent-environment interaction lasts for $H$ time steps, after which the episode ends.

Regarding the generative phase more specifically, the choice for an approximate optimal policy is done using a doubling trick: a new policy is only chosen after the agent has interacted enough times with the environment during exploration phases. In our case, this means only after the total number of interactions (time steps) has doubled since the last policy update. When this condition is reached at the $k$-th episode, the classical or quantum backward induction algorithms (Algorithms 2 to 4) with sampling access to stochastic kernels $p$ are employed to obtained an approximately optimal policy. The algorithm is allowed to use the oracle $\mathcal{C}_p$ or $\mathcal{O}_p$ a number of $\frac{1}{2}H\log_2 k$ times, the amount of time steps in exploration phases since the last update. During $K$ episodes, meaning $T = HK$ time

---

**Algorithm 7** Classical/quantum online-learning algorithm for finite-horizon MDPs

---

**Input:** Compact state space $\mathcal{X}$ with $\frac{1}{n}$-net $\mathcal{S}_n$, finite action space $\mathcal{A}$, horizon $H$, rewards $r : \mathcal{X} \times \mathcal{A} \to [0,1]$, failure probability $\delta \in (0,1)$, parameters $L, \alpha \geq 0$.

1: **for** episodes $k \in [K]$ **do**
       **Generative phase**
2:     **if** $k = 2^{\lfloor \log_2 k \rfloor}$ **then**
3:         **Classical:** By using Algorithm 2, choose a policy $\pi^{(k)} \in \Pi^{\mathrm{D}}$ such that, with failure probability at most $\frac{\delta}{\lceil \log_2 K \rceil}$ and for some constant $c > 0$,

$$\|V_1^{\pi^{(k)}} - V_1^*\|_\infty \leq c \sqrt{\frac{H^2 |\mathcal{S}_n| A}{k} \log \left( \frac{H |\mathcal{S}_n| A \log K}{\delta} \right)} + cLn^{-\alpha} H^2$$

3:         **Quantum:** By using Algorithm 3 or Algorithm 4, choose a policy $\pi^{(k)} \in \Pi^{\mathrm{D}}$ such that, with failure probability at most $\frac{\delta}{\lceil \log_2 K \rceil}$ and for some constant $c > 0$,

$$\|V_1^{\pi^{(k)}} - V_1^*\|_\infty \leq c \frac{|\mathcal{S}_n|}{k} \min\{HA, H^2 \sqrt{A}\} \log^2 \left( \frac{H |\mathcal{S}_n| A \log K}{\delta} \right) + cLn^{-\alpha} H^2$$

4:     **else**
5:         $\pi^{(k)} \leftarrow \pi^{(k-1)}$
6:     **end if**
       **Exploration phase**
7:     Observe initial state $x_1^{(k)}$
8:     **for** $t \in [H]$ **do**
9:         Choose action $a_t^{(k)} = \pi_t^{(k)}(x_t^{(k)})$ and obtain reward $r_t^{(k)} \leftarrow r(x_t^{(k)}, a_t^{(k)})$
10:       Observe next state $x_{t+1}^{(k)} \sim p(\cdot | x_t^{(k)}, a_t^{(k)})$
11:    **end for**
12: **end for**

---

steps, the policy is updated at most $\lceil \log_2 K \rceil$ times. In episodes where the doubling trick condition has not been satisfied yet, the policy from the previous episode is selected as the new current policy.

Unlike prior RL algorithms for finite-horizon MDPs (Kearns & Singh, 2002; Brafman & Tennenholtz, 2002; Strehl et al., 2006; Auer et al., 2008; Dann et al., 2017; Wang et al., 2021), Algorithm 7 does not keep an approximation $\widetilde{p}$ for the true stochastic kernel $p$ via the state-action pairs $(x_t, a_t)$ observed during exploration phases and therefore does not employ $\widetilde{p}$ to calculate approximate optimal policies. Since it is possible to interact with the true MDP $M$ in a generative manner using oracles $\mathcal{C}_p$ or $\mathcal{O}_p$, picking policies can be done in a more straightforward way, which ultimately is the cause for our improved regrets.

We start by analysing in the theorem below the regret bound of the classical version of Algorithm 7, after which we move on to its quantum version.

**Theorem 22** (Classical finite-horizon regret bound). *Let $M = \langle \mathcal{X}, \mathcal{A}, H, p, r \rangle$ be a finite-horizon MDP. Then the regret $\mathrm{Regret}_H(T)$ of the classical version of Algorithm 7 is upper-bounded after $T$ steps, with probability at least $1 - \delta$, by*

$$\widetilde{O} \left( \sqrt{H |\mathcal{S}_n| AT \log \frac{H |\mathcal{S}_n| A}{\delta}} + HTLn^{-\alpha} \right),$$

*where $\widetilde{O}(\cdot)$ omits poly log log terms. If $\mathcal{X} = [0,1]^D$, in which case $|\mathcal{S}_n| = O(n^D)$, then setting $n = T^{\frac{1}{D+2\alpha}}$ yields the regret (for $L$ constant)*

$$\widetilde{O} \left( T^{\frac{D+\alpha}{D+2\alpha}} \left( H + \sqrt{HA \log \frac{HAT}{\delta}} \right) \right).$$

If $\mathcal{X} = \mathcal{S}$ is finite with size $S$, in which case $L = 0$, then the regret is

$$\widetilde{O}\left(\sqrt{HSAT \log \frac{HSA}{\delta}}\right).$$

*Proof.* The total regret over $K$ episodes ($T = HK$ time steps) is

$$\mathrm{Regret}_H(T) := \sum_{k=1}^{K} \left(V_1^*(x_1^{(k)}) - V_1^{\pi^{(k)}}(x_1^{(k)})\right).$$

The policy $\pi^{(k)}$ is updated every time $k$ is a power of two ($k = 2^{\lfloor \log_2 k \rfloor}$), to a total of $\lceil \log_2 K \rceil$ times. At each update, we employ Algorithm 2 with $O(Hk)$ calls to oracle $\mathcal{C}_p$ in order to obtain a policy $\pi^{(k)}$ such that, according to Theorem 14, with probability at least $1 - \frac{\delta}{\lceil \log_2 K \rceil}$, for all $x \in \mathcal{X}$,

$$V_1^*(x) - \frac{C}{\sqrt{k}} - 12H^2 L n^{-\alpha} \leq V_1^{\pi^{(k)}}(x) \leq V_1^*(x), \quad \text{where } C = \widetilde{O}\left(\sqrt{H^2 |\mathcal{S}_n| A \log \frac{H |\mathcal{S}_n| A}{\delta}}\right)$$

and $\widetilde{O}(\cdot)$ omits poly log log terms. This means that, with probability at least $1 - \delta$, the regret is upper-bounded as

$$\mathrm{Regret}_H(T) \leq \sum_{k=1}^{K}\left(\frac{C}{\sqrt{k}} + 12H^2 L n^{-\alpha}\right) \leq \frac{C\sqrt{K}}{\sqrt{2}-1} + 12H^2 K L n^{-\alpha} = \frac{C}{\sqrt{2}-1}\sqrt{\frac{T}{H}} + 12HTLn^{-\alpha},$$

where we used Fact 1 in order to bound $\sum_{k=1}^{K} \frac{1}{\sqrt{k}} \leq \frac{\sqrt{K}}{\sqrt{2}-1}$. $\qquad \square$

**Theorem 23** (Quantum finite-horizon regret bound). *Let $M = \langle \mathcal{X}, \mathcal{A}, H, p, r \rangle$ be a finite-horizon MDP. Then the regret $\mathrm{Regret}_H(T)$ of the quantum version of Algorithm 7 is upper-bounded after $T$ steps, with probability at least $1 - \delta$, by*

$$\widetilde{O}\left(|\mathcal{S}_n| \min\left\{HA, H^2\sqrt{A}\log\frac{H|\mathcal{S}_n|}{\delta}\right\}\log\frac{T}{H}\log\frac{H|\mathcal{S}_n|A}{\delta} + HTLn^{-\alpha}\right),$$

*where $\widetilde{O}(\cdot)$ omits poly log log terms. If $\mathcal{X} = [0,1]^D$, in which case $|\mathcal{S}_n| = O(n^D)$, then setting $n = T^{\frac{1}{D+\alpha}}$ yields the regret (for $L$ constant)*

$$\widetilde{O}\left(T^{\frac{D}{D+\alpha}}\min\left\{HA, H^2\sqrt{A}\log\frac{HT}{\delta}\right\}\log\frac{T}{H}\log\frac{HTA}{\delta}\right).$$

*If $\mathcal{X} = \mathcal{S}$ is finite with size $S$, in which case $L = 0$, then the regret is*

$$\widetilde{O}\left(S\min\left\{HA, H^2\sqrt{A}\log\frac{HS}{\delta}\right\}\log\frac{T}{H}\log\frac{HSA}{\delta}\right).$$

*Proof.* The proof is the same as Theorem 22, the main difference being that now we employ Algorithm 3 or Algorithm 4 with $O(Hk)$ calls to oracle $\mathcal{O}_p$ in order to obtain a policy $\pi^{(k)}$ such that, according to Theorems 16 and 17, with probability at least $1 - \frac{\delta}{\lceil \log_2 K \rceil}$, for all $x \in \mathcal{X}$,

$$V_1^*(x) - \frac{C}{k} - 8H^2 L n^{-\alpha} \leq V_1^{\pi^{(k)}}(x) \leq V_1^*(x),$$

$$\text{where } C = \widetilde{O}\left(|\mathcal{S}_n| \min\left\{HA, H^2\sqrt{A}\log\frac{H|\mathcal{S}_n|}{\delta}\right\}\log\frac{H|\mathcal{S}_n|A}{\delta}\right)$$

and $\widetilde{O}(\cdot)$ omits poly log log terms. This means that, with probability at least $1 - \delta$, the regret is upper-bounded as

$$\mathrm{Regret}_H(T) \leq \sum_{k=1}^{K}\left(\frac{C}{k} + 8H^2 L n^{-\alpha}\right) \leq C\ln K + 8H^2 K L n^{-\alpha} = C\ln\frac{T}{H} + 8HTLn^{-\alpha}. \quad \square$$

# F ALGORITHMS FOR ONLINE LEARNING OF INFINITE-HORIZON MDPs

In this section, we give quantum and classical online algorithms for learning infinite-horizon MDPs under the generative-exploration model of Appendix B. Once again, thanks to the freedom in interacting with the environment during generative episodes using oracles $\mathcal{O}_p$ or $\mathcal{C}_p$, our algorithms achieve improved regret bounds.

Both our quantum and classical algorithms, presented together in Algorithm 8, are schematically simple. Like several classical RL algorithms (Ortner & Ryabko, 2012; Auer & Ortner, 2006; Auer et al., 2008; Bartlett & Tewari, 2009; Lakshmanan et al., 2015; Fruit et al., 2018), they proceed in episodes, during each of which a chosen policy remains fixed. Each episode is split into a generative phase and an exploration phase, which were explained in Appendix B. In our quantum RL algorithm, during the $k$-th generative phase, a $O(\log(\tau_k)/\tau_k)$-optimal policy for the exploration phase is chosen using Algorithm 6 with quantum sampling access to stochastic kernels $p$ and quantum access to rewards $r$, where $\tau_k$ is the number of time steps taken in the previous exploration phase. Similarly, our classical RL algorithm chooses a $O(\sqrt{\log(\tau_k)/\tau_k})$-optimal policy using Algorithm 5 instead. After the generative phase ends, the exploration phase starts, during which the chosen policy is employed to accumulate rewards until a termination criteria is reached. Unlike the case for finite-horizon MDPs where each exploration phase lasts the horizon $H$, for infinite-horizon MPDs the exploration phases can be arbitrarily long and it is dependent on the agent to end the episode and initiate the next generate phase. In our case, we again use a doubling trick to move into a new episode once the number of time steps within all exploration phases doubles. In other words, the doubling trick is employed here to end an exploration phase, while for finite-horizon MPDs it was employed to decide whether to update a policy within a generate phase or not.

Unlike prior classical RL algorithms (Ortner & Ryabko, 2012; Auer & Ortner, 2006; Auer et al., 2008; Bartlett & Tewari, 2009; Lakshmanan et al., 2015; Fruit et al., 2018), Algorithm 8 does not keep estimates of the true stochastic kernels given the state-action pairs $(x_t, a_t)$ observed during exploration phases, and therefore, it does not adhere to the standard "optimism-in-the-face-of-uncertainty" principle of maintaining a set of plausible MDPs $\mathcal{M}$ that contains the true MDP $M$ with high probability. Since it is possible to directly interact with the true MDP $M$ via quantum or classical-accessible environments (oracles $\mathcal{O}_p$ or $\mathcal{C}_p$), the choice for an approximate optimal policy does not need to take all MDPs in $\mathcal{M}$ into account.

## F.1 IN-PATH REGRET

We study the performance of Algorithm 8 both in terms of the in-path regret $\mathrm{Regret}_\infty^{\mathrm{path}}(T)$ and the expected regret $\mathrm{Regret}_\infty^{\mathbb{E}}(T)$, starting with the former.

**Theorem 24** (Classical infinite-horizon in-path regret bound). *Let $M = \langle \mathcal{X}, \mathcal{A}, p, r \rangle$ be an infinite-horizon weakly communicating MDP with $\mathrm{sp}(h^*) \leq \Lambda$. Let $\mathcal{S}_n$ be an $\frac{1}{n}$-net for $\mathcal{X}$. Assume that $p$ and $r$ satisfy Assumption 10 with parameters $L, \alpha \geq 0$. Assume the optimal Bellman operator $\mathcal{L}$ of $M$ is an 1-stage $\nu$-span contraction for $\nu \in [0, 1)$. For any $T, n \in \mathbb{N}$, the in-path regret $\mathrm{Regret}_\infty^{\mathrm{path}}(T)$ of the classical version of Algorithm 8 is upper-bounded after $T$ steps, with probability at least $1 - \delta$, by*

$$\widetilde{O}\left( \min\left\{ \Lambda + 1, \frac{\Lambda}{1-\nu} \right\} \sqrt{\frac{|\mathcal{S}_n|A}{(1-\nu)^2} \frac{T \log T}{\log \frac{1}{\nu}} \log \frac{|\mathcal{S}_n|AT}{\delta}} + \frac{T(1+\Lambda)Ln^{-\alpha}}{1-\nu} \right),$$

*where $\widetilde{O}(\cdot)$ omits $\mathrm{poly}\log\log$ terms. If $\mathcal{X} = [0,1]^D$, in which case $|\mathcal{S}_n| = O(n^D)$, and setting $n = T^{\frac{1}{D+2\alpha}}$, the in-path regret is (for constant $\nu, L$)*

$$\widetilde{O}\left( \Lambda T^{\frac{D+\alpha}{D+2\alpha}} \sqrt{A \log T \log \frac{AT}{\delta}} \right).$$

*If $\mathcal{X} = \mathcal{S}$ is finite with size $S$, in which case $L = 0$, then the in-path regret is (for constant $\nu$)*

$$\widetilde{O}\left( \Lambda \sqrt{SAT \log T \log \frac{SAT}{\delta}} \right).$$

---

**Algorithm 8** Classical/quantum online-learning algorithm for infinite-horizon MDPs

---

**Input:** Compact state space $\mathcal{X}$ with $\frac{1}{n}$-net $\mathcal{S}_n$, finite action space $\mathcal{A}$, upper bound $\Lambda$ on optimal bias span, rewards $r : \mathcal{X} \times \mathcal{A} \to [0, 1]$, failure probability $\delta \in (0, 1)$, parameters $L, \alpha \geq 0$.

1: $t \leftarrow 1$ and $\tau_1 \leftarrow 1$
2: **for** episodes $k = 1, 2, \ldots$ **do**
    **Generative phase**
3:    **Classical:** By using Algorithm 5, choose decision rule $\widetilde{d}_k \in \mathcal{D}^{\mathrm{D}}$ such that, with failure probability at most $\frac{\delta}{8T^{5/4}}$ and for some constant $c > 0$,

$$g^{\widetilde{d}_k^\infty}(x) \geq g^* - c\Lambda\sqrt{\frac{|\mathcal{S}_n|A\log T}{\tau_k}\log\frac{|\mathcal{S}_n|AT}{\delta}} - c(1 + \Lambda)Ln^{-\alpha}$$

3:    **Quantum:** By using Algorithm 6, choose decision rule $\widetilde{d}_k \in \mathcal{D}^{\mathrm{D}}$ such that, with failure probability at most $\frac{\delta}{8T^{5/4}}$ and for some constant $c > 0$,

$$g^{\widetilde{d}_k^\infty}(x) \geq g^* - c\Lambda\frac{|\mathcal{S}_n|\sqrt{A}\log T}{\tau_k}\log^2\frac{|\mathcal{S}_n|AT}{\delta} - c(1 + \Lambda)Ln^{-\alpha}$$

    **Exploration phase**
4:    $\tau_{k+1} \leftarrow t$ and observe random initial state $x_t$
5:    **while** $t < 2\tau_{k+1}$ **do**
6:        Choose an action $a_t \sim \widetilde{d}_k(x_t)$ and obtain reward $r_t \leftarrow r(x_t, a_t)$
7:        Observe next state $x_{t+1} \sim p(\cdot|x_t, a_t)$
8:        $t \leftarrow t + 1$
9:    **end while**
10: **end for**

---

*Proof.* Let $t_k$ be the starting time of episode $k$ and $\tau_k = t_{k+1} - t_k$ the length of episode $k$. Given $s \in \mathcal{S}_n$, let $n_k(s, a) := |\{t_k \leq \tau < t_{k+1} : x_\tau \in \mathcal{X}(s), a_\tau = a\}|$ be the total number of visits in state-action pairs $(x, a)$ during (exploitation phase in) episode $k$ such that $x \in \mathcal{X}(s)$ and $n_k(s) := |\{t_k \leq \tau < t_{k+1} : x_\tau \in \mathcal{X}(s)\}| = \sum_{a \in \mathcal{A}} n_k(s, a)$. We shall abuse notation and let $n_k(x, a) := |\{t_k \leq \tau < t_{k+1} : (x_\tau, a_\tau) = (x, a)\}|$ be the total number of visits in state-action pairs $(x, a)$ during (exploitation phase in) episode $k$. This means that $n_k(x, a)$ is either zero or a Dirac delta depending on $(x, a) \in \mathcal{X} \times \mathcal{A}$ from the exploitation phase of episode $k$. We note that $\int_{x \in \mathcal{X}} \sum_{a \in \mathcal{A}} n_k(x, a)\mathrm{d}x = \sum_{s \in \mathcal{S}_n, a \in \mathcal{A}} n_k(s, a)$.

Let the in-path regret in episode $k$ be

$$\Delta_k := \int_{\mathcal{X}} n_k(\mathrm{d}x)\big(g^* - r_{\widetilde{d}_k}(x)\big).$$

Let $m$ denote the number of episodes. Since the policy $\widetilde{d}_k^\infty$ changes at most $\lceil\log_2 T\rceil$ times during $m$ episodes, $m \leq \lceil\log_2 T\rceil$. The in-path regret of Algorithm 8 is thus $\mathrm{Regret}_\infty^{\mathrm{path}}(T) = \sum_{k=1}^m \Delta_k$.

Now let $\widetilde{g}_k := \frac{1}{2}\big(\max_{x \in \mathcal{X}}\{u_{t^*+1}(x) - u_{t^*}(x)\} + \min_{x \in \mathcal{X}}\{u_{t^*+1}(x) - u_{t^*}(x)\}\big)$ be the output of Algorithm 5 that uses $O(\tau_k)$ calls to oracle $\mathcal{C}_p$. According to Theorem 20, with probability at least $1 - \frac{\delta}{8T^{5/4}}$,

$$\widetilde{g}_k \geq g^* - C\sqrt{\frac{\log_2 \tau_k}{\tau_k}} - \frac{13(1 + \Lambda)Ln^{-\alpha}}{1 - \nu}$$

$$\text{where } C = O\left(\min\left\{1 + \Lambda, \frac{\Lambda}{1 - \nu}\right\}\sqrt{\frac{|\mathcal{S}_n|A}{(1 - \nu)^2}\frac{1}{\log\frac{1}{\nu}}\log\frac{|\mathcal{S}_n|AT}{\delta}}\right),$$

which implies that

$$\Delta_k \leq \int_{\mathcal{X}} n_k(\mathrm{d}x)\big(\widetilde{g}_k - r_{\widetilde{d}_k}(x)\big) + \sum_{s \in \mathcal{S}_n} n_k(s)\left(\frac{13(1+\Lambda)Ln^{-\alpha}}{1-\nu} + C\sqrt{\frac{\log_2 \tau_k}{\tau_k}}\right)$$

$$= \int_{\mathcal{X}} n_k(\mathrm{d}x)\big(\widetilde{g}_k - r_{\widetilde{d}_k}(x)\big) + \tau_k\frac{13(1+\Lambda)Ln^{-\alpha}}{1-\nu} + C\sum_{s \in \mathcal{S}_n} n_k(s)\sqrt{\frac{\log_2 \tau_k}{\tau_k}}.$$

We first bound the term $\int_{\mathcal{X}} n_k(\mathrm{d}x)\big(\widetilde{g}_k - \sum_{a \in \mathcal{A}} r_{\widetilde{d}_k}(x)\big)$. Once $\mathrm{sp}(u_{t^*+1} - u_{t^*}) \leq \frac{3\varepsilon}{2} + \frac{18(1+\Lambda)Ln^{-\alpha}}{1-\nu}$ at iteration $t^*$ in Algorithm 5, then

$$|u_{t^*+1}(x) - u_{t^*}(x) - \widetilde{g}_k| \leq \frac{3C}{2}\sqrt{\frac{\log_2 \tau_k}{\tau_k}} + \frac{18(1+\Lambda)Ln^{-\alpha}}{1-\nu} \qquad \forall x \in \mathcal{X}.$$

We can expand

$$(\mathcal{L}u_{t^*})(x) = r_{\widetilde{d}_k}(x) + \int_{\mathcal{X}} p_{\widetilde{d}_k}(\mathrm{d}x'|x)u_{t^*}(x').$$

Together with the fact that $\|u_{t^*+1} - \mathcal{L}u_{t^*}\|_\infty \leq \frac{C(1-\nu)}{4}\sqrt{\frac{\log \tau_k}{\tau_k}} + 4(1+\Lambda)Ln^{-\alpha}$, this implies that, $\forall x \in \mathcal{X}$,

$$\left|\big(\widetilde{g}_k - r_{\widetilde{d}_k}(x)\big) - \left(\int_{\mathcal{X}} p_{\widetilde{d}_k}(\mathrm{d}x'|x)u_{t^*}(x') - u_{t^*}(x)\right)\right| \leq 2C\sqrt{\frac{\log_2 \tau_k}{\tau_k}} + \frac{22(1+\Lambda)Ln^{-\alpha}}{1-\nu},$$

and so

$$\Delta_k \leq \int_{\mathcal{X}} n_k(\mathrm{d}x)\left(\int_{\mathcal{X}} p_{\widetilde{d}_k}(\mathrm{d}x'|x)u_{t^*}(x') - u_{t^*}(x)\right) + \tau_k\frac{35(1+\Lambda)Ln^{-\alpha}}{1-\nu} + 3C\sum_{s \in \mathcal{S}_n} n_k(s)\sqrt{\frac{\log_2 \tau_k}{\tau_k}}. \tag{13}$$

Since $\int_{\mathcal{X}} p_{\widetilde{d}_k}(\mathrm{d}x'|x) = 1$ for all $x \in \mathcal{X}$, we can replace $u_{t^*}(x)$ in Eq. (13) with

$$w_k(x) := u_{t^*}(x) - \frac{1}{2}\left(\max_{x' \in \mathcal{X}}\{u_{t^*}(x')\} + \min_{x' \in \mathcal{X}}\{u_{t^*}(x')\}\right).$$

Note that $\mathrm{sp}(w_k) = \mathrm{sp}(u_{t^*}) \leq \min\{4\Lambda + 4, \frac{2\Lambda}{1-\nu}\}$ since all functions $(u_t)_{t \in \mathbb{N}}$ generated by Algorithm 5 have bounded bias span. Also, $\|w_k\|_\infty = \frac{1}{2}\mathrm{sp}(w_k) \leq \min\{2\Lambda + 2, \frac{\Lambda}{1-\nu}\}$. We now rewrite the first term in Eq. (13) as

$$\int_{\mathcal{X}} n_k(\mathrm{d}x)\left(\int_{\mathcal{X}} p_{\widetilde{d}_k}(\mathrm{d}x'|x)w_k(x') - w_k(x)\right)$$

$$= \sum_{t=t_k}^{t_{k+1}-1}\left(\int_{\mathcal{X}} p_{\widetilde{d}_k}(\mathrm{d}x'|x_t)w_k(x') - w_k(x_t)\right)$$

$$= w_k(x_{t_{k+1}}) - w_k(x_{t_k}) + \sum_{t=t_k}^{t_{k+1}-1}\left(\int_{\mathcal{X}} p_{\widetilde{d}_k}(\mathrm{d}x'|x_t)w_k(x') - w_k(x_{t+1})\right)$$

$$= w_k(x_{t_{k+1}}) - w_k(x_{t_k}) + \sum_{t=t_k}^{t_{k+1}-1} X_t.$$

Notice that the sequence $X_t := \int_{\mathcal{X}} p_{\widetilde{d}_k}(\mathrm{d}x|x_t)w_k(x) - w_k(x_{t+1})$ is a sequence of martingale differences. Since $|X_t| \leq 2\|w_k\|_\infty \leq \min\{4\Lambda + 4, \frac{2\Lambda}{1-\nu}\}$, by the Azuma-Hoeffding inequality,

$$\Pr\left[\sum_{t=1}^{T} X_t \geq \min\left\{4\Lambda + 4, \frac{2\Lambda}{1-\nu}\right\}\sqrt{2T\ln\frac{8T^{5/4}}{\delta}}\right] \leq \frac{\delta}{8T^{5/4}}.$$

Thus, we can bound the desired term as (already summing over all episodes)

$$\sum_{k=1}^{m} \int_{\mathcal{X}} n_k(\mathrm{d}x)\left(\int_{\mathcal{X}} p_{\widetilde{d}_k}(\mathrm{d}x'|x)w_k(x') - w_k(x)\right) \le \min\left\{4\Lambda + 4, \frac{2\Lambda}{1-\nu}\right\}\left(\lceil\log_2 T\rceil + \sqrt{2T\ln\frac{8T^{5/4}}{\delta}}\right)$$

with probability at least $1 - \frac{\delta}{8T^{5/4}}$.

Regarding $\sum_{k=1}^{m}\sum_{s\in\mathcal{S}_n} n_k(s)\sqrt{\frac{\log_2 \tau_k}{\tau_k}} \le \sqrt{2\lceil\log_2 T\rceil}\sum_{k=1}^{m}\sum_{s\in\mathcal{S}_n}\frac{n_k(s)}{\sqrt{t_k}}$ (since $\tau_k \ge \frac{t_k}{2}$), we can use Fact 1 in order to bound

$$\sum_{k=1}^{m}\sum_{s\in\mathcal{S}_n}\frac{n_k(s)}{\sqrt{t_k}} = \sum_{k=1}^{m}\frac{\sum_{s\in\mathcal{S}_n} n_k(s)}{\sqrt{\sum_{i=1}^{k-1}\sum_{s\in\mathcal{S}} n_k(s)}} \le \frac{\sqrt{T}}{\sqrt{2}-1}.$$

Putting everything together, with probability at least $1 - \frac{\delta}{4T^{5/4}}$, the total in-path regret is at most

$$2\min\left\{4\Lambda + 4, \frac{2\Lambda}{1-\nu}\right\}\sqrt{2T\ln\frac{8T^{5/4}}{\delta}} + \frac{3\sqrt{2}C}{\sqrt{2}-1}\sqrt{T\lceil\log_2 T\rceil} + T\frac{35(1+\Lambda)Ln^{-\alpha}}{1-\nu}.$$

Finally, since $\sum_{T=1}^{\infty}\frac{\delta}{4T^{5/4}} < \delta$, the above bound is valid for all $T \in \mathbb{N}$. $\qquad\square$

We now prove upper bounds on the in-path regret for the quantum version of Algorithm 8, where the choice for an approximate optimal policy is done quantumly via Algorithm 6.

**Theorem 25** (Quantum infinite-horizon in-path regret bound). *Let $M = \langle\mathcal{X}, \mathcal{A}, p, r\rangle$ be an infinite-horizon weakly communicating MDP with $\mathrm{sp}(h^*) \le \Lambda$. Let $\mathcal{S}_n$ be an $\frac{1}{n}$-net for $\mathcal{X}$. Assume that $p$ and $r$ satisfy Assumption 10 with parameters $L, \alpha \ge 0$. Assume the optimal Bellman operator $\mathcal{L}$ of $M$ is an 1-stage $\nu$-span contraction for $\nu \in [0,1)$. For any $T, n \in \mathbb{N}$, the in-path regret $\mathrm{Regret}_{\infty}^{\mathrm{path}}(T)$ of the quantum version of Algorithm 8 is upper-bounded after $T$ steps, with probability at least $1 - \delta$, by*

$$\widetilde{O}\left(\min\left\{1+\Lambda, \frac{\Lambda}{1-\nu}\right\}\left(\sqrt{T\log\frac{T}{\delta}} + \frac{|\mathcal{S}_n|\sqrt{A}}{1-\nu}\frac{\log^2 T}{\log\frac{1}{\nu}}\log\frac{|\mathcal{S}_n|AT}{\delta}\log\frac{|\mathcal{S}_n|T}{\delta}\right) + \frac{T(1+\Lambda)Ln^{-\alpha}}{1-\nu}\right),$$

*where $\widetilde{O}(\cdot)$ omits* poly log log *terms. If $\mathcal{X} = [0,1]^D$, in which case $|\mathcal{S}_n| = O(n^D)$, and setting $n = T^{\frac{1}{D+\alpha}}$, the in-path regret is (for constant $\nu, L$)*

$$\widetilde{O}\left(\Lambda\sqrt{T\log\frac{T}{\delta}} + \Lambda T^{\frac{D}{D+\alpha}}\sqrt{A}\log^2 T\log\frac{AT}{\delta}\log\frac{T}{\delta}\right).$$

*If $\mathcal{X} = \mathcal{S}$ is finite with size $S$, in which case $L = 0$, then the in-path regret is (for constant $\nu$)*

$$\widetilde{O}\left(\Lambda\sqrt{T\log\frac{T}{\delta}} + \Lambda S\sqrt{A}\log^2 T\log\frac{SAT}{\delta}\log\frac{ST}{\delta}\right).$$

*Proof.* The proof is similar to Theorem 24, so we shall point out the main changes. Once again, let $\widetilde{g}_k := \frac{1}{2}\left(\max_{x\in\mathcal{X}}\{u_{t^*+1}(x) - u_{t^*}(x)\} + \min_{x\in\mathcal{X}}\{u_{t^*+1}(x) - u_{t^*}(x)\}\right)$ be the output of Algorithm 6 that uses $O(\tau_k)$ calls to oracle $\mathcal{O}_p$ and $\mathcal{O}_p^{\dagger}$. According to Theorem 21, with probability at least $1 - \frac{\delta}{8T^{5/4}}$,

$$\widetilde{g}_k \ge g^* - C\frac{\log_2 \tau_k}{\tau_k} - \frac{13(1+\Lambda)Ln^{-\alpha}}{1-\nu}$$

where $C = O\left(\min\left\{1+\Lambda, \frac{\Lambda}{1-\nu}\right\}\frac{|\mathcal{S}_n|\sqrt{A}}{1-\nu}\frac{1}{\log\frac{1}{\nu}}\log\frac{|\mathcal{S}_n|AT}{\delta}\log\frac{|\mathcal{S}_n|T}{\delta}\right),$

which implies that

$$\Delta_k \le \int_{\mathcal{X}} n_k(\mathrm{d}x)\left(\widetilde{g}_k - r_{\widetilde{d}_k}(x)\right) + \tau_k\frac{13(1+\Lambda)Ln^{-\alpha}}{1-\nu} + C\sum_{s\in\mathcal{S}_n} n_k(s)\frac{\log_2 \tau_k}{\tau_k}.$$

We can follow the same steps as the proof of Theorem 24 (approximate $\widetilde{g}_k$ with $u_{t^*+1}(x) - u_{t^*}(x)$) to obtain the bound

$$\Delta_k \leq \int_{\mathcal{X}} n_k(\mathrm{d}x) \left( \int_{\mathcal{X}} p_{\widetilde{d}_k}(\mathrm{d}x'|x) u_{t^*}(x') - u_{t^*}(x) \right) + \tau_k \frac{35(1+\Lambda)Ln^{-\alpha}}{1-\nu} + 3C \sum_{s \in \mathcal{S}_n} n_k(s) \frac{\log_2 \tau_k}{\tau_k}. \tag{14}$$

By replacing $u_{t^*}(x)$ with $w_k(x) := u_{t^*}(x) - \frac{1}{2}(\max_{x' \in \mathcal{X}}\{u_{t^*}(x')\} + \min_{x' \in \mathcal{X}}\{u_{t^*}(x')\})$, we can once again, by the Azuma-Hoeffding inequality, bound the first term in Eq. (14) with probability at least $1 - \frac{\delta}{8T^{5/4}}$ as (already summing over all episodes)

$$\sum_{k=1}^{m} \int_{\mathcal{X}} n_k(\mathrm{d}x) \left( \int_{\mathcal{X}} p_{\widetilde{d}_k}(\mathrm{d}x'|x) w_k(x') - w_k(x) \right) \leq \min\left\{ 4\Lambda + 4, \frac{2\Lambda}{1-\nu} \right\} \left( \lceil \log_2 T \rceil + \sqrt{2T \ln \frac{8T^{5/4}}{\delta}} \right).$$

Regarding the term $\sum_{k=1}^{m} \sum_{s \in \mathcal{S}_n} n_k(s) \frac{\log_2 \tau_k}{\tau_k} \leq 2\lceil \log_2 T \rceil \sum_{k=1}^{m} \sum_{s \in \mathcal{S}_n} \frac{n_k(s)}{t_k}$ (since $\tau_k \geq \frac{t_k}{2}$), we can use Lemma 2 to obtain

$$\sum_{k=1}^{m} \sum_{s \in \mathcal{S}} \frac{n_k(s)}{t_k} = \sum_{k=1}^{m} \frac{\sum_{s \in \mathcal{S}} n_k(s)}{\sum_{i=1}^{k-1} \sum_{s \in \mathcal{S}} n_k(s)} \leq 4 \log_2 T.$$

This means that, with probability at least $1 - \frac{\delta}{4T^{5/4}}$, the total in-path regret is

$$\mathrm{Regret}_{\infty}^{\mathrm{path}}(T) \leq 2 \min\left\{ 4\Lambda + 4, \frac{2\Lambda}{1-\nu} \right\} \sqrt{2T \ln \frac{8T^{5/4}}{\delta}} + 24C\lceil \log_2 T \rceil^2 + T \frac{35(1+\Lambda)Ln^{-\alpha}}{1-\nu}.$$

Finally, since $\sum_{T=1}^{\infty} \frac{\delta}{4T^{5/4}} < \delta$, the above bound is valid for all $T \in \mathbb{N}$. $\qquad \square$

### F.2 Expected Regret

We now turn our attention to the expected regret $\mathrm{Regret}_{\infty}^{\mathbb{E}}(T)$ of Algorithm 8 instead of its in-path regret. Fortunately, the analysis is much simpler in this case, similarly to the finite-horizon case in Theorems 22 and 23.

**Theorem 26** (Classical infinite-horizon expected regret bound). *Let $M = \langle \mathcal{X}, \mathcal{A}, p, r \rangle$ be an infinite-horizon weakly communicating MDP with $\mathrm{sp}(h^*) \leq \Lambda$. Let $\mathcal{S}_n$ be an $\frac{1}{n}$-net for $\mathcal{X}$. Assume that $p$ and $r$ satisfy Assumption 10 with parameters $L, \alpha \geq 0$. Assume the optimal Bellman operator $\mathcal{L}$ of $M$ is an 1-stage $\nu$-span contraction for $\nu \in [0, 1)$. For any $T, n \in \mathbb{N}$, the expected regret $\mathrm{Regret}_{\infty}^{\mathbb{E}}(T)$ of the classical version of Algorithm 8 is upper-bounded after $T$ steps, with probability at least $1 - \delta$, by*

$$\widetilde{O}\left( \min\left\{ \Lambda + 1, \frac{\Lambda}{1-\nu} \right\} \sqrt{\frac{|\mathcal{S}_n|A}{(1-\nu)^2} \frac{T \log T}{\log \frac{1}{\nu}} \log \frac{|\mathcal{S}_n|AT}{\delta}} + \frac{T(1+\Lambda)Ln^{-\alpha}}{1-\nu} \right),$$

*where $\widetilde{O}(\cdot)$ omits $\mathrm{poly} \log \log$ terms. If $\mathcal{X} = [0, 1]^D$, in which case $|\mathcal{S}_n| = O(n^D)$, and setting $n = T^{\frac{1}{D+2\alpha}}$, the expected regret is (for constant $\nu, L$)*

$$\widetilde{O}\left( \Lambda T^{\frac{D+\alpha}{D+2\alpha}} \sqrt{A \log T \log \frac{AT}{\delta}} \right).$$

*If $\mathcal{X} = \mathcal{S}$ is finite with size $S$, in which case $L = 0$, then the expected regret is (for constant $\nu$)*

$$\widetilde{O}\left( \Lambda \sqrt{SAT \log T \log \frac{SAT}{\delta}} \right).$$

*Proof.* The total expected regret after $T$ time steps spread over $m \leq \lceil \log_2 T \rceil$ episodes is

$$\mathrm{Regret}_{\infty}^{\mathbb{E}}(T) = \sum_{k=1}^{m} \tau_k \left( g^* - \min_{x \in \mathcal{X}} g^{\widetilde{d}_k^{\infty}}(x) \right),$$

where $\widetilde{d}_k \in \mathcal{D}^{\mathrm{D}}$ is the decision rule employed in episode $k$ and $\tau_k$ is the length of episode $k$. According to Theorem 20, the output policy $\widetilde{d}_k^\infty \in \Pi^{\mathrm{D}}$ of Algorithm 5 using $O(\tau_k)$ calls to oracle $\mathcal{C}_p$ is such that, with probability at least $1 - \frac{\delta}{4T^{5/4}}$, for all $x \in \mathcal{X}$,

$$g^{\widetilde{d}_k^\infty}(x) \geq g^* - C\sqrt{\frac{\log_2 \tau_k}{\tau_k}} - \frac{26(1+\Lambda)Ln^{-\alpha}}{1-\nu},$$

$$\text{where } C = O\left(\min\left\{1+\Lambda, \frac{\Lambda}{1-\nu}\right\}\sqrt{\frac{|\mathcal{S}_n|A}{(1-\nu)^2}\frac{1}{\log\frac{1}{\nu}}\log\frac{|\mathcal{S}_n|AT}{\delta}}\right),$$

which implies that $\mathrm{Regret}_\infty^{\mathbb{E}}(T)$ is upper-bounded by

$$\sum_{k=1}^m \left(C\sqrt{\tau_k \log_2 \tau_k} + \tau_k \frac{26(1+\Lambda)Ln^{-\alpha}}{1-\nu}\right) \leq \frac{\sqrt{2}C}{\sqrt{2}-1}\sqrt{T\lceil \log_2 T\rceil} + T\frac{26(1+\Lambda)Ln^{-\alpha}}{1-\nu},$$

using Fact 1 in order to bound $\sum_{k=1}^m \sqrt{\tau_k \log_2 \tau_k} \leq \sqrt{2\lceil \log_2 T\rceil}\sum_{k=1}^m \frac{\tau_k}{\sqrt{t_k}} \leq \frac{\sqrt{2}}{\sqrt{2}-1}\sqrt{T\lceil \log_2 T\rceil}$.

Finally, since $\sum_{T=1}^\infty \frac{\delta}{4T^{5/4}} < \delta$, the above bound is valid for all $T \in \mathbb{N}$. $\qquad\square$

**Theorem 27** (Quantum infinite-horizon expected regret bound). *Let $M = \langle \mathcal{X}, \mathcal{A}, p, r\rangle$ be an infinite-horizon weakly communicating MDP with $\mathrm{sp}(h^*) \leq \Lambda$. Let $\mathcal{S}_n$ be an $\frac{1}{n}$-net for $\mathcal{X}$. Assume that $p$ and $r$ satisfy Assumption 10 with parameters $L, \alpha \geq 0$. Assume the optimal Bellman operator $\mathcal{L}$ of $M$ is an 1-stage $\nu$-span contraction for $\nu \in [0,1)$. For any $T, n \in \mathbb{N}$, the expected regret $\mathrm{Regret}_\infty^{\mathbb{E}}(T)$ of the quantum version of Algorithm 8 is upper-bounded after $T$ steps, with probability at least $1 - \delta$, by*

$$\widetilde{O}\left(\min\left\{1+\Lambda, \frac{\Lambda}{1-\nu}\right\}\frac{|\mathcal{S}_n|\sqrt{A}}{1-\nu}\frac{\log^2 T}{\log\frac{1}{\nu}}\log\frac{|\mathcal{S}_n|AT}{\delta}\log\frac{|\mathcal{S}_n|T}{\delta} + \frac{T(1+\Lambda)Ln^{-\alpha}}{1-\nu}\right),$$

*where $\widetilde{O}(\cdot)$ omits $\mathrm{poly}\log\log$ terms. If $\mathcal{X} = [0,1]^D$, in which case $|\mathcal{S}_n| = O(n^D)$, and setting $n = T^{\frac{1}{D+\alpha}}$, the expected regret is (for constant $\nu, L$)*

$$\widetilde{O}\left(\Lambda T^{\frac{D}{D+\alpha}}\sqrt{A}\log^2 T \log\frac{AT}{\delta}\log\frac{T}{\delta}\right).$$

*If $\mathcal{X} = \mathcal{S}$ is finite with size $S$, in which case $L = 0$, then the expected regret is (for constant $\nu$)*

$$\widetilde{O}\left(\Lambda S\sqrt{A}\log^2 T \log\frac{SAT}{\delta}\log\frac{ST}{\delta}\right).$$

*Proof.* The proof is basically the same as Theorem 26, the main difference being that now we employ Algorithm 6 with $O(\tau_k)$ calls to oracle $\mathcal{O}_p$ in order to obtain a stationary policy $\widetilde{d}_k^\infty \in \Pi^{\mathrm{D}}$ such that, according to Theorem 21, with probability at least $1 - \frac{\delta}{4T^{5/4}}$, for all $x \in \mathcal{X}$,

$$g^{\widetilde{d}_k^\infty}(x) \geq g^* - C\frac{\log_2 \tau_k}{\tau_k} - \frac{26(1+\Lambda)Ln^{-\alpha}}{1-\nu},$$

$$\text{where } C = O\left(\min\left\{1+\Lambda, \frac{\Lambda}{1-\nu}\right\}\frac{|\mathcal{S}_n|\sqrt{A}}{1-\nu}\frac{1}{\log\frac{1}{\nu}}\log\frac{|\mathcal{S}_n|AT}{\delta}\log\frac{|\mathcal{S}_n|T}{\delta}\right),$$

which implies that

$$\mathrm{Regret}_\infty^{\mathbb{E}}(T) \leq \sum_{k=1}^m \left(C\log_2 \tau_k + \tau_k\frac{26(1+\Lambda)Ln^{-\alpha}}{1-\nu}\right) \leq C\lceil \log_2 T\rceil^2 + T\frac{26(1+\Lambda)Ln^{-\alpha}}{1-\nu}.$$

Since $\sum_{T=1}^\infty \frac{\delta}{4T^{5/4}} < \delta$, the above bound is valid for all $T \in \mathbb{N}$. $\qquad\square$

