# OpenReview forum: "A Bit of Freedom Goes a Long Way: Classical and Quantum Algorithms for Reinforcement Learning under a Generative Model"
_ICLR.cc/2026/Conference — Submitted to ICLR 2026_

### Official Review · Reviewer_qj3A · 2025-10-20

**Soundness:** 4
**Presentation:** 3
**Contribution:** 2
**Rating:** 6
**Confidence:** 3

**Summary:**

The manuscript deals with regret bounds of online RL in MDPs with finite horizons and MDPs with infinite horizons. Classical and quantum algorithms in the special case with oracles are considered. This is a purely theoretical work, covering a total of 49 pages.

**Strengths:**

* Very careful preparation with comprehensive discussion of the literature.
* Innovative approach to this established research topic.

Further comments:\
Although it is unusual to mention selected publications in the abstract, I think this is good here because it makes it very concrete and carefully presented.

The use of many footnotes is rather unusual. However, I think it fits in with the very careful style of the work.

One could criticize the enormous length of the appendix, but I think it is well done, because the main text can be understood well without the appendix, and the appendix provides useful additional information.

**Weaknesses:**

I can't identify any clear weaknesses.

Perhaps your own contribution could be mentioned in one central place and described in more concrete terms there.

Further comments:

In “One of the most famous measures is that of regret,” I don't like the word “famous.”

Similarly, in “The secret of the improved performance,” I don't like the word “secret.”

It's probably a matter of taste, but I think “Conclusion” is more accurate than “Conclusions.”

„a RL“ -> „an RL“

**Questions:**

* Which relevance do the results have for further research?
* Which relevance could the results potentially have for future applications?

---

> ### Author Response · Authors · 2025-11-21
>
> We thank the reviewer for the useful comments! We are happy to drop words like "famous" and "secret" and change "conclusion" to "conclusions".
>
> We believe that our results push the knowledge of what quantum computers can and cannot do regarding RL. More precisely, they give evidence that quantum computers can learn much faster than classical computers in the right setting, which helps to decide when we should employ quantum computers and when we should not. Moreover, "how faster can one learn" also depends on the measure considered, as we have shown that a substantial learning advantage can be obtained in one (novel) measure of regret but not necessarily in another. Finally, our results also place further emphasis on the open question of a quantum RL model without a generative phase.

---

### Official Review · Reviewer_zBSn · 2025-10-29

**Soundness:** 4
**Presentation:** 3
**Contribution:** 2
**Rating:** 2
**Confidence:** 4

**Summary:**

This paper proposes quantum and classical algorithms for online reinforcement learning in both finite-horizon and infinite-horizon MDPs. The authors design quantum versions of variance-reduced backward induction and robust value iteration, achieving query complexities that improve the best classical bounds by factors of $\sqrt{A}$ and smaller H-dependence. They introduce an online “exploration–generative” protocol and prove regret bounds that are minimax-optimal for classical algorithms and enjoy an exponential-in-T advantage for a newly defined expected regret when quantum oracles are used. Results are extended to continuous state spaces via Hölder-smooth discretization.

**Strengths:**

1.	The paper provides solid theoretical proofs for its results.
2.	For the online learning problem of MDPs, when we introduce quantum algorithm, it is difficult to define proper regret. The authors introduce a novel model with classical exploration phases and classical/quantum generative phases to solve this difficulty.

**Weaknesses:**

1.	In section 2 (to compute optimal policies), the authors consider undiscounted version of MDPs (both finite-horizon and infinite-horizon). In my understanding, the discounted version is more important and the quantum algorithm in this version has already been proposed. The motivation and technique challenges of the undiscounted version are not clearly explained.
2.	In section 3 (online learning version), the authors introduce a novel model which splits the interaction into two types of phases: classical exploration phases and classical/quantum generative phases. Although this idea solves the difficulty to define proper regret of the quantum algorithm, it undermines the most critical challenge in online learning: the algorithm must balance exploration and exploitation. This significantly reduces the problem's difficulty. I don’t think it is reasonable.
3.	In section 3, the authors require that the length of generative phase is at most $O(\tau)$ where $\tau$ is the length of the previous exploration phase. We must have some limitations of generative phase, but such a restriction appears arbitrary and lacks justification.

**Questions:**

1.	Related to weakness 1: Is there any special motivation to consider quantum algorithm for finite-horizon and infinite-horizon undiscounted MDPs? Comparing to the quantum algorithms in previous work for infinite-horizon discounted MDPs, what is the main technique challenges for undiscounted version?
2.	Related to weakness 2: In practical scenarios, is there any motivation for proposing such model?
3.	Related to weakness 3: why should we require that the length of generative phase is at most $O(\tau)$？

---

> ### Author Response · Authors · 2025-11-21
>
> We thank the reviewer for all the useful comments, which will be helpful in improving our paper. We appreciate the high score on soundness! Below we try to address all the reviewer's comments.
>
> Question 1.
>
> The discounted setting is one of the simplest cases. It is no wonder then that quantum algorithms have already been proposed. Actually, quantum algorithms have ONLY been proposed for the discounted setting due to its simplicity, with the exception of Luo et al. (2025), which appeared online around the same time as ours. Finite-horizon and infinite-horizon, specially the latter, are much more complicated and require finer analyses. We would then like to argue that a substantial part of the classical literature on MDPs have been focused on understanding the infinite-horizon setting, with important works like Auer et al. (2008), Bartlett \& Tewari (2009), Lakshmanan et al. (2015), Fruit et al. (2018). Moreover, we believe it to be unfair to dismiss any interesting result on the undiscounted setting for the sole reason of not being the discounted setting. If that were the case, then any new result on undiscounted MDPs, classical or quantum, should be dismissed.
>
> Regarding the technique challenges, let's start with the finite-horizon setting. The optimal value iteration algorithm from Sidford et al. (2018a) (Algorithm 2) must compute $H$ different quantities at the beginning of each new epoch, in contrast to the discounted setting, where only one quantity must be computed. This is not a problem in the classical case, as samples can be reused, but this is absolutely not the case in the quantum setting. Quantum samples cannot be reused. This (naively) incurs into an extra $H$ factor in the sample complexity. In order to improve this factor, we employ an advanced technique called quantum multivariate mean estimation subroutine. The result is a $O(\sqrt{H})$ overhead. Still, it is an open question whether this factor can be removed.
>
> On the other hand, the main technique challenges regarding the infinite-horizon setting is the setting itself: the definition of average reward is more involved (which led us to define a new measure of regret!), the contraction properties and chain structure of the underlying MDP must usually be imposed. There is no big challenge in quantising the classical value iteration algorithm apart from employing quantum minimum finding and quantum mean estimation. More effort was put in improving the dependence on $\Lambda$ and $\nu$, hence the presence of a minimum in the sample complexity of Theorems 20 and 21. The main technique challenge comes in Appendix F, where we show that the choice for regret is vital in the quantum case, a result which we find to be very interesting.
>
> Question 2.
>
> We'll try to argue in favor of our model. The reviewer is right in that splitting the interaction solves the difficulty to define proper quantum regret, but this seems the only way out so far. It is a good open question whether it is possible to define a meaningful and useful quantum RL model without a generative phase. ALL previous quantum works had to assume a regret-free phase in order to obtain a $polylog(T)$ regret. Our model, however, does not undermine the exploration-exploitation balance. This is achieved by imposing that the length of generative phase be at most $O(\tau)$. The challenge of whether the agent should prolong the exploration phase in order to obtain a better policy down the line or switch to the generative phase earlier to get an earlier, but less improved, policy is still present.
>
> The main reason for such a model is to test the limits of quantum computers' power over classical computers in RL models. Our paper answers this question. Having access to short periods of free (offline) interaction can have a huge impact on the learning process when using a quantum computer, while the same is not true using a classical computer (our classical regrets are similar to the standard regrets from the literature). In a more practical scenario, our exploration-generative RL model could describe the interaction between an agent and an unknown (black-box) quantum computer in an online-fashion wherein the agent has limited access to the quantum computer due to, e.g. high traffic or monetary reasons, interspersed with periods of free interaction due to the quantum computer being idle.
>
> Question 3.
>
> It seemed natural to equate the length of two consecutive exploration and generative phases, but it is possible to consider more general relations as briefly mentioned at the end of the conclusions. As written, "another direction is to explore variants of our RL model, e.g. allowing more or less samples to the oracles than $O(\tau)$. It is not hard to see that allowing $O(\tau^2)$ oracle calls during generative phases would lead to a $polylog(T)$ regret also for classical algorithms, while $O(\tau^c)$ oracle calls for $c<1$ would reinflate the dependence on $T$ to polynomial for quantum algorithms."

---

### Official Review · Reviewer_Mu5H · 2025-10-30

**Soundness:** 2
**Presentation:** 3
**Contribution:** 2
**Rating:** 4
**Confidence:** 3

**Summary:**

The authors propose both classical and quantum online algorithms for reinforcement learning (RL) tasks. These algorithms are rooted in an exploration-generative RL framework, wherein the agent can occasionally interact freely with the environment by utilizing access to a simulator of the environment. The authors claim that their proposed algorithms address and mitigate common RL challenges, such as “optimism in the face of uncertainty” and “posterior sampling.”

**Strengths:**

The paper is well-written and engaging. The discussed related work is extensive and provides a good overview on similar research.

**Weaknesses:**

However, the applicability of the proposed algorithms is strictly confined to RL scenarios where a generative model with comprehensive knowledge of the state and action spaces—as well as the reward function—is available. Moreover, it assumes the transition probabilities can be queried through an oracle. Such oracles are not novel in RL research and are frequently associated with methods categorized as model-based RL. Past research has already explored quantum oracles extensively, particularly focusing on achieving quantum advantage via quantum sampling.

A key point of concern is the lack of clarity regarding the benefits of separating the exploration phase (classical) from the policy learning phase (classical/quantum). If the agent has access to a flawless oracle, it is unclear why a classical exploration on the MDP (Markov Decision Process) would be necessary. Additionally, the rationale behind limiting access to the oracle (referred to as “the true MDP”) needs to be elaborated. For the paper to be more impactful, it should include a clear justification for the algorithm’s design choices as well as an explanation of how, in practical settings, the oracles representing the true MDP might be obtained.

Minor Issues:
The citation format in the abstract, “Ganguly et al. (arXiv’23) and Zhong et al. (ICML’24),” is unconventional and should follow standard citation styles.
Page 9: There is a typo in “a RL.” It should be corrected to “an RL.”

**Questions:**

Why is the access to the oracle limited?
Why is the oracle not used to learn an optimal policy offline?
How can in practical settings oracles representing the true MDP be obtained?

---

> ### Author Response · Authors · 2025-11-21
>
> We thank the reviewer for all the useful comments, specially about our paper being well-written and engaging and the related work being extensive. Below we try to address all of the reviewer's comments
>
> Addressing main weaknesses:
>
> We cordially disagree that past research has already explored quantum oracles extensively. The opposite! For computing optimal policies in a generative fashion, we are only aware of a few works, all for infinite-horizon discounted MDPs, as mentioned in the introduction. The only work on finite-horizon MDPs is Luo et al. "Quantum algorithms for finite-horizon Markov decision processes", which appeared online around the same time as ours. Finally, there are no quantum works on infinite-horizon average-reward MDPs apart from ours, which was even praised by reviewer 6JTM.
>
> Regarding the actual RL model of online learning MDPs, our results on finite horizon improves upon the works of Ganguly et al. (2023) and Zhong et al. (2024). On the other hand, our work is also the first to consider the online learning of infinite-horizon average-reward MDPs. Finally, we are the first to tackle to continuous state space setting. We would then like to argue that the quantum RL landscape has been poorly explored so far.
>
> We thank the feedback about the lack of clarity regarding our model. We will try to make the following clearer in the paper. We would like to point the reviewer to our answers to the other reviewers, but we will try to clarify our model here as clearly as possible. The most important point is: there is currently NO meaningful and interesting quantum RL model that does not separate the exploration phase from the policy learning phase! As discussed in the introduction and Appendix B, ALL previous quantum RL models had to assume a phase with no incurred regret in order to learn a policy more efficiently. The existence of a quantum RL model without a generative (or offline) phase that yields a $polylog(T)$ regret is actually a great open problem! So one reason for separating the phases is due to necessity. Another reason is to explore the extend of quantum computers' power. As shown by our paper, having access to short periods of free (offline) interaction can have a huge impact on the learning process when using a quantum computer, while the same is not true when using a classical computer (our classical regrets are similar to the standard regrets from the literature). We deem such a result as an important difference between classical and quantum computers in RL.
>
> Regarding how the oracles behind the MDP could be obtained, if the classical RL environment can be viewed as a computer program with a source code, then it is possible to transform such a source code into a quantum oracle: we simply translate the source code into a Boolean circuit that draws samples from the distribution $p(\cdot|x,a)$ and employ standard techniques in quantum computing to convert such a Boolean circuit into a quantum circuit implementing the oracles from our paper. Our exploration-generative RL model could then describe the interaction between an agent and a (black-box) quantum computer in an online-fashion wherein the agent has limited access to the quantum computer due to, e.g., high traffic or monetary reasons, interspersed with periods of free interaction due to the quantum computer being idle.
>
> Questions:
>
> 1. To expand on what has already been said, the access to the oracle is limited to ensure a meaningful learning model, otherwise the agent could query the oracle as many times as possible to learn a policy that is as close as possible to optimal and incur a regret as low as desired.
>
> 2. In some sense, we could argue that the policy is actually learnt offline during the generative phase, at least up to a limit imposed by the previous exploration phase (the reason for such a limit was answered above). We could have equivalently named the "generative phase" as an "offline phase", and the "exploration phase" as an "online phase".
>
> 3. This was answered above.

---

### Official Review · Reviewer_6JTM · 2025-11-10

**Soundness:** 3
**Presentation:** 2
**Contribution:** 2
**Rating:** 4
**Confidence:** 3

**Summary:**

In the context of quantum RL, this paper studies both computing near-optimal policies under a generative model for (generalized) tabular MDPs, and
online reinforcement learning (regret minimization) in finite-horizon and infinite-horizon average-reward MDPs. The basic setting for this generalized tabular MDP is that the state space $\mathcal{X}$, although continuous, can be approximated with an $\epsilon$-net by the Holder continuous condition, reducing the problem to a standard tabular RL.

For the generative model setting, this paper proposes a novel algorithm with a quantum maximum finding subroutine based on the classical value iteration algorithm, which reduces the sample complexity from $O(A)$ to $O(\sqrt{A})$ in the finite horizon MDP. The usage of quantum mean estimation also provides a quadratic improvement with regards to $\epsilon$, which has been investigated in the literature [1]. For infinite-horizon average-reward MDP, this paper revises the classical (extended) value iteration in favor of quantum mean estimation, and achieves a sample complexity of $\tilde{O}(\Lambda |\mathcal{S}_n| \sqrt{A} / (1 - \nu) \epsilon)$ in the case that the Bellman operator is a $\nu$-contraction.

For the general online exploration setting, this paper studies the regret minimization with a new formulation. The whole exploration process is divided into two phase: the exploration phase and the generative phase. The agent is allowed to interact *freely* with the quantum MDP in the generative phase, as if there is a generative model and without incurring any regret. In the exploration phase, the agent executes current policy and incurs corresponding regret. The total length of a generative phase must be the same order of previous exploration phase. This paper proposes a unified algorithm for both finite horizon MDP and infinite horizon MDP, where the agent uses the algorithms of generative model setting to approximate the optimal policy in the generative phase, and the execute this policy in the exploration phase. The regret for the finite horizon setting is $\tilde{O}(\min(HSA, H^2S\sqrt{A} \log T))$, improving on the order of $S, A$ of previous works. The regret for the infinite horizon setting is $\tilde{O}(\Lambda \sqrt{T} + \Lambda S \sqrt{A} \log^2 T)$, where the term $\Lambda \sqrt{T}$ can be omitted if an expected regret is used so as to achieve the $\mathrm{poly} \log T$ regret.

[1]. Zhong et al., Provably efficient exploration in quantum reinforcement learning with logarithmic worst-case regret.

**Strengths:**

1. The application of quantum computing (e.g., the quantum mean estimation subroutine) to online exploration of RL is an interesting and important problem. This paper pushes the boundary of this problem.

2. As far as I know, this is the first paper investigating quantum RL in the setting of infinite-horizon average-reward MDP. It shows that the sample complexity can also be improved quadratically with the help of quantum mean estimation.

3. The idea of applying quantum maximum finding subroutine in the hoeffding-style value iteration algorithm is novel, saving a $\sqrt{A}$ in the regret bound and sample complexity.

**Weaknesses:**

1. There is a major problem in the formulation of this "exploration-generation" two-phase procedure, which assumes that the agent can use the oracles as *a generative model* in the generation phase *without incurring any regret*. There are indeed many works in the literature of classical RL using this idea of "lazy update" to design sample-efficient algorithms such as [1, 2], but none of these works assume the access to a generative model nor assume the data collection phase incurs no regret. They can use this lazy update because *they use the data from the "exploration" phase to estimate the value function directly*, instead of collecting new data to estimate the value function in a new "generation" phase. In this work, however, the agent is able to use a generative model to collect grand new data in the generation phase, which means the agent is able to explore the MDP in an **arbitrary** state-action distribution. On the other hand, the core assumption of online RL for unknown environments is that the agent should create favorable state-action distribution themselves by taking good policies to find the high-rewarded state and action. The basic assumption here is, you can **NOT** collect any data from the states that you never know how to get there in an online exploration problem (e.g., see all of the references from line 365 to 367 in this paper), which is also the case for any real-world tasks. Therefore, there is a large gap between the real online exploration RL and the two-phase model proposed in this paper. All the results of Section 3 shall be RL with a generative model instead of online exploration.

2. For the results of the quantum algorithms in the infinite horizon MDP, there is an extra contraction measure $\nu$ of the Bellman operator in the sample complexity, which does not appear in classical RL literature like current SOTA [5]. The sample complexity bound has a $(1 - \nu)^{-2}$ dependency for the classical setting and a dependency of $(1 - \nu)^{-1}$ for the quantum setting. This term mainly comes from the pipeline of value iteration of Algorithm 5. What is the reason to introduce $\nu$ here? Is it because if one uses a conventional decomposition of regret like [6] then an extra martingale term of $O(\sqrt{T})$ will appear? This contraction term somehow makes it harder to evaluate the sample complexity bound of Result 2.

[1]. Zhong et al., Provably efficient exploration in quantum reinforcement learning with logarithmic worst-case regret.

[2]. Jacksch et al., Near-optimal Regret Bounds for Reinforcement Learning.

[3]. Fruit et al., Efficient Bias-Span-Constrained Exploration-Exploitation in Reinforcement Learning.

[4]. Ayoub et al., Model-Based Reinforcement Learning with Value-Targeted Regression.

[5]. Zhang et al., Sharper Model-free Reinforcement Learning for Average-reward Markov Decision Processes.

[6]. Bartlett et al., REGAL: A Regularization based Algorithm for Reinforcement Learning in Weakly Communicating MDPs.

**Questions:**

1. What is the data collected in the exploration phase used for in Algorithm 1?

2. As the essential setting of this paper is tabular MDP, why is the paper using a continuous state space $\mathcal{X}$ with a $1/n$-covering $|\mathcal{S}_n|$ but all of the bounds (except for $\mathcal{X} = [0, 1]^D$) depends on $|\mathcal{S}_n|$? It is quite a common sense that one can use the $\epsilon$-net trick to apply the results of tabular RL to a continuous space, but this does not lead to any improvement since the size of the $\epsilon$-net has the same order as the original space $\mathcal{X}$. The analysis for the discretizaiton error of $\mathcal{S}_n$ in the Holder continuity seems to only overly complicate the paper. A right example here is the Theorem 1 of [4], which uses a log-covering number as the complexity measure.

3. Is is possible to use the reduction from an average-reward MDP to a discounted reward MDP mentioned in [5] to get rid of the contraction measure $\nu$?

4. Why the paper is consistently using "backward induction algorithm" to name the well-know value itration algorithm in RL?

5. What is $\mathcal{L}0$ in line 1993?

6. It would be better to provide extra sections in the appendix to introduce the algorithms and summarize the results since the algorithms are not given in the main context, instead of mixing the algorithms and the full proof into a single section.

[4]. Ayoub et al., Model-Based Reinforcement Learning with Value-Targeted Regression.

[5]. Zhang et al., Sharper Model-free Reinforcement Learning for Average-reward Markov Decision Processes.

---

> ### Author Response · Authors · 2025-11-21
>
> We thank the reviewer for all the useful comments, which will be helpful in improving our paper. Below we try to address all of the reviewer's comments.
>
> Addressing main weaknesses:
>
> 1. We cordially and firmly disagree with the reviewer that Ref. [1] does not assume access to a generative model nor assume the data collection phase incurs no regret. Ref. [1] absolutely splits the learning model into an exploration phase where regret is incurred and a generative phase with no regret. Every time the algorithm from Ref. [1] updates the tentative probability distribution via quantum multi-dimensional amplitude estimation, it enters into a regret-free phase which was not clearly pointed out by the authors. We discuss the RL model from Ref. [1] in Appendix B and refer the reviewer to it. The only difference between both RL models is that we can manipulate the environment state within the generative phase, while Ref. [1] assumes that the environment state is fixed when moving from an exploration to a generative phase. This means that Ref. [1] must shift between phases much more often than us, and is the reason for their lazy update idea. Since we assume a bit more freedom in the generative phase, there is no need for the lazy update. It is a good open question whether we can incorporate the lazy update idea into our work if we restrict our generative phase a bit more like Ref. [1] (it might involve performing value iteration in a streaming-like fashion), but we would like to stress that Ref. [1] definitely assumes a generative phase without regret. Actually, we are not aware of any quantum RL model with $polylog(T)$ regret that does not assume a regret-free phase. We deem the existence of a quantum RL model without a generative phase to be a very important open problem.
>
> 2. We were not aware of Ref. [5], but from a quick look at their results we could spot the presence of the "effective horizon" $(1-\gamma)^{-1}$ of the discounted MDP in several complexities, e.g., Lemma 8, and $\gamma$ basically works as a contraction measure similar to $\nu$. Therefore we are not sure about the validity of this reviewer's point. Regardless, the main function of $\nu$ is to argue that value iteration converges after a finite amount of steps and is a standard assumption in the literature, e.g., we refer the reviewer to Fruit et al. (2018) and to Chapters 8 and 9 from Puterman's book "Markov decision processes: Discrete stochastic dynamic programming".
>
> Questions:
>
> 1. In our algorithms the data collected in the exploration phase are not employed anywhere, but it is actually a good question whether they can provide any advantage! It might be possible to incorporate them into the value iteration algorithm in a meaningful way, but we have not explored this direction. We do not believe, though, that it can provide any advantage in the quantum case given the different complexities (the exploration data is classical and would probably lead to a $\sqrt{\varepsilon}$-optimal policy versus the $\varepsilon$-optimal policy from the quantum value iteration).
>
> 2. We tackled the tabular and continuous cases together, in a unified way! All cases, including the case $\mathcal{X}=[0,1]^D$, can be solved by considering an $\epsilon$-net $\mathcal{S}_n$. Obviously, in the finite case, the $\epsilon$-net can be set as the state space itself, so $\mathcal{S}_n = \mathcal{X}$ in the finite case. We didn't consider $\epsilon$-nets in the tabular case because they can lead to an improvement, but just to tackle both cases, finite and continuous, together! We obviously didn't want to have one section for tabular MDPs and another section for continuous MDPs, it would be redundant. And the reason to consider continuous MDPs was to compare our results with Ortner \& Ryabko (2012) and Lakshmanan et al. (2015).
>
> 3. It is a good question whether the dependence on $\nu$ can be removed and we would have to read Ref. [5] carefully as we were not aware of it, but as mentioned above, we are not sure they completely remove the contraction measure $\nu$ (although their techniques could improve our complexities).
>
> 4. If we employed the name "backward induction algorithm" it was due to Puterman's book mentioned above, see its Section 4.5, but we are happy to just employ value iteration.
>
> 5. $\mathcal L 0$ is the operator $\mathcal L$ applied to the all-zeroes vector represented by $0$. We should make this clearer.
>
> 6. We don't fully understand the reviewer's proposal on providing extra sections in the appendix to introduce the algorithms. Doesn't Appendix C introduce and summarise the main techniques regarding value iteration for finite-horizon MDPs, while Appendix D does the same for infinite-horizon MDPs? And don't Appendices E and F do the same for online learning of finite and infinite-horizon MDPs? Would the reviewer like clear and informal statements of the main complexities right at the beginning of the appendices?

---

### Author Response · Authors · 2025-12-01
**Summary and Conclusions**

We would like to use this space to summarise our work and clarify some misunderstandings from the reviewers. In our work:
- We carefully formalise and explore a modified reinforcement learning model wherein the agent can freely interact with the environment  from time to time without incurring in any regret (offline mode), while most of the time the agent interacts with the environment in a standard (classical) fashion with regret (online mode).
- We propose a quantum algorithm for efficiently computing an approximate optimal policy for *finite-horizon MDPs* given quantum sampling access to the transition probabilities. Alongside the very recent and concurrent work of Luo et al. (ICML'25), both works are the first to present such an algorithm (our complexity is actually slightly better than Luo et al. (ICML'25)).
- We propose the **first** quantum algorithm for efficiently computing an approximate optimal policy for *infinite-horizon average-reward MDPs* given quantum sampling access to the transition probabilities.
- We employ our novel quantum and prior classical algorithms for optimal policies within our RL model to obtain new and improved regret bounds.
- Our classical regret bounds, both for finite-horizon and infinite-horizon MDPs, are better than prior bounds in the literature with respect to the size of the action and state spaces, while maintaining the $O(\sqrt{T})$ dependence on the number of time steps $T$.
- Our finite-horizon quantum regret bound improves upon previous works by Ganguly et al. (arXiv’23) and Zhong et al. (ICML’24) with respect to the size of the action and state spaces, while still maintaining the $O(\log{T})$ dependence, exponentially better than classical bounds.
- Our infinite-horizon quantum regret bound is **completely novel** and improves prior classical bounds with respect to several parameters but still maintains the $O(\sqrt{T})$ dependence.
- We propose a **novel measure of regret** with respect to which our infinite-horizon quantum algorithm has $O(\log{T})$ regret bound, exponentially better than its classical counterparts. We are not aware of another result where the employed measure of regret is crucial to properly gauge the performance of an algorithm.
- We generalise all of the above results to the case when the state space is continuous and not finite.
---
Based on the reviewers comments, we would like to clarify a few points regarding our (quantum) RL model:
- The regret-free interactions between agent and environment are limited by previous standard interactions where the agent *does* incur into regret, meaning that there is still an exploration-exploitation dilemma present in the model.
- Our hybrid model can be defined classically or quantumly depending on how the agent interacts freely with the environment.
- If the standard (classical) RL environment can be viewed as a computer program with a source code, then it is possible to transform such a source code into a quantum oracle by simply translating the code into a Boolean circuit that draws samples from the distribution $p(\cdot|s,a)$ and by then employing standard techniques in quantum computing to convert such a Boolean circuit into a quantum circuit.
- The quantum version of our RL model solves the fundamental problem that the regret is not properly defined when the agent interacts with the environment in a quantum fashion.
- Our RL model could describe the interaction between an agent and a (black-box) quantum computer in an online-fashion wherein the agent has limited access to the quantum computer due to, e.g., high traffic or monetary reasons, interspersed with periods of free interaction due to the quantum computer being idle.
- **There is currently no meaningful and useful quantum RL model that does not assume a regret-free interaction phase!** The existence of such a RL model is an *open question*.

---
Finally, we would like to address and point out *objectively wrong* statements made by some reviewers:
- Reviewer 6JTM argued that the model of Zhong et al. (ICML'24) does not have a regret-free interaction phase. This is clearly wrong as carefully explained in our Appendix B.
- Reviewer 6JTM mentioned the work of Zhang et al. to suggest that the dependence on $\nu$ can be removed, but the results of Zhang et al. also seem to depend on $\nu$.
- Reviewer 6JTM argued that the use of $\epsilon$-nets didn't improve our results and is thus unnecessary, but we used them simply to unify the analysis of both finite and continuous state space MDPs.
- Reviewer Mu5H believes that quantum oracles have been explored extensively, but this is far from the truth in the *undiscounted* setting. Almost all of the cited works in our paper are in the *discounted* setting.

---

### Meta-Review · Area_Chair_DT3V · 2025-12-10

**Summary:**

This paper presents rigorous classical and quantum algorithms for finite- and infinite-horizon MDPs within an exploration–generative RL model, with strong technical development and valuable finite-horizon improvements. However, the submission overstates novelty in the infinite-horizon quantum setting and omits prior closely related work (e.g., "Quantum Speedups in Regret Analysis of Infinite Horizon Average-Reward Markov Decision Processes," ICML 2025) that already reports exponential quantum improvements for average-reward MDPs. The paper therefore does not correctly position its main claimed contribution. In addition, the proposed hybrid RL model remains insufficiently justified, and several reviewers noted that allowing regret-free generative phases weakens comparability to standard online RL. While the analyses are sound, these unresolved issues prevent confidence in the claimed significance. Reviewer concerns regarding novelty, model realism, and regret interpretation remain largely outstanding after rebuttal.

**Reviewer Concerns:**

The authors successfully clarified several technical points raised by reviewers, including the role of the contraction parameter in the infinite-horizon analysis, the purpose of Hölder nets and why they unify the tabular and continuous settings, and the origin of the terminology around “backward induction.”

Major concerns about the online RL model remain unresolved. Reviewers questioned whether allowing a regret-free generative phase is consistent with standard online exploration, whether it undermines the exploration–exploitation challenge, and why the length of this phase should scale with the preceding exploration phase; the rebuttal defends the model but does not materially justify its realism or necessity. Reviewers also raised conceptual concerns about the appropriateness and motivation of the proposed regret formulation in the quantum setting, which were not fully clarified. Finally, the broader issue of how to interpret the claimed improvements within this nonstandard RL framework remains outstanding, as reviewers continue to view the modeling choice as limiting the impact and comparability of the results.

**Reviewer Scores:**

I do not expect scores would have changed much.

---

### Decision · Program_Chairs · 2026-01-26

Reject